# Exploring the Noise Robustness of Online Conformal Prediction

**Huajun Xi[1], Kangdao Liu[1,2], Hao Zeng[1], Wenguang Sun[3], Hongxin Wei[1]***

[1]Department of Statistics and Data Science, Southern University of Science and Technology
[2]Department of Computer and Information Science, University of Macau
[3]Center for Data Science, Zhejiang University

## Abstract

Conformal prediction is an emerging technique for uncertainty quantification that constructs prediction sets guaranteed to contain the true label with a predefined probability. Recent work develops online conformal prediction methods that adaptively construct prediction sets to accommodate distribution shifts. However, existing algorithms typically assume *perfect label accuracy* which rarely holds in practice. In this work, we investigate the robustness of online conformal prediction under uniform label noise with a known noise rate. We show that label noise causes a persistent gap between the actual mis-coverage rate and the desired rate $\alpha$, leading to either overestimated or underestimated coverage guarantees. To address this issue, we propose a novel loss function *robust pinball loss*, which provides an unbiased estimate of clean pinball loss without requiring ground-truth labels. Theoretically, we demonstrate that robust pinball loss enables online conformal prediction to eliminate the coverage gap under uniform label noise, achieving a convergence rate of $\mathcal{O}(T^{-1/2})$ for both empirical and expected coverage errors (i.e., absolute deviation of the empirical and expected mis-coverage rate from the target level $\alpha$). This loss offers a general solution to the uniform label noise, and is complementary to existing online conformal prediction methods. Extensive experiments demonstrate that robust pinball loss enhances the noise robustness of various online conformal prediction methods by achieving a precise coverage guarantee and improved efficiency.

## 1 Introduction

Machine learning techniques are revolutionizing decision-making in high-stakes domains, such as autonomous driving [1] and medical diagnostics [2]. It is crucial to ensure the reliability of model predictions in these contexts, as wrong predictions can result in serious consequences. While various techniques have been developed for uncertainty estimation, including confidence calibration [3] and Bayesian neural networks [4], they typically lack rigorous theoretical guarantees. *Conformal prediction* addresses this limitation by establishing a systematic framework to construct prediction sets with provable coverage guarantee [5, 6, 7, 8]. Notably, this framework requires no parametric assumptions about the data distribution and can be applied to any black-box predictor, which makes it a powerful technique for uncertainty quantification.

Recent research extends conformal prediction to *online* setting where the data arrives in a sequential order [9, 10, 11, 12, 13, 14]. These methods provably achieve the desired coverage property under arbitrary distributional changes. However, previous studies typically assume *perfect label accuracy*, an assumption that seldom holds true in practice due to the common occurrence of noisy labels in online learning [15, 16, 17]. Recent work [18] proves that online conformal prediction can achieve a

---

*Correspondence to: Hongxin Wei <weihx@sustech.edu.cn>

39th Conference on Neural Information Processing Systems (NeurIPS 2025).

conservative coverage guarantee under uniform label noise, leading to unnecessarily large prediction sets. However, their analysis relies on a strong distributional assumption of non-conformity score and cannot quantify the specific deviation of coverage guarantees. These limitations motivate us to establish a general theoretical framework for this problem and develop a noise-robust algorithm that maintains precise coverage guarantees while producing small prediction sets.

In this work, we present a general theoretical framework for analyzing how uniform label noise affects the performance of standard online conformal prediction, i.e., adaptive conformal inference (dubbed ACI) [9]. Notably, our theoretical results are independent of the distributional assumptions on the non-conformity scores made in the previous work [18]. In particular, we demonstrate that label noise causes a persistent gap between the actual mis-coverage rate and the desired rate $\alpha$, with higher noise rates resulting in larger gaps. This gap can lead to either overestimated or underestimated coverage guarantees, which depend on the size of the prediction sets (see Proposition 3.1).

To address this challenge, we propose a novel loss function *robust pinball loss*, which provides an unbiased estimate of clean pinball loss value without requiring access to ground-truth labels. Specifically, we construct the robust pinball loss as a weighted combination of the pinball loss computed with respect to noisy scores and the pinball loss with scores of all classes. We prove that this loss is equivalent to the pinball loss under clean labels in expectation. Theoretically, we demonstrate that robust pinball loss enables ACI to eliminate the coverage gap under uniform label noise. It achieves a convergence rate of $\mathcal{O}(T^{-1/2})$ for both empirical and expected coverage errors (i.e., absolute deviation of the empirical and expected mis-coverage rate from the target level $\alpha$). Notably, our robust pinball loss offers a general solution to the uniform label noise, and is complementary to existing online conformal prediction methods.

To verify the effectiveness of the robust pinball loss, we conduct extensive experiments on CIFAR-100 [19] and ImageNet [20] with synthetic uniform label noise. In particular, we integrate the proposed loss into ACI with constant [9] and dynamic learning rates [12], and strongly adaptive online conformal prediction [11]. Empirical results show that the robust pinball loss enhances the noise robustness of online conformal prediction by eliminating the coverage gap caused by the label noise. Thus, these methods achieve both long-run coverage and local coverage rate that are close to the target $1 - \alpha$, and improved prediction set efficiency. For example, on ImageNet with error rate $\alpha = 0.1$, noise rate $\epsilon = 0.15$ and dynamic learning rate, ACI with the standard pinball loss deviates from the target coverage level of $0.9$, exhibiting a coverage gap of $8.372\%$ and an average set size of $171.2$. In contrast, ACI equipped with the robust pinball loss achieves a negligible coverage gap of $0.183\%$ and a prediction set size of $13.10$. In summary, our method consistently enhances the noise robustness of various online conformal prediction methods by achieving a precise coverage guarantee and improved efficiency.

We summarize our contributions as follows:

- We present a general theoretical framework for analyzing the effect of uniform label noise on the coverage of online conformal prediction. Our theoretical results are independent of the distributional assumptions made in the previous work [18].

- To address the issue of label noise, we propose a novel loss function – *robust pinball loss* that enhances the noise robustness of online conformal prediction. This loss is complementary to online conformal prediction algorithms and can be seamlessly integrated with these methods.

- We empirically validate that our method can be applied to various online conformal prediction methods and non-conformity score functions. It is straightforward to implement and does not require sophisticated changes to the framework of online conformal prediction.

## 2 Preliminary

**Online conformal prediction.** We study the problem of generating prediction sets in *online* classification where the data arrives in a sequential order [9, 12]. Formally, we consider a sequence of data points $(X_t, Y_t)$, $t \in \mathbb{N}^+$, which are sampled from a joint distribution $\mathcal{P}_{\mathcal{X}\mathcal{Y}}$ over the input space $\mathcal{X} \subset \mathbb{R}^d$, and the label space $\mathcal{Y} = \{1, \ldots, K\}$. In online conformal prediction, the goal is to construct prediction sets $\mathcal{C}_t(X_t)$, $t \in \mathbb{N}^+$, that provides *precise* coverage guarantee: $\lim_{T \to +\infty} \frac{1}{T} \sum_{t=1}^{T} \mathbb{1}\{Y_t \notin \mathcal{C}_t(X_t)\} = \alpha$, where $\alpha \in (0, 1)$ denotes a user-specified error rate.

At each time step $t$, we construct a prediction set $\mathcal{C}_t(X_t)$ by

$$\mathcal{C}_t(X_t) = \{y \in \mathcal{Y} \,:\, \mathcal{S}(X_t, y) \leq \hat{\tau}_t\}. \tag{1}$$

where $\hat{\tau}_t$ is a data-driven threshold, and $\mathcal{S} : \mathcal{X} \times \mathcal{Y} \to \mathbb{R}$ denotes a *non-conformity score* function that measures the deviation between a data sample and the training data. For example, given a pre-trained classifier $f : \mathcal{X} \to \mathbb{R}^K$, the LAC score [21] is defined as $\mathcal{S}(X, Y) = 1 - \hat{\pi}_Y(X)$, where $\hat{\pi}_Y(X) = \sigma_Y(f(X))$ denotes the softmax probability of instance $X$ for class $Y$, and $\sigma$ is the softmax function. For notation shorthand, we use $S_t$ to denote the random variable $\mathcal{S}(X_t, Y_t)$ and use $S_{t,y}$ to denote $\mathcal{S}(X_t, y)$ for a given class $y \in \mathcal{Y}$. Following previous work [12, 22], we will assume that the non-conformity score function is bounded, and the threshold is specifically initialized:

**Assumption 2.1.** The score is bounded by $\mathcal{S}(\cdot, \cdot) \in [0, 1]$.

**Assumption 2.2.** The threshold is initialized by $\hat{\tau}_1 \in [0, 1]$.

In online conformal prediction, a representative method is adaptive conformal inference (ACI) [9], which updates the threshold $\hat{\tau}_t$ with *pinball loss*:

$$l_{1-\alpha}(\tau, s) = \alpha(\tau - s)\mathbb{1}\{\tau \geq s\} + (1-\alpha)(s - \tau)\mathbb{1}\{\tau \leq s\},$$

where $\tau$ denotes a threshold and $s$ is a non-conformity score. As the label $Y_t$ of $X_t$ is observed after model prediction, the threshold is then updated via *online gradient descent*:

$$\hat{\tau}_{t+1} = \hat{\tau}_t - \eta \cdot \nabla_{\hat{\tau}_t} l_{1-\alpha}(\hat{\tau}_t, S_t) = \hat{\tau}_t + \eta \cdot (\mathbb{1}\{Y_t \notin \mathcal{C}_t(X_t)\} - \alpha) \tag{2}$$

where $\nabla_\tau l_{1-\alpha}(\tau, s)$ denotes the gradient of pinball loss w.r.t the threshold $\tau$, and $\eta > 0$ is the learning rate. The optimization will increase the threshold if the prediction set $\mathcal{C}_t(X_t)$ fails to encompass the label $Y_t$, resulting in more conservative predictions in future instances (and vice versa).

We use the *empirical coverage error* and the *expected coverage error* to evaluate the coverage performance. The empirical coverage error measures the absolute deviation of the mis-coverage rate from the target level $\alpha$, while the expected coverage error quantifies the absolute deviation in expectation. In particular, for any $T \in \mathbb{N}^*$, we define

$$\text{EmErr}(T) = \left| \frac{1}{T} \sum_{t=1}^{T} \mathbb{1}\{Y_t \notin \mathcal{C}_t(X_t)\} - \alpha \right|, \quad \text{ExErr}(T) = \left| \frac{1}{T} \sum_{t=1}^{T} \mathbb{P}\{Y_t \notin \mathcal{C}_t(X_t)\} - \alpha \right|.$$

**Uniform Label noise.** In this paper, we focus on the issue of noisy labels in online learning, a common occurrence in the real world. This is primarily due to the dynamic nature of real-time data streams and the potential for human error or sensor malfunctions during label collection. Let $(X_t, \tilde{Y}_t)$ be the data sequence with label noise, and $\tilde{S}_t = \mathcal{S}(X_t, \tilde{Y}_t)$ be the noisy non-conformity score. In this work, we focus on the setting of uniform label noise [18, 23], i.e., the correct label is replaced by a label that is randomly sampled from the $K$ classes with a fixed probability $\epsilon \in (0, 1)$:

$$\tilde{Y}_t = Y_t \cdot \mathbb{1}\{U \geq \epsilon\} + \bar{Y} \cdot \mathbb{1}\{U \leq \epsilon\},$$

where $U$ is uniformly distributed over $[0, 1]$, and $\bar{Y}$ is uniformly sampled from the set of classes $\mathcal{Y}$. We assume the probability $\epsilon$ (i.e., the noise rate) is known, in alignment with prior works [18, 23, 24]. This assumption is practical as the noise rate can be estimated from historical data [25, 26, 27].

Recent work [18] investigates the noise robustness of online conformal prediction under uniform label noise, with a strong distributional assumption. Their analysis demonstrates that noisy labels will lead to a conservative long-run coverage guarantee, with the assumption that noisy score distribution stochastically dominates the clean score distribution, i.e., $\mathbb{P}\{\tilde{S} \leq s\} \leq \mathbb{P}\{S \leq s\}, \forall s \in \mathbb{R}$. The distributional assumption is too strong to ensure valid coverage under general cases. Moreover, their analysis fails to quantify the specific deviation of coverage guarantees. These limitations motivate us to establish a general theoretical framework for this problem and develop a noise-robust algorithm for online conformal prediction.

## 3 The impact of label noise on online conformal prediction

In this section, we theoretically analyze the impacts of uniform label noise on ACI [9]. In this case, the threshold $\hat{\tau}_t$ is updated by

$$\hat{\tau}_{t+1} = \hat{\tau}_t - \eta \cdot \nabla_{\hat{\tau}_t} l_{1-\alpha}(\hat{\tau}_t, \tilde{S}_t), \tag{3}$$

where $\tilde{S}_t$ is the noisy non-conformity score. As shown in Eq. (3), the essence of online conformal prediction is to update the threshold $\hat{\tau}_t$ with the gradient of pinball loss. However, the gradient estimates can be biased if the observed labels are corrupted. Formally, with high probability:

$$\nabla_{\hat{\tau}_t} l_{1-\alpha}(\hat{\tau}_t, S_t) \neq \nabla_{\hat{\tau}_t} l_{1-\alpha}(\hat{\tau}_t, \tilde{S}_t).$$

This bias in gradient estimation can result in two potential consequences: conformal predictors may either fail to maintain the desired coverage or suffer from reduced efficiency (i.e., generating large prediction sets). We formalize these consequences in the following proposition:

**Proposition 3.1.** *Consider online conformal prediction under uniform label noise with noise rate $\epsilon \in (0, 1)$. Given Assumptions 2.1 and 2.2, when updating the threshold according to Eq. (3), then for any $\delta \in (0, 1)$ and $T \in \mathbb{N}^+$, the following bound holds with probability at least $1 - \delta$:*

$$\alpha - \frac{1}{T} \sum_{t=1}^{T} \mathbb{1}\left\{Y_t \notin \mathcal{C}_t(X_t)\right\} \leq \frac{A}{\sqrt{T}} + \frac{B}{T} + \frac{\epsilon}{1-\epsilon} \cdot \frac{1}{T} \sum_{t=1}^{T} \left((1-\alpha) - \frac{1}{K}\mathbb{E}\left[|\mathcal{C}_t(X_t)|\right]\right),$$

*where*

$$A = \frac{2-\epsilon}{1-\epsilon} \sqrt{2 \log\left(\frac{4}{\delta}\right)} \quad and \quad B = \frac{1}{1-\epsilon}\left(\frac{1}{\eta} + 1 - \alpha\right).$$

**Interpretation.** The proof is provided in Appendix E.1. In Proposition 3.1, we evaluate the long-run mis-coverage rate using $\frac{1}{T} \sum_{t=1}^{T} \mathbb{1}\{Y_t \notin \mathcal{C}_t(X_t)\}$. Our analysis shows that label noise introduces a coverage gap of

$$\frac{\epsilon}{1-\epsilon} \cdot \frac{1}{T} \sum_{t=1}^{T} \left((1-\alpha) - \frac{1}{K}\mathbb{E}\left[|\mathcal{C}_t(X_t)|\right]\right),$$

between the actual mis-coverage rate and desired mis-coverage rate $\alpha$ (the term $\frac{A}{\sqrt{T}} + \frac{B}{T}$ will diminish and eventually approach zero, as $T$ increases). In particular, the effect of label noise on the coverage guarantee can manifest in two distinct scenarios, depending on the prediction set size:

- When the prediction sets are small such that $\mathbb{E}[|\mathcal{C}_t(X_t)|] \leq K(1-\alpha)$, label noise can result in *over-coverage* of prediction sets: $\frac{1}{T} \sum_{t=1}^{T} \mathbb{1}\{Y_t \notin \mathcal{C}_t(X_t)\} \leq \alpha$. In this scenario, a higher noise rate $\epsilon$ results in a larger coverage gap.

- When the prediction sets are large such that $\mathbb{E}[|\mathcal{C}_t(X_t)|] \geq K(1-\alpha)$, label noise causes *under-coverage* of prediction sets. This situation generally occurs only when the model significantly underperforms on the task, which is uncommon in practical applications (See details in Appendix B.3)).

**Empirical verification.** To verify our theoretical result, we compare the performance of ACI under different noise rates. In particular, we compute the long-run coverage, defined as $\text{Cov}(T) = \frac{1}{T} \sum_{t=1}^{T} \mathbb{1}\{Y_t \in \mathcal{C}_t(X_t)\}$, under noise rate $\epsilon \in \{0.05, 0.1, 0.15\}$, with a ResNet18 model on CIFAR-100 and ImageNet datasets. We employ LAC score [21] to generate prediction sets with error rate $\alpha = 0.1$, and use noisy labels to update the threshold with a constant learning rate $\eta = 0.05$. The experimental results in Figure 1 validate our theoretical analysis: label noise introduces discrepancies between the actual and target coverage rates $1 - \alpha$, with higher noise rates resulting in a more pronounced coverage gap.

Overall, our results show that label noise significantly impacts the coverage guarantee of online conformal prediction. The size of the prediction set determines whether coverage is inflated or deflated, and the noise rate controls how much coverage is changed. In the following, we introduce a robust pinball loss, which addresses the issue of label noise.

## 4 Method

### 4.1 Robust pinball loss

Our theoretical analysis establishes that biased gradient estimates arising from label noise can significantly impact the coverage properties of online conformal prediction. Therefore, the key

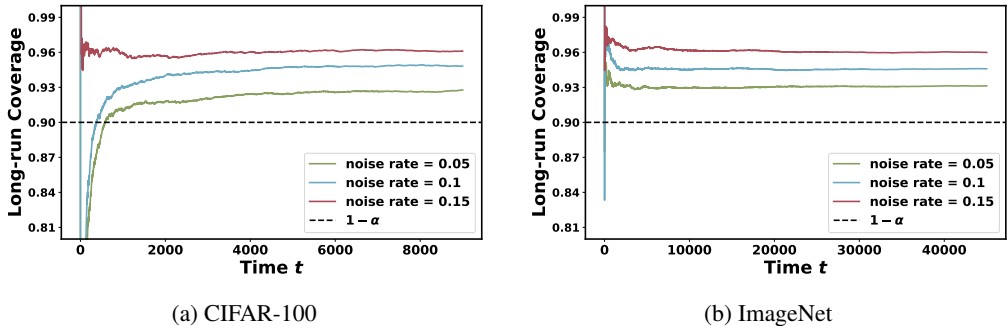

(a) CIFAR-100            (b) ImageNet

Figure 1: Performance of ACI [9] with a constant learning rate $\eta = 0.05$ under different noise rates, using ResNet18 on CIFAR-100 and ImageNet datasets. The results validate that label noise introduces a coverage gap, with higher noise rates resulting in a more pronounced gap.

challenge of the noisy setting lies in how to obtain unbiased gradient estimates without requiring ground-truth labels. In this work, we propose a novel loss function - *robust pinball loss*, which provides an unbiased estimate of the clean pinball loss value without requiring access to ground-truth labels. We begin by developing the intuition behind how to approximate the clean pinball loss under uniform label noise.

Consider a data sample $(X, Y)$ with a noisy label $\tilde{Y}$, we denote the clean non-conformity score as $S = \mathcal{S}(X, Y)$, the noisy score as $\tilde{S} = \mathcal{S}(X, \tilde{Y})$, and the score for an arbitrary class $y \in \mathcal{Y}$ as $S_y = \mathcal{S}(X, y)$. Under a uniform label noise with noise rate $\epsilon \in (0, 1)$, the distributions of these scores have the following relationship: $\mathbb{P}\{S \leq s\} = \frac{1}{1-\epsilon}\mathbb{P}\{\tilde{S} \leq s\} - \frac{\epsilon}{K(1-\epsilon)}\sum_{y=1}^{K}\mathbb{P}\{S_y \leq s\}$, for an arbitrary number $s \in \mathbb{R}$. We formally establish this equation in Lemma F.1 with a rigorous proof. This correlation motivates the following approximation:

$$\mathbb{1}\{S \leq s\} \approx \frac{1}{1-\epsilon}\mathbb{1}\{\tilde{S} \leq s\} - \frac{\epsilon}{K(1-\epsilon)}\sum_{y=1}^{K}\mathbb{1}\{S_y \leq s\}.$$

The above decomposition suggests that the clean pinball loss can be approximated by replacing its indicator function with the above expression. Inspired by this, we propose the *robust pinball loss* as:

$$\tilde{l}_{1-\alpha}(\tau, \tilde{S}, \{S_y\}_{y=1}^{K}) = \frac{1}{1-\epsilon}l_{1-\alpha}(\tau, \tilde{S}) - \frac{\epsilon}{K(1-\epsilon)}\sum_{y=1}^{K}l_{1-\alpha}(\tau, S_y). \tag{4}$$

The following theoretical properties demonstrate how this loss function mitigates label noise bias:

**Proposition 4.1.** *The robust pinball loss defined in Eq.* (4) *satisfies the following two properties:*

$$(1)\mathbb{E}_S\left[l_{1-\alpha}(\tau, S)\right] = \mathbb{E}_{\tilde{S}, S_y}\left[\tilde{l}_{1-\alpha}(\tau, \tilde{S}, \{S_y\}_{y=1}^{K})\right],$$
$$(2)\mathbb{E}_S\left[\nabla_\tau l_{1-\alpha}(\tau, S)\right] = \mathbb{E}_{\tilde{S}, S_y}\left[\nabla_\tau \tilde{l}_{1-\alpha}(\tau, \tilde{S}, \{S_y\}_{y=1}^{K})\right].$$

**Interpretation.** The proof can be found in Appendix E.2. In Proposition 4.1, the first property ensures that our robust pinball loss matches the expected value of the true pinball loss, while the second guarantees that the gradients of both losses have the same expectation. These properties establish that updating the threshold with robust pinball loss is equivalent to updating with clean pinball loss in expectation.

In summary, Proposition 4.1 establishes the validity of our robust pinball loss *in expectation*. Notably, our robust pinball loss offers a general solution to the uniform label noise, and is complementary to existing online conformal prediction methods. In the following sections, we apply the robust pinball loss to ACI with constant [9] and dynamic learning rates [12]. We will show that by updating the threshold with the proposed loss, ACI eliminates the coverage gap caused by the label noise.

## 4.2 Convergence with constant learning rate

We now analyze the convergence of the coverage rate of ACI under uniform label noise with a constant learning rate schedule [9]. In particular, we update the threshold of ACI with respect to the robust pinball loss:

$$
\hat{\tau}_{t+1} = \hat{\tau}_t - \eta \cdot \nabla_{\hat{\tau}_t} \tilde{l}_{1-\alpha}(\hat{\tau}_t, \tilde{S}_t, \{S_{t,y}\}_{y=1}^K)
$$

$$
= \hat{\tau}_t - \eta \cdot \frac{1}{1-\epsilon} \left[ \mathbb{1}\left\{ \tilde{S}_t \le \hat{\tau}_t \right\} - (1-\alpha) \right] + \eta \cdot \frac{\epsilon}{K(1-\epsilon)} \sum_{y=1}^K [\mathbb{1}\left\{ S_{t,y} \le \hat{\tau}_t \right\} - (1-\alpha)].
$$

(5)

For notation shorthand, we denote:

$$
\mathbb{E}\left[ \nabla_{\hat{\tau}_t} l_{1-\alpha} \right] = \mathbb{E}_{S_t}\left[ \nabla_{\hat{\tau}_t} l_{1-\alpha}(\hat{\tau}_t, S_t) \right], \quad \mathbb{E}\left[ \nabla_{\hat{\tau}_t} \tilde{l}_{1-\alpha} \right] = \mathbb{E}_{\tilde{S}_t, S_{t,y}}\left[ \nabla_{\hat{\tau}_t} \tilde{l}_{1-\alpha}(\hat{\tau}_t, \tilde{S}_t, \{S_{t,y}\}_{y=1}^K) \right].
$$

We first present the results for expected coverage error:

**Proposition 4.2.** *Under the same assumptions in Proposition 3.1, when updating the threshold according to Eq. (5), then for any $\delta \in (0,1)$ and $T \in \mathbb{N}^+$, the following bound holds with probability at least $1 - \delta$:*

$$
\mathrm{ExErr(T)} \le \sqrt{\frac{\log(2/\delta)}{1-\epsilon}} \cdot \frac{1}{\sqrt{T}} + \left( \frac{1}{\eta} - \alpha + \frac{\epsilon}{1-\epsilon} \right) \cdot \frac{1}{T}.
$$

**Proof Sketch.** The proof (in Appendix E.3) relies on the following decomposition:

$$
\mathrm{ExErr(T)} = \left| \sum_{t=1}^T \mathbb{E}\left[ \nabla_{\hat{\tau}_t} l_{1-\alpha} \right] \right| \le \underbrace{\left| \sum_{t=1}^T \mathbb{E}\left[ \nabla_{\hat{\tau}_t} \tilde{l}_{1-\alpha} \right] - \sum_{t=1}^T \nabla_{\hat{\tau}_t} \tilde{l}_{1-\alpha} \right|}_{(a)} + \underbrace{\left| \sum_{t=1}^T \nabla_{\hat{\tau}_t} \tilde{l}_{1-\alpha} \right|}_{(b)}.
$$

Part (a) converges to zero in probability at rate $\mathcal{O}(T^{-1/2})$ by the Azuma–Hoeffding inequality, and part (b) achieves a convergence rate of $\mathcal{O}(T^{-1})$ following standard online conformal prediction theory. Combining the two parts establishes the desired upper bound.

Building on Proposition 4.2, we now provide an upper bound for the empirical coverage error:

**Proposition 4.3.** *Under the same assumptions in Proposition 3.1, when updating the threshold according to Eq. (5), then for any $\delta \in (0,1)$ and $T \in \mathbb{N}^+$, the following bound holds with probability at least $1 - \delta$:*

$$
\mathrm{EmErr(T)} \le \frac{2-\epsilon}{1-\epsilon} \sqrt{2 \log\left( \frac{4}{\delta} \right)} \cdot \frac{1}{\sqrt{T}} + \left( \frac{1}{\eta} - \alpha + \frac{\epsilon}{1-\epsilon} \right) \cdot \frac{1}{T}.
$$

**Proof Sketch.** The proof (detailed in Appendix E.4) employs a similar decomposition:

$$
\mathrm{EmErr(T)} = \left| \sum_{t=1}^T \nabla_{\hat{\tau}_t} l_{1-\alpha} \right| \le \underbrace{\left| \sum_{t=1}^T \nabla_{\hat{\tau}_t} l_{1-\alpha} - \sum_{t=1}^T \mathbb{E}\left[ \nabla_{\hat{\tau}_t} l_{1-\alpha} \right] \right|}_{(a)} + \underbrace{\left| \sum_{t=1}^T \mathbb{E}\left[ \nabla_{\hat{\tau}_t} l_{1-\alpha} \right] \right|}_{(b)}.
$$

The analysis shows that both terms achieve $\mathcal{O}(T^{-1/2})$ convergence: part (a) through the Azuma–Hoeffding inequality, and part (b) via Proposition 4.2.

**Remark 4.4.** It is worth noting that our method achieves a $\mathcal{O}(T^{-1/2})$ convergence rate for empirical coverage error even in the absence of noise (i.e., $\epsilon = 0$), which is slightly slower than the $\mathcal{O}(T^{-1})$ rate achieved by standard online conformal prediction theory [9, 12]. This is because our analysis relies on martingale-based concentration to handle label noise, leading to the $\mathcal{O}(T^{-1/2})$ rate.

## 4.3 Convergence with dynamic learning rate

Recent work [12] highlights a limitation of constant learning rates: while coverage holds on average over time, the *instantaneous* coverage rate $\mathrm{Cov}(\hat{\tau}_t) = \mathbb{P}\left\{ S \le \hat{\tau}_t \right\}$ would exhibit substantial temporal

variability (see Proposition 1 in [12]). Thus, they extend ACI to *dynamic* learning rate schedule where $\eta_t$ can change over time for updating the threshold. In this section, we apply the robust pinball loss to the dynamic learning rate schedule. Specifically, we update the threshold by:

$$
\begin{aligned}
\hat{\tau}_{t+1} &= \hat{\tau}_t - \eta_t \cdot \nabla_{\hat{\tau}_t} \tilde{l}_{1-\alpha}(\hat{\tau}_t, \tilde{S}_t, \{S_{t,y}\}_{y=1}^K) \\
&= \hat{\tau}_t - \eta_t \cdot \frac{1}{1-\epsilon}\left[\mathbb{1}\left\{\tilde{S}_t \leq \hat{\tau}_t\right\} - (1-\alpha)\right] + \eta_t \cdot \frac{\epsilon}{K(1-\epsilon)}\sum_{y=1}^K\left[\mathbb{1}\left\{S_{t,y} \leq \hat{\tau}_t\right\} - (1-\alpha)\right].
\end{aligned}
\tag{6}
$$

For the convergence of coverage rate, our theoretical results show that, under dynamic learning rate, NR-OCP achieves convergence rates of $\mathrm{EmCovErr(T)} = \mathcal{O}(T^{-1/2})$ and $\mathrm{ExCovErr(T)} = \mathcal{O}(T^{-1/2})$. The proofs are presented in Appendix E.5 and E.6.

**Proposition 4.5.** *Under the same assumptions in Proposition 3.1, when updating the threshold according to Eq. (6), then for any $\delta \in (0,1)$ and $T \in \mathbb{N}^+$, the following bound holds with probability at least $1 - \delta$:*

$$
\mathrm{ExErr(T)} \leq \sqrt{\frac{\log(2/\delta)}{1-\epsilon}} \cdot \frac{1}{\sqrt{T}} + \left[\left(1 + \max_{1 \leq t \leq T-1}\eta_t \cdot \frac{1+\epsilon}{1-\epsilon}\right)\sum_{t=1}^T\left|\eta_t^{-1} - \eta_{t-1}^{-1}\right|\right] \cdot \frac{1}{T}.
$$

**Proposition 4.6.** *Under the same assumptions in Proposition 3.1, when updating the threshold according to Eq. (6), then for any $\delta \in (0,1)$ and $T \in \mathbb{N}^+$, the following bound holds with probability at least $1 - \delta$:*

$$
\mathrm{EmErr(T)} \leq \frac{2-\epsilon}{1-\epsilon}\sqrt{2\log\left(\frac{4}{\delta}\right)} \cdot \frac{1}{\sqrt{T}} + \left[\left(1 + \max_{1 \leq t \leq T-1}\eta_t \cdot \frac{1+\epsilon}{1-\epsilon}\right)\sum_{t=1}^T\left|\eta_t^{-1} - \eta_{t-1}^{-1}\right|\right] \cdot \frac{1}{T}.
$$

In Proposition 4.5 and 4.6, we establish that for any sequence of learning rates that satisfies the condition $\sum_{t=1}^T\left|\eta_t^{-1} - \eta_{t-1}^{-1}\right|/T \to 0$ as $T \to +\infty$, both expected and empirical coverage errors asymptotically vanish with a convergence rate of $\mathcal{O}(T^{-1/2})$. Therefore, by applying our method, the long-term coverage rate would approach the desired target level of $1 - \alpha$.

Online learning theory (see Theorem 2.13 of [28]) established that ACI achieves a *regret* bound:

$$
\mathrm{Reg}(T) = \sum_{t=1}^T l_{1-\alpha}(\hat{\tau}_t, S_t) - \min_\tau \sum_{t=1}^T l_{1-\alpha}(\tau, S_t) = \mathcal{O}(T^{-\frac{1}{2}}),
$$

with optimally chosen $\eta_t$, which serves as a helpful measure alongside coverage [11]. We provide a regret analysis for our method in Appendix B.1. In addition, we formally establish the convergence of $\hat{\tau}_t$ towards the global minima in Appendix B.2 by extending the standard convergence analysis of stochastic gradient descent (see Theorem 5.3 in [29]).

# 5 Experiments

## 5.1 Experimental setups

**Datasets and setup.** We evaluate the performance of NR-OCP under uniform label noise. The experiments include both constant $\eta = 0.05$ and dynamic learning rates $\eta_t = 1/t^{1/2+\varepsilon}$ with $\varepsilon = 0.1$, following prior work [12]. We use CIFAR-100 [19] and ImageNet [20] datasets with synthetic label noise. On ImageNet, we use four pre-trained classifiers from TorchVision [30] - ResNet18, ResNet50 [31], DenseNet121 [32] and VGG16 [33]. On CIFAR-100, we train these models for 200 epochs using SGD with a momentum of 0.9, a weight decay of 0.0005, and a batch size of 128. We set the initial learning rate as 0.1, and reduce it by a factor of 5 at 60, 120 and 160 epochs.

**Conformal prediction algorithms.** We integrate the robust pinball loss into ACI with constant [9] and dynamic learning rates [12], and strongly adaptive online conformal prediction [11]. We apply LAC [21], APS [34], RAPS [35] and SAPS [36] to generate prediction sets. A detailed description of these non-conformity scores is provided in Appendix B.4. In addition, we use error rates $\alpha \in \{0.1, 0.05\}$ in experiments.

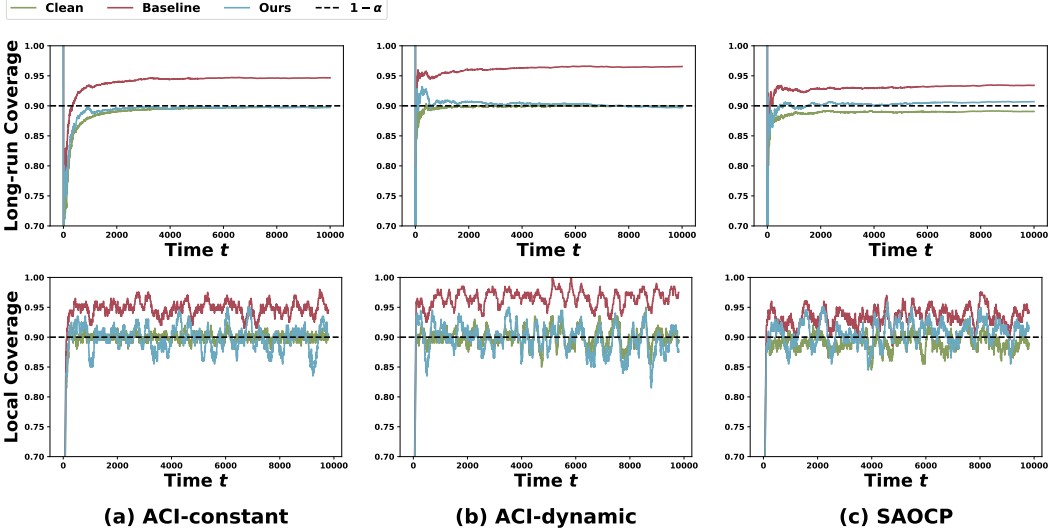

Figure 2: Long-run coverage and local coverage performance of various methods under uniform noisy labels with noise rate $\epsilon = 0.05$. We apply robust pinball loss to ACI with (a) constant [9] and (b) dynamic learning rate [12], and (c) SAOCP [11]. We employ LAC to generate prediction sets with $\alpha = 0.1$, using ResNet18 on CIFAR100. "Baseline" and "Clean" denote the online conformal prediction with standard pinball loss, using noisy and clean labels.

**Metrics.** We employ four evaluation metrics: *long-run coverage (Cov)*, *local coverage (LocalCov)*, *coverage gap (CovGap)* and *prediction set size (Size)*. We use long-run coverage and local coverage to present the dynamics of coverage during the optimization. Formally, for any $T \in \mathbb{N}^+$,

$$\text{Cov}(T) = \frac{1}{T} \sum_{t=1}^{T} \mathbb{1}\left\{Y_t \in \mathcal{C}_t(X_t)\right\}, \quad \text{LocalCov}(T) = \frac{1}{L} \sum_{t=T}^{T+L} \mathbb{1}\left\{Y_t \in \mathcal{C}_t(X_t)\right\},$$

In particular, long-run coverage measures the coverage rate over the first $T$ time steps, while local coverage is over an interval with length $L$. In the experiments, we employ a length of $L = 200$. Besides, the coverage gap and the prediction set size are computed over the full test set, indicating the final performance of the online conformal prediction method. Formally, given a test dataset $\mathcal{I}_{test}$,

$$\text{CovGap} = \left| \frac{1}{|\mathcal{I}_{test}|} \sum_{t \in \mathcal{I}_{test}} \mathbb{1}\left\{Y_t \in \mathcal{C}_t(X_t)\right\} - (1 - \alpha) \right|, \quad \text{Size} = \frac{1}{|\mathcal{I}_{test}|} \sum_{t \in \mathcal{I}_{test}} |\mathcal{C}_t(X_t)|.$$

Small prediction sets are preferred as they can provide precise predictions, thereby enabling accurate human decision-making in real-world scenarios [37].

## 5.2 Main results

**Robust pinball loss enhances the noise robustness of existing online conformal prediction methods.** In figure 2, we apply robust pinball loss to ACI with constant [9] and dynamic learning rate [12], and SAOCP [11]. We evaluate these methods with long-run coverage and local coverage. We use the LAC score and a ResNet18 model on CIFAR100, with error rate $\alpha = 0.1$ and noise rate $\epsilon = 0.05$. The results demonstrate that the proposed robust pinball loss enables these methods to achieve a long-run coverage and local coverage close to the target coverage rate of $1 - \alpha$. In summary, the empirical results highlight the versatility of our robust pinball loss, demonstrating its potential to enhance the noise robustness across diverse algorithms.

**Robust pinball loss is effective across various settings.** In Table 1, we incorporate robust pinball loss into ACI with constant [9] and dynamic learning rate [12], under different noise rate $\epsilon$, error rate $\alpha$ and non-conformity scores. We evaluate the prediction sets with coverage gap and prediction set size. Due to space constraints, we report the average performance across four non-conformity score functions. The performance on each score function is provided in Appendix B.5. The results

Table 1: Average performance of different methods under uniform noisy labels across 4 score functions, using ResNet18. The performance on each score function is provided in Appendix B.5. "Baseline" denotes the ACI with standard pinball loss. We include two learning rate schedules: constant learning rate $\eta = 0.05$ and dynamic learning rates $\eta_t = 1/t^{1/2+\varepsilon}$ where $\varepsilon = 0.1$. "↓" indicates smaller values are better and **Bold** numbers are superior results.

| LR Schedule | Error rate | Method | CIFAR100 | | | | ImageNet | | | |
| --- | --- | --- | --- | --- | --- | --- | --- | --- | --- | --- |
| | | | CovGap(%) ↓ | | Size ↓ | | CovGap(%) ↓ | | Size ↓ | |
| | | | $\alpha=0.1$ | $\alpha=0.05$ | $\alpha=0.1$ | $\alpha=0.05$ | $\alpha=0.1$ | $\alpha=0.05$ | $\alpha=0.1$ | $\alpha=0.05$ |
| Constant | $\epsilon=0.05$ | Baseline | 3.942 | 2.867 | 11.93 | 30.66 | 4.163 | 3.289 | 101.0 | 231.7 |
| | | Ours | **0.386** | **0.183** | **7.786** | **16.54** | **0.084** | **0.272** | **67.37** | **143.3** |
| | $\epsilon=0.1$ | Baseline | 7.139 | 3.753 | 21.81 | 46.93 | 7.509 | 4.068 | 178.4 | 389.7 |
| | | Ours | **0.270** | **0.428** | **8.815** | **17.96** | **0.095** | **0.194** | **81.87** | **157.3** |
| | $\epsilon=0.15$ | Baseline | 8.032 | 4.147 | 36.95 | 59.92 | 8.566 | 4.348 | 318.0 | 513.2 |
| | | Ours | **0.520** | **0.395** | **9.396** | **19.41** | **0.198** | **0.144** | **90.52** | **163.4** |
| Dynamic | $\epsilon=0.05$ | Baseline | 4.106 | 2.780 | 6.527 | 21.97 | 4.498 | 2.584 | 31.15 | 128.1 |
| | | Ours | **0.170** | **0.658** | **2.958** | **10.95** | **0.120** | **0.242** | **7.502** | **47.41** |
| | $\epsilon=0.1$ | Baseline | 7.414 | 3.733 | 18.18 | 37.18 | 7.249 | 3.595 | 99.54 | 214.9 |
| | | Ours | **0.414** | **0.217** | **3.361** | **12.01** | **0.079** | **0.321** | **10.34** | **67.03** |
| | $\epsilon=0.15$ | Baseline | 8.403 | 4.211 | 29.41 | 48.24 | 8.372 | 4.127 | 171.2 | 274.9 |
| | | Ours | **0.214** | **0.195** | **4.065** | **12.98** | **0.183** | **0.256** | **13.10** | **78.64** |

show that robust pinball loss allows ACI to eliminate the coverage gap, achieving precise coverage guarantees while significantly improving the long-run efficiency of prediction sets. For example, on ImageNet with error rate $\alpha = 0.1$, noise rate $\epsilon = 0.15$ and dynamic learning rate, ACI employing the robust pinball loss achieves a negligible coverage gap of 0.183% and a prediction set size of 13.10. We report additional results on various models in Appendix B.6. Overall, the robust pinball loss is effective across different settings, including various label noise, error rate, non-conformity score functions, and model architectures.

In real-world scenarios where the noise rate could be unknown, it can be estimated from historical data [25, 26, 27]. In Appendix C, we leverage an existing approach to estimate the noise rate. Then, we employ this estimated noise rate in our method. We show that in this circumstance, the robust pinball loss can improve the noise robustness of online conformal prediction algorithms.

## 6 Conclusion

In this work, we investigate the robustness of online conformal prediction under uniform label noise with a known noise rate, in both constant and dynamic learning rate schedules. Our theoretical analysis shows that the presence of label noise causes a deviation between the actual and desired mis-coverage rate $\alpha$, with higher noise rates resulting in larger gaps. To address this issue, we propose a novel loss function *robust pinball loss*, which provides an unbiased estimate of clean pinball loss without requiring ground-truth labels. We theoretically establish that this loss is equivalent to the pinball loss under clean labels in expectation. In our theoretical analysis, we show that robust pinball loss enables online conformal prediction to eliminate the coverage gap caused by the label noise, achieving a convergence rate of $\mathcal{O}(T^{-1/2})$ for both empirical and expected coverage errors under uniform label noise. This loss offers a general solution to the uniform label noise, and is complementary to existing online conformal prediction methods. Extensive experiments demonstrate that the proposed loss enhances the noise robustness of various online conformal prediction methods by eliminating the coverage gap caused by the label noise. Notably, our loss function is effective across different label noise, error rate, non-conformity score functions, and model architectures.

**Limitation.** As the first step to explore the label noise issue in online conformal prediction, our analysis and method are limited to the setting of uniform noisy labels with a known noise rate. We believe it will be interesting to develop online conformal prediction algorithms that are robust to various types of label noise with fewer assumptions in the future.

## Acknowledgements

This research is supported by the Shenzhen Fundamental Research Program (Grant No. JCYJ20230807091809020). We gratefully acknowledge the support of the Center for Computational Science and Engineering at the Southern University of Science and Technology for our research.

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

# A  Related work

Conformal prediction [38, 5] is a statistical framework for uncertainty qualification. In the literature, many conformal prediction methods have been proposed across various domains, such as regression [39, 40], image classification [35, 41, 42], outlier detection [43, 44, 45], and large language models [46, 47]. Conformal prediction is also deployed in other real-world applications, such as human-in-the-loop decision-making [37], automated vehicles [48], and scientific computation [49]. In what follows, we introduce the most related works in two settings: online learning and noise robustness.

**Online conformal prediction.** Conventional conformal prediction algorithms provide coverage guarantees under the assumption of data exchangeability. However, in real-world online scenarios, the data distribution may evolve over time, violating the exchangeability assumption [50, 51]. To address this challenge, recent research develops online conformal prediction methods that adaptively construct prediction sets to accommodate distribution shift [9, 10, 11, 12, 13, 14]. Building on online convex optimization techniques [52, 53, 54, 55, 56, 28], these methods employ online gradient descent with pinball loss to provably achieve desired coverage under arbitrary distributional changes [9]. Still, these algorithms typically assume perfect label accuracy, an assumption that rarely holds in practice, given the prevalence of noisy labels in online learning [15, 16, 17]. In this work, we theoretically show that label noise can significantly affect the long-run mis-coverage rate through biased pinball loss gradients, leading to either inflated or deflated coverage guarantee.

**Noise-robust conformal prediction.** The issue of label noise has been a common challenge in machine learning with extensive studies [57, 27, 58, 59, 60, 61, 62, 63, 64, 65]. In the context of conformal prediction, recent works develop noise-robust conformal prediction algorithms for both uniform noise [23] and noise transition matrix [24]. The most relevant work [18] shows that online conformal prediction maintains valid coverage when noisy scores stochastically dominate clean scores. Our analysis extends this work by removing the assumption on noisy and clean scores, offering a more general theoretical framework for understanding the impact of uniform label noise.

# B  Additional results

## B.1  Regret analysis

As [11] demonstrates, regret serves as a helpful performance measure alongside coverage. In particular, it can identify algorithms that achieve valid coverage guarantees through impractical means. For example, prediction sets that alternate between empty and full sets with frequencies $\{\alpha, 1-\alpha\}$ satisfy coverage bounds on any distribution but have linear regret on simple distributions (see detailed proof in Appendix A.2 of [11]). Drawing from standard online learning theory (see Theorem 2.13. of [28]), we analyze the regret bound for our method. Let $\tau^* := \arg\min_{\hat{\tau}} \sum_{t=1}^{T} \tilde{l}_{1-\alpha}(\tau^*, \tilde{S}_t, \{S_{t,y}\}_{y=1}^{K})$, and define the regret as

$$\text{Reg}(T) = \sum_{t=1}^{T} \left( \tilde{l}_{1-\alpha}(\hat{\tau}_t, \tilde{S}_t, \{S_{t,y}\}_{y=1}^{K}) - \tilde{l}_{1-\alpha}(\tau^*, \tilde{S}_t, \{S_{t,y}\}_{y=1}^{K}) \right).$$

This leads to the following regret bound (the proof is provided in Appendix **??**):

**Proposition B.1.** *Consider online conformal prediction under uniform label noise with noise rate $\epsilon \in (0,1)$. Given Assumptions 2.1 and 2.2, when updating the threshold according to Eq. (6), for any $T \in \mathbb{N}^+$, we have:*

$$\text{Reg}(T) \leq \frac{1}{2\eta_T} \left( 1 + \max_{1 \leq t \leq T-1} \eta_t \cdot \frac{1+\epsilon}{1-\epsilon} \right)^2 + \left( \frac{1+\epsilon}{1-\epsilon} \right)^2 \cdot \sum_{t=1}^{T} \frac{\eta_t}{2}.$$

**Example 1: constant learning rate.** With a constant learning rate $\eta_t \equiv \eta$, our method yields the following regret bound:

$$\text{Reg}(T) \leq \frac{1}{\eta} \left( 1 + \eta \cdot \frac{1+\epsilon}{1-\epsilon} \right)^2 + \left( \frac{1+\epsilon}{1-\epsilon} \right)^2 \cdot \frac{T\eta}{2}.$$

This linear regret aligns with the observation in [12]: the constant learning rate schedule, while providing valid coverage on average, lead to significant temporal variability in Coverage($\hat{\tau}_t$) (see Proposition 1 in [12]). The linear regret bound provides additional theoretical justification for this instability.

**Example 2: decaying learning rate.** Suppose the proposed method updates the threshold with a decaying learning rate schedule: $\eta_t = (1 - \epsilon/1 + \epsilon + \eta_0) \cdot \sqrt{t}$, where $\eta_0 \in \mathbb{R}$. The inequality $\sum_{t=1}^{T} 1/\sqrt{t} \leq 2\sqrt{T}$ follows that

$$\text{Reg}(T) \leq 2 \left[ \frac{1+\epsilon}{1-\epsilon} + \eta_0 \cdot \left( \frac{1+\epsilon}{1-\epsilon} \right)^2 \right] \cdot \sqrt{T}$$

This sublinear regret bound $\mathcal{O}(\sqrt{T})$ implies that the decaying learning rate schedule achieves superior convergence compared to the linear regret of constant learning rates.

## B.2 Convergence analysis of $\hat{\tau}_t$

We analyze the convergence of our method toward global minima by combining the convergence analysis of SGD (see Theorem 5.3 in [29]) and self-calibration inequality of pinball loss (see Theorem 2.7. in [66]). Following [66], we first establish assumptions on the distribution of $S$. Let $R := 2S - 1 \in [0, 1]$, with $\gamma$ denoting the $1 - \alpha$ quantile of $R$. This indicates that $\gamma = 2\tau - 1$, where $\tau$ is the $1 - \alpha$ quantile of $S$. We make the following assumption:

**Assumption B.2.** There exists constants $b > 0$, $q \geq 2$, and $\varepsilon_0 > 0$ such that $\mathbb{P}\{R = \hat{\gamma}\} \geq b |\hat{\gamma} - \gamma|^{q-2}$ holds for all $\hat{\tau} \in [\tau - \varepsilon_0, \tau + \varepsilon_0]$.

Furthermore, let $\beta = b/(q - 1)$ and $\delta = \beta(2\varepsilon_0)^{q-1}$. This leads to the following proposition (the proof is provided in Appendix **??**):

**Proposition B.3.** *Consider online conformal prediction under uniform label noise with noise rate $\epsilon \in (0, 1)$. Given Assumptions 2.1, 2.2 and B.2, when updating the threshold according to Eq. (6), for any $T \in \mathbb{N}^+$, we have:*

$$|\bar{\tau} - \tau|^q \leq \frac{q(1-q)}{b} \left( \frac{1}{2\varepsilon_0} \right)^q \cdot \left[ \frac{(\hat{\tau}_1 - \tau^*)^2}{2 \sum_{t=1}^{T} \eta_t} + \frac{\sum_{t=1}^{T} \eta_t^2}{\sum_{t=1}^{T} \eta_t} \cdot \left( \frac{1+\epsilon}{1-\epsilon} \right)^2 \right]$$

*where $\bar{\tau} = \sum_{t=1}^{T} \eta_t \hat{\tau}_t / \sum_{t=1}^{T} \eta_t$.*

## B.3 Why does standard online conformal prediction rarely exhibit under-coverage under label noise?

As established in Proposition 3.1, the label noise introduces a gap of $\frac{\epsilon}{1-\epsilon} \cdot \frac{1}{T} \sum_{t=1}^{T} \left( (1-\alpha) - \frac{1}{K} \mathbb{E}\left[\mathcal{C}_t(X_t)\right] \right)$ between the actual mis-coverage rate and desired mis-coverage rate $\alpha$. This result indicates that when the prediction sets are sufficiently large such that $(1 - \alpha) \leq \frac{1}{K} \mathbb{E}\left[\mathcal{C}_t(X_t)\right]$, a high noise rate $\epsilon$ decreases this upper bound, resulting in under-coverage prediction sets, i.e., $\frac{1}{T} \sum_{t=1}^{T} \mathbb{1}\{Y_t \notin \mathcal{C}_t(X_t)\} \geq \alpha$. However, this scenario only arises when $\mathbb{E}\left[\mathcal{C}_t(X_t)\right] \geq K(1 - \alpha)$. To illustrate, considering CIFAR-100 with $K = 100$ classes and error rate $\alpha = 0.1$, under-coverage would require $\mathbb{E}\left[\mathcal{C}_t(X_t)\right] \geq 90$, a condition that remains improbable even with random predictions. Moreover, in such extreme cases, as $(1 - \alpha) - \frac{1}{K} \mathbb{E}\left[\mathcal{C}_t(X_t)\right]$ approaches 0, the coverage gap becomes negligible, resulting in coverage rates that approximate the desired $1 - \alpha$. We verify this empirically in Figure 3, where we implement online conformal prediction using an untrained, randomly initialized ResNet18 model on CIFAR-10 and CIFAR-100 with noise rates $\epsilon = 0.05, 0.1, 0.15$. The results confirm that even in this extreme scenario, the coverage gap remains negligible, with coverage rates approaching the desired 0.9.

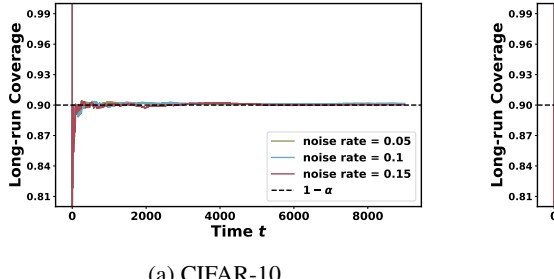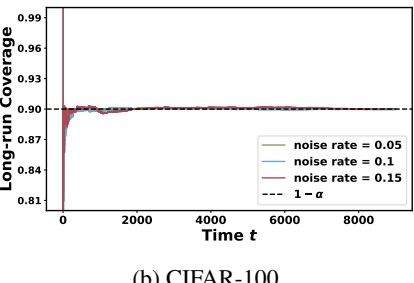

|            (a) CIFAR-10            |            (b) CIFAR-100           |

Figure 3: Performance of standard online conformal prediction under different noise rates, with a ResNet18 model on CIFAR-10 and CIFAR-100 datasets. We use noisy labels to update the threshold with decaying learning rates $\eta_t = 1/t^{1/2+\varepsilon}$ where $\varepsilon = 0.1$.

## B.4 Common non-conformity scores

**Adaptive Prediction Set (APS).** [34] In the APS method, the non-conformity score of a data pair $(\boldsymbol{x}, y)$ is calculated by accumulating the sorted softmax probability, defined as:

$$\mathcal{S}_{APS}(\boldsymbol{x}, y) = \pi_{(1)}(\boldsymbol{x}) + \cdots + u \cdot \pi_{o(y, \pi(\boldsymbol{x}))}(\boldsymbol{x}),$$

where $\pi_{(1)}(\boldsymbol{x}), \pi_{(2)}(\boldsymbol{x}), \cdots, \pi_{(K)}(\boldsymbol{x})$ are the sorted softmax probabilities in descending order, and $o(y, \pi(\boldsymbol{x}))$ denotes the order of $\pi_y(\boldsymbol{x})$, i.e., the softmax probability for the ground-truth label $y$. In addition, the term $u$ is an independent random variable that follows a uniform distribution on $[0, 1]$.

**Regularized Adaptive Prediction Set (RAPS).** [35] The non-conformity score function of RAPS encourages a small set size by adding a penalty, as formally defined below:

$$\mathcal{S}_{RAPS}(\boldsymbol{x}, y) = \pi_{(1)}(\boldsymbol{x}) + \cdots + u \cdot \pi_{o(y, \pi(\boldsymbol{x}))}(\boldsymbol{x}) + \lambda \cdot (o(y, \pi(\boldsymbol{x})) - k_{reg})^+,$$

where $(z)^+ = \max\{0, z\}$, $k_{reg}$ controls the number of penalized classes, and $\lambda$ is the penalty term.

**Sorted Adaptive Prediction Set (SAPS).** [36] Recall that APS calculates the non-conformity score by accumulating the sorted softmax values in descending order. However, the softmax probabilities typically exhibit a long-tailed distribution, allowing for easy inclusion of those tail classes in the prediction sets. To alleviate this issue, SAPS discards all the probability values except for the maximum softmax probability when computing the non-conformity score. Formally, the non-conformity score of SAPS for a data pair $(\boldsymbol{x}, y)$ can be calculated as

$$S_{saps}(\boldsymbol{x}, y, u; \hat{\pi}) := \begin{cases} u \cdot \hat{\pi}_{max}(\boldsymbol{x}), & \text{if } o(y, \hat{\pi}(\boldsymbol{x})) = 1, \\ \hat{\pi}_{max}(\boldsymbol{x}) + (o(y, \hat{\pi}(\boldsymbol{x})) - 2 + u) \cdot \lambda, & \text{else,} \end{cases}$$

where $\lambda$ is a hyperparameter representing the weight of ranking information, $\hat{\pi}_{max}(\boldsymbol{x})$ denotes the maximum softmax probability and $u$ is a uniform random variable.

## B.5 Additional experiments on different non-conformity score functions

We evaluate NR-OCP (Ours) against the standard online conformal prediction (Baseline) that updates the threshold with noisy labels for both constant and dynamic learning rate schedules (see Eq. (3)). We use LAC (Table 2), APS (Table 3), RAPS (Table 4) and SAPS (Table 5) scores to generate prediction sets with error rates $\alpha \in \{0.1, 0.05\}$, and employ noise rates $\epsilon \in \{0.05, 0.1, 0.15\}$. A detailed description of these non-conformity scores is provided in Appendix B.4.

Table 2: Performance of different methods under uniform noisy labels with LAC score, using ResNet18. 'Baseline' denotes the standard online conformal prediction methods. We include two learning rate schedules: constant learning rate $\eta = 0.05$ and dynamic learning rates $\eta_t = 1/t^{1/2+\varepsilon}$ where $\varepsilon = 0.1$. "↓" indicates smaller values are better and **Bold** numbers are superior results.

| LR Schedule | Error rate | Method | CIFAR100 | | | | ImageNet | | | |
| | | | CovGap(%) ↓ | | Size ↓ | | CovGap(%) ↓ | | Size ↓ | |
| | | | $\alpha = 0.1$ | $\alpha = 0.05$ | $\alpha = 0.1$ | $\alpha = 0.05$ | $\alpha = 0.1$ | $\alpha = 0.05$ | $\alpha = 0.1$ | $\alpha = 0.05$ |
|---|---|---|---|---|---|---|---|---|---|---|
| Constant | $\epsilon = 0.05$ | Baseline | 2.744 | 1.900 | 31.61 | 56.84 | 2.747 | 1.601 | 364.5 | 599.1 |
| | | Ours | **0.289** | **0.122** | **22.78** | **47.31** | **0.018** | **0.138** | **254.7** | **530.6** |
| | $\epsilon = 0.1$ | Baseline | 4.911 | 2.967 | 43.03 | 67.58 | 4.578 | 2.656 | 488.1 | 707.6 |
| | | Ours | **0.056** | **0.189** | **26.57** | **54.18** | **0.027** | **0.156** | **312.4** | **584.9** |
| | $\epsilon = 0.15$ | Baseline | 6.056 | 3.533 | 54.11 | 75.06 | 5.791 | 3.200 | 571.5 | 759.3 |
| | | Ours | **0.378** | **0.344** | **28.39** | **57.53** | **0.251** | **0.076** | **346.7** | **603.4** |
| Dynamic | $\epsilon = 0.05$ | Baseline | 3.978 | 2.833 | 11.54 | 41.75 | 4.333 | 2.889 | 87.27 | 391.8 |
| | | Ours | **0.089** | **0.067** | **4.290** | **26.56** | **0.031** | **0.020** | **16.46** | **150.5** |
| | $\epsilon = 0.1$ | Baseline | 6.756 | 3.844 | 30.07 | 60.17 | 6.960 | 3.933 | 284.6 | 600.2 |
| | | Ours | **0.233** | **0.222** | **5.394** | **30.38** | **0.091** | **0.313** | **27.76** | **227.9** |
| | $\epsilon = 0.15$ | Baseline | 7.844 | 4.289 | 42.69 | 70.67 | 8.098 | 4.276 | 447.9 | 704.4 |
| | | Ours | **0.456** | **0.067** | **8.359** | **33.60** | **0.131** | **0.158** | **38.89** | **272.5** |

Table 3: Performance of different methods under uniform noisy labels with APS score, using ResNet18. 'Baseline' denotes the standard online conformal prediction methods. We include two learning rate schedules: constant learning rate $\eta = 0.05$ and dynamic learning rates $\eta_t = 1/t^{1/2+\varepsilon}$ where $\varepsilon = 0.1$. "↓" indicates smaller values are better and **Bold** numbers are superior results.

| LR Schedule | Error rate | Method | CIFAR100 | | | | ImageNet | | | |
| | | | CovGap(%) ↓ | | Size ↓ | | CovGap(%) ↓ | | Size ↓ | |
| | | | $\alpha = 0.1$ | $\alpha = 0.05$ | $\alpha = 0.1$ | $\alpha = 0.05$ | $\alpha = 0.1$ | $\alpha = 0.05$ | $\alpha = 0.1$ | $\alpha = 0.05$ |
|---|---|---|---|---|---|---|---|---|---|---|
| Constant | $\epsilon = 0.05$ | Baseline | 4.556 | 2.933 | 5.862 | 21.17 | 4.728 | 3.508 | 13.35 | 79.88 |
| | | Ours | **2.667** | **0.311** | **2.788** | **6.763** | **0.144** | **0.211** | **4.879** | **14.09** |
| | $\epsilon = 0.1$ | Baseline | 7.511 | 3.822 | 15.71 | 37.19 | 8.402 | 4.388 | 67.25 | 224.2 |
| | | Ours | **1.333** | **0.367** | **2.861** | **6.302** | **0.119** | **0.246** | **4.797** | **14.21** |
| | $\epsilon = 0.15$ | Baseline | 8.533 | 4.188 | 29.76 | 51.63 | 9.380 | 4.644 | 192.4 | 355.3 |
| | | Ours | **0.500** | **0.233** | **2.915** | **6.813** | **0.288** | **0.139** | **4.787** | **14.71** |
| Dynamic | $\epsilon = 0.05$ | Baseline | 4.000 | 2.433 | 4.693 | 13.03 | 4.157 | 2.122 | 11.31 | 31.38 |
| | | Ours | **0.278** | **0.466** | **2.427** | **5.475** | **0.108** | **0.255** | **4.431** | **13.36** |
| | $\epsilon = 0.1$ | Baseline | 7.211 | 3.366 | 11.14 | 21.91 | 6.791 | 3.137 | 27.78 | 54.14 |
| | | Ours | **0.400** | **0.456** | **2.716** | **5.515** | **0.028** | **0.277** | **4.513** | **13.30** |
| | $\epsilon = 0.15$ | Baseline | 8.288 | 3.911 | 19.62 | 31.19 | 7.928 | 3.675 | 47.88 | 80.33 |
| | | Ours | **0.188** | **0.211** | **2.479** | **5.790** | **0.264** | **0.088** | **4.321** | **14.03** |

Table 4: Performance of different methods under uniform noisy labels with RAPS score, using ResNet18. 'Baseline' denotes the standard online conformal prediction methods. We include two learning rate schedules: constant learning rate $\eta = 0.05$ and dynamic learning rates $\eta_t = 1/t^{1/2+\varepsilon}$ where $\varepsilon = 0.1$. "↓" indicates smaller values are better and **Bold** numbers are superior results.

| LR Schedule | Error rate | Method | CIFAR100 | | | | ImageNet | | | |
| | | | CovGap(%) ↓ | | Size ↓ | | CovGap(%) ↓ | | Size ↓ | |
| | | | $\alpha = 0.1$ | $\alpha = 0.05$ | $\alpha = 0.1$ | $\alpha = 0.05$ | $\alpha = 0.1$ | $\alpha = 0.05$ | $\alpha = 0.1$ | $\alpha = 0.05$ |
|---|---|---|---|---|---|---|---|---|---|---|
| Constant | $\epsilon = 0.05$ | Baseline | 3.978 | 3.533 | 5.275 | 26.71 | 4.716 | 4.509 | 13.73 | 164.7 |
| | | Ours | **0.533** | **0.089** | **2.971** | **6.265** | **0.051** | **0.673** | **5.217** | **14.21** |
| | $\epsilon = 0.1$ | Baseline | 7.656 | 4.278 | 15.25 | 50.25 | 8.689 | 4.84 | 85.19 | 395.5 |
| | | Ours | **0.356** | **0.200** | **3.126** | **6.370** | **0.184** | **0.271** | **5.264** | **14.76** |
| | $\epsilon = 0.15$ | Baseline | 8.896 | 4.467 | 36.03 | 62.27 | 9.733 | 4.933 | 305.6 | 576.3 |
| | | Ours | **0.489** | **0.560** | **3.500** | **7.693** | **0.224** | **0.227** | **5.616** | **19.81** |
| Dynamic | $\epsilon = 0.05$ | Baseline | 4.322 | 3.254 | 4.883 | 19.42 | 4.642 | 3.457 | 12.57 | 60.94 |
| | | Ours | **0.233** | **0.211** | **2.586** | **5.344** | **0.067** | **0.078** | **4.305** | **13.73** |
| | $\epsilon = 0.1$ | Baseline | 8.378 | 4.400 | 19.68 | 44.43 | 8.084 | 4.311 | 54.42 | 153.7 |
| | | Ours | **0.756** | **0.122** | **2.891** | **5.942** | **0.009** | **0.140** | **4.381** | **14.67** |
| | $\epsilon = 0.15$ | Baseline | 9.022 | 4.600 | 33.87 | 57.87 | 9.256 | 4.978 | 133.3 | 239.8 |
| | | Ours | **0.011** | **0.440** | **2.821** | **5.290** | **0.231** | **0.264** | **4.578** | **15.59** |

Table 5: Performance of different methods under uniform noisy labels with SAPS score, using ResNet18. 'Baseline' denotes the standard online conformal prediction methods. We include two learning rate schedules: constant learning rate $\eta = 0.05$ and dynamic learning rates $\eta_t = 1/t^{1/2+\varepsilon}$ where $\varepsilon = 0.1$. "↓" indicates smaller values are better and **Bold** numbers are superior results.

| LR Schedule | Error rate | Method | CIFAR100 | | | | ImageNet | | | |
| | | | CovGap(%) ↓ | | Size ↓ | | CovGap(%) ↓ | | Size ↓ | |
| | | | $\alpha = 0.1$ | $\alpha = 0.05$ | $\alpha = 0.1$ | $\alpha = 0.05$ | $\alpha = 0.1$ | $\alpha = 0.05$ | $\alpha = 0.1$ | $\alpha = 0.05$ |
| Constant | $\epsilon = 0.05$ | Baseline | 4.489 | 3.100 | 4.982 | 17.92 | 4.462 | 3.537 | 12.53 | 83.04 |
| | | Ours | **0.455** | **0.211** | **2.604** | **5.810** | **0.124** | **0.067** | **4.702** | **14.32** |
| | $\epsilon = 0.1$ | Baseline | 8.478 | 3.944 | 13.25 | 32.71 | 8.368 | 4.389 | 73.14 | 231.5 |
| | | Ours | **0.533** | **0.955** | **2.701** | **4.987** | **0.051** | **0.102** | **5.013** | **15.37** |
| | $\epsilon = 0.15$ | Baseline | 8.644 | 4.400 | 27.89 | 50.73 | 9.360 | 4.613 | 202.3 | 361.7 |
| | | Ours | **0.711** | **0.444** | **2.778** | **5.609** | **0.029** | **0.135** | **4.971** | **15.68** |
| Dynamic | $\epsilon = 0.05$ | Baseline | 4.122 | 2.600 | 4.992 | 13.68 | 4.859 | 1.866 | 13.46 | 28.28 |
| | | Ours | **0.078** | **1.889** | **2.528** | **6.417** | **0.275** | **0.615** | **4.813** | **12.06** |
| | $\epsilon = 0.1$ | Baseline | 7.311 | 3.322 | 11.84 | 22.19 | 7.162 | 3.000 | 31.37 | 51.64 |
| | | Ours | **0.267** | **0.067** | **2.444** | **6.218** | **0.186** | **0.553** | **4.715** | **12.23** |
| | $\epsilon = 0.15$ | Baseline | 8.456 | 4.044 | 21.44 | 33.21 | 8.204 | 3.577 | 55.78 | 75.26 |
| | | Ours | **0.200** | **0.063** | **2.602** | **7.230** | **0.104** | **0.515** | **4.594** | **12.43** |

## B.6   Additional experiments on different model architectures

We evaluate NR-OCP (Ours) against the standard online conformal prediction (Baseline) that updates the threshold with noisy labels for both constant and dynamic learning rate schedules (see Eq. (3) and Eq. (**??**)), employing ResNet50 (Table 6), DenseNet121 (Table 7), and VGG16 (Table 8). We use LAC scores to generate prediction sets with error rates $\alpha \in \{0.1, 0.05\}$, and employ noise rates $\epsilon \in \{0.05, 0.1, 0.15\}$.

Table 6: Performance of different methods under uniform noisy labels with LAC score, using ResNet50. 'Baseline' denotes the standard online conformal prediction methods. We include two learning rate schedules: constant learning rate $\eta = 0.05$ and dynamic learning rates $\eta_t = 1/t^{1/2+\varepsilon}$ where $\varepsilon = 0.1$. "↓" indicates smaller values are better and **Bold** numbers are superior results.

| LR Schedule | Error rate | Method | CIFAR100 | | | | ImageNet | | | |
| | | | CovGap(%) ↓ | | Size ↓ | | CovGap(%) ↓ | | Size ↓ | |
| | | | $\alpha = 0.1$ | $\alpha = 0.05$ | $\alpha = 0.1$ | $\alpha = 0.05$ | $\alpha = 0.1$ | $\alpha = 0.05$ | $\alpha = 0.1$ | $\alpha = 0.05$ |
| Constant | $\epsilon = 0.05$ | Baseline | 3.244 | 1.783 | 28.58 | 56.08 | 3.226 | 1.982 | 294.4 | 575.7 |
| | | Ours | **0.033** | **0.089** | **16.26** | **46.04** | **0.073** | **0.035** | **188.5** | **466.8** |
| | $\epsilon = 0.1$ | Baseline | 5.267 | 2.879 | 42.09 | 67.25 | 5.326 | 2.935 | 425.9 | 674.2 |
| | | Ours | **0.125** | **0.122** | **17.88** | **49.06** | **0.231** | **0.067** | **241.1** | **506.4** |
| | $\epsilon = 0.15$ | Baseline | 6.369 | 3.445 | 53.34 | 74.03 | 6.584 | 3.453 | 531.4 | 758.5 |
| | | Ours | **0.223** | **0.216** | **21.62** | **50.14** | **0.202** | **0.056** | **263.1** | **549.4** |
| Dynamic | $\epsilon = 0.05$ | Baseline | 4.166 | 2.724 | 10.91 | 41.95 | 4.397 | 3.104 | 44.86 | 361.8 |
| | | Ours | **0.115** | **0.189** | **3.120** | **18.35** | **0.082** | **0.048** | **8.377** | **96.68** |
| | $\epsilon = 0.1$ | Baseline | 6.984 | 3.921 | 30.09 | 61.96 | 7.279 | 4.073 | 232.8 | 582.4 |
| | | Ours | **0.205** | **0.310** | **4.143** | **27.87** | **0.113** | **0.178** | **10.83** | **170.2** |
| | $\epsilon = 0.15$ | Baseline | 8.093 | 4.223 | 43.90 | 70.67 | 8.406 | 4.453 | 417.9 | 705.9 |
| | | Ours | **0.244** | **0.043** | **4.368** | **27.81** | **0.133** | **0.059** | **21.98** | **178.0** |

Table 7: Performance of different methods under uniform noisy labels with LAC score, using DenseNet121. 'Baseline' denotes the standard online conformal prediction methods. We include two learning rate schedules: constant learning rate $\eta = 0.05$ and dynamic learning rates $\eta_t = 1/t^{1/2+\varepsilon}$ where $\varepsilon = 0.1$. "↓" indicates smaller values are better and **Bold** numbers are superior results.

| LR Schedule | Error rate | Method | CIFAR100 | | | | ImageNet | | | |
| | | | CovGap(%) ↓ | | Size ↓ | | CovGap(%) ↓ | | Size ↓ | |
| | | | $\alpha=0.1$ | $\alpha=0.05$ | $\alpha=0.1$ | $\alpha=0.05$ | $\alpha=0.1$ | $\alpha=0.05$ | $\alpha=0.1$ | $\alpha=0.05$ |
|---|---|---|---|---|---|---|---|---|---|---|
| Constant | $\epsilon=0.05$ | Baseline | 3.978 | 3.533 | 5.275 | 26.71 | 4.716 | 4.509 | 13.73 | 164.7 |
| | | Ours | **0.533** | **0.089** | **2.971** | **6.265** | **0.051** | **0.673** | **5.217** | **14.21** |
| | $\epsilon=0.1$ | Baseline | 7.656 | 4.278 | 15.25 | 50.25 | 8.689 | 4.84 | 85.19 | 395.5 |
| | | Ours | **0.356** | **0.200** | **3.126** | **6.370** | **0.184** | **0.271** | **5.264** | **14.76** |
| | $\epsilon=0.15$ | Baseline | 8.896 | 4.467 | 36.03 | 62.27 | 9.733 | 4.933 | 305.6 | 576.3 |
| | | Ours | **0.489** | **0.560** | **3.500** | **7.693** | **0.224** | **0.227** | **5.616** | **19.81** |
| Dynamic | $\epsilon=0.05$ | Baseline | 4.322 | 3.254 | 4.883 | 19.42 | 4.642 | 3.457 | 12.57 | 60.94 |
| | | Ours | **0.233** | **0.211** | **2.586** | **5.344** | **0.067** | **0.078** | **4.305** | **13.73** |
| | $\epsilon=0.1$ | Baseline | 8.378 | 4.400 | 19.68 | 44.43 | 8.084 | 4.311 | 54.42 | 153.7 |
| | | Ours | **0.756** | **0.122** | **2.891** | **5.942** | **0.009** | **0.140** | **4.381** | **14.67** |
| | $\epsilon=0.15$ | Baseline | 9.022 | 4.600 | 33.87 | 57.87 | 9.256 | 4.978 | 133.3 | 239.8 |
| | | Ours | **0.011** | **0.440** | **2.821** | **5.290** | **0.231** | **0.264** | **4.578** | **15.59** |

Table 8: Performance of different methods under uniform noisy labels with LAC score, using VGG16. 'Baseline' denotes the standard online conformal prediction methods. We include two learning rate schedules: constant learning rate $\eta = 0.05$ and dynamic learning rates $\eta_t = 1/t^{1/2+\varepsilon}$ where $\varepsilon = 0.1$. "↓" indicates smaller values are better and **Bold** numbers are superior results.

| LR Schedule | Error rate | Method | CIFAR100 | | | | ImageNet | | | |
| | | | CovGap(%) ↓ | | Size ↓ | | CovGap(%) ↓ | | Size ↓ | |
| | | | $\alpha=0.1$ | $\alpha=0.05$ | $\alpha=0.1$ | $\alpha=0.05$ | $\alpha=0.1$ | $\alpha=0.05$ | $\alpha=0.1$ | $\alpha=0.05$ |
|---|---|---|---|---|---|---|---|---|---|---|
| Constant | $\epsilon=0.05$ | Baseline | 1.911 | 1.033 | 51.13 | 73.03 | 3.164 | 1.995 | 297.8 | 565.3 |
| | | Ours | **0.078** | **0.144** | **43.50** | **67.97** | **0.002** | **0.031** | **206.6** | **488.1** |
| | $\epsilon=0.1$ | Baseline | 3.474 | 1.700 | 58.08 | 76.44 | 4.878 | 2.975 | 439.8 | 669.4 |
| | | Ours | **0.224** | **0.267** | **45.48** | **68.28** | **0.217** | **0.119** | **229.4** | **511.2** |
| | $\epsilon=0.15$ | Baseline | 4.489 | 2.411 | 64.16 | 80.16 | 6.173 | 3.626 | 544.2 | 746.8 |
| | | Ours | **0.300** | **0.189** | **44.43** | **71.81** | **0.168** | **0.004** | **281.9** | **551.3** |
| Dynamic | $\epsilon=0.05$ | Baseline | 2.678 | 1.388 | 40.96 | 66.81 | 4.608 | 3.044 | 51.64 | 374.1 |
| | | Ours | **0.244** | **0.004** | **28.92** | **59.26** | **0.062** | **0.013** | **5.316** | **112.7** |
| | $\epsilon=0.1$ | Baseline | 4.189 | 2.567 | 50.40 | 74.38 | 7.142 | 4.067 | 244.4 | 580.6 |
| | | Ours | **0.289** | **0.133** | **30.81** | **61.08** | **0.195** | **0.071** | **10.65** | **152.1** |
| | $\epsilon=0.15$ | Baseline | 5.733 | 3.044 | 60.46 | 78.65 | 8.413 | 4.451 | 432.2 | 708.9 |
| | | Ours | **0.440** | **0.144** | **35.67** | **63.38** | **0.119** | **0.282** | **18.39** | **224.6** |

## C   The impact of misestimated noise rate on NR-OCP

In this section, we conduct an additional experiment where the noise rate is unknown and needs to be estimated. In particular, we apply an existing algorithm [61] to estimate the noise rate without access to clean data during the training of the pre-trained model. Then, we employ this estimated noise rate in our method for online conformal prediction. The experiments are conducted on CIFAR-10 dataset, using ResNet18 model with noise rates $\epsilon \in \{0.1, 0.2, 0.3\}$. We employ LAC score to generate prediction sets and use a constant learning rate $\eta = 0.05$. The results demonstrate that our method consistently achieves much lower coverage gaps and smaller prediction sets compared to the baseline. This highlights the robustness and applicability of our method even when the noise rate is not precisely estimated.

Table 9

| Noise rate | Estimated Noise rate | Method | CovGap (%) | Size |
|---|---|---|---|---|
| $\epsilon = 0.1$ | $\hat{\epsilon} = 0.09$ | Baseline | 6.99 | 2.47 |
|  |  | Ours | 0.82 | 0.94 |
| $\epsilon = 0.2$ | $\hat{\epsilon} = 0.23$ | Baseline | 8.72 | 5.41 |
|  |  | Ours | 1.67 | 0.98 |
| $\epsilon = 0.3$ | $\hat{\epsilon} = 0.26$ | Baseline | 9.04 | 6.72 |
|  |  | Ours | 2.32 | 1.15 |

# D  Additional information on Strongly adaptive online conformal prediction

## D.1  Strongly adaptive online conformal prediction

The SAOCP algorithm leverages techniques for minimizing the strongly adaptive regret [67] to perform online conformal prediction. In particular, SAOCP is a meta-algorithm that manages multiple experts, where each expert is itself an arbitrary online learning algorithm taking charge of its own active interval that has a finite *lifetime*. At each $t \in [T]$, SAOCP instantiates a new expert $\mathcal{A}_t$ with active interval $[t, t + L(t) - 1]$, where $L(t)$ is the lifetime:

$$L(t) = g \cdot \max_{n \in \mathbb{Z}_{\geq 1}} \{2^n : t \equiv 0 \pmod{2^n}\}, \tag{7}$$

where $g \in \mathbb{Z}_{\geq 1}$ is a multiplier for the lifetime of each expert. It can be shown that at most $g \log_2 t$ experts are active at any time $t$ under the choice of $L(t)$ in (7), resulting in a total runtime of $O(T \log T)$ for SAOCP when $g = \Theta(1)$. At each time $t$, the threshold $\hat{\tau}_t$ is obtained by aggregating the predictions of the active experts.

$$\hat{\tau}_t = \sum_{i \in \text{Active}(t)} p_{i,t} \hat{\tau}_t^i$$

where the weight $\{p_{i,t}\}_{i \in [t]}$ is computed by the coin betting scheme [67, 68].

**Choice of expert.** SAOCP employs Scale-Free OGD (dubbed SF-OGD) [68] as its expert. In particular, SF-OGD is a variant of OGD that decays its effective learning rate based on cumulative past gradient norms.

## D.2  Noise-robust strongly adaptive online conformal prediction

In the following, we present a pseudo-algorithm of NR-SAOCP. The essence of this method is to update the threshold with our robust pinball loss. In Algorithm 1, we demonstrate how NR-SAOCP manages multiple experts to update the threshold. In Algorithm 2, we present how each expert updates the corresponding threshold.

---

**Algorithm 1** Noise-Robust Strongly Adaptive Online Conformal Prediction (NR-SAOCP)

---

**Require:** Target coverage $1 - \alpha \in (0, 1)$, initial threshold $\hat{\tau}_0$.

1: **for** $t = 1, \ldots, T$ **do**
2:   Initialize new expert $\mathcal{A}_t = $ NR-SF-OGD$(\alpha \leftarrow \alpha; \eta \leftarrow 1/\sqrt{3}; \hat{\tau}_1 \leftarrow \hat{\tau}_{t-1})$ (Algorithm 2), and set weight $w_{t,t} = 0$
3:   Compute active set Active$(t) = \{i \in [T] : t - L(i) \leq i \leq t\}$, where $L(i)$ is defined in Eq. (7)

4:   Compute prior probability $\pi_i \propto i^{-2} (1 + \lfloor w_{t,i} \rfloor_+) \mathbb{1} \{i \in \text{Active}(t)\}$, compute un-normalized probability $\hat{p}_i = \pi_i [w_{t,i}]_+$ for all $i \in [t]$, normalize $p = \hat{p}/||\hat{p}||_1$ if $||\hat{p}||_1 > 0$, else $p = \pi$
5:   Update the threshold $\hat{\tau}_t = \sum_{i \in \text{Active}(t)} p_i \hat{\tau}_t^i$
6:   Observe input $X_t$ and construct prediction set $\hat{C}_t(X_t)$ as in Eq. (1)
7:   Observe true label $Y_t \in \mathcal{Y}$, compute non-conformity score $S_t = \mathcal{S}(X_t, Y_t)$ and $S_{t,y} = \mathcal{S}(X_t, y)$ for all $y \in \mathcal{Y}$
8:   **for** $i \in \text{Active}(t)$ **do**
9:     Update expert $A_i$ with robust pinball loss and obtain next predicted radius $\hat{\tau}_{t+1}^i$
10:    Compute

$$g_{i,t} = \begin{cases} \tilde{l}(\hat{\tau}_t, S_t, \{S_{t,y}\}_{y=1}^K) - \tilde{l}(\hat{\tau}_t^i, S_t, \{S_{t,y}\}_{y=1}^K) & \text{if } w_{t,i} > 0 \\ [\tilde{l}(\hat{\tau}_t, S_t, \{S_{t,y}\}_{y=1}^K) - \tilde{l}(\hat{\tau}_t^i, S_t, \{S_{t,y}\}_{y=1}^K)]_+ & \text{if } w_{t,i} \leq 0 \end{cases}$$

     where $\tilde{l}$ is the robust pinball loss defined in Eq. (4).
11:    Update expert weight $w_{i,t+1} = \frac{1}{t-i+1} \left( \sum_{j=i}^t g_{i,j} \right) \left( 1 + \sum_{j=i}^t w_{i,j} g_{i,j} \right)$
12:   **end for**
13: **end for**

---

---

**Algorithm 2** Noise-Robust Scale-Free Online Gradient Descent (NR-SF-OGD)

---

**Require:** $\alpha \in (0, 1)$, learning rate $\eta > 0$, initial threshold $\hat{\tau}_1 \in \mathbb{R}$

1: **for** $t \geq 1$ **do**
2:   Observe input $X_t$ and construct prediction set $\hat{C}_t(X_t)$ as in Eq. (1)
3:   Observe true label $Y_t \in \mathcal{Y}$, compute non-conformity score $S_t = \mathcal{S}(X_t, Y_t)$ and $S_{t,y} = \mathcal{S}(X_t, y)$ for all $y \in \mathcal{Y}$
4:   Update the threshold:

$$\hat{\tau}_{t+1} = \hat{\tau}_t - \eta \frac{\nabla \tilde{l}_{\hat{\tau}_t}(\hat{\tau}_t, S_t, \{S_{t,y}\}_{y=1}^K)}{\sqrt{\sum_{i=1}^t \|\nabla_{\hat{\tau}_i} \tilde{l}(\hat{\tau}_i, S_i, \{S_{i,y}\}_{y=1}^K)\|_2^2}}$$

     where $\tilde{l}$ is the robust pinball loss defined in Eq. (4).
5: **end for**

---

# E  Omitted proofs

## E.1  Proof for Proposition 3.1

**Lemma E.1.** *Given Assumptions 2.1 and 2.2, when updating the threshold according to Eq. (3), for any $T \in \mathbb{N}^+$, we have*

$$-\alpha\eta \leq \hat{\tau}_t \leq 1 + (1-\alpha)\eta. \tag{8}$$

*Proof.* We prove this by induction. First, we know $\hat{\tau}_1 \in [0, 1]$ by assumption, which indicates that Eq. (8) is satisfied at $t = 1$. Then, we assume that Eq. (8) holds for $t = T$, and we will show that $\hat{\tau}_{T+1}$ lies in this range. Consider three cases:

**Case 1.** If $\hat{\tau}_T \in [0, 1]$, we have

$$\hat{\tau}_{T+1} = \hat{\tau}_T - \eta \cdot \nabla_{\hat{\tau}_t} l_{1-\alpha}(\hat{\tau}_t, \tilde{S}_t) \overset{(a)}{\in} [-\alpha\eta, 1 + (1-\alpha)\eta]$$

where (a) is because $\nabla_{\hat{\tau}_t} l_{1-\alpha}(\hat{\tau}_t, \tilde{S}_t) \in [\alpha - 1, \alpha]$.

**Case 2.** Consider the case where $\hat{\tau}_T \in [1, 1 + (1-\alpha)\eta]$. The assumption that $\tilde{S} \in [0, 1]$ implies $\mathbb{1}\left\{\tilde{S} > \hat{\tau}_t\right\} = 0$. Thus, we have

$$\nabla_{\hat{\tau}_t} l_{1-\alpha}(\hat{\tau}_t, \tilde{S}_t) = -\mathbb{1}\left\{\tilde{S} > \hat{\tau}_t\right\} + \alpha = \alpha$$

which follows that

$$\hat{\tau}_{T+1} = \hat{\tau}_T - \eta \cdot \nabla_{\hat{\tau}_t} l_{1-\alpha}(\hat{\tau}_t, \tilde{S}_t) = \hat{\tau}_T - \eta\alpha \in [1 - \eta\alpha, 1 + (1-\alpha)\eta] \subset [-\alpha\eta, 1 + (1-\alpha)\eta]$$

**Case 3.** Consider the case where $\hat{\tau}_T \in [-\alpha\eta, 0]$. The assumption that $\tilde{S} \in [0, 1]$ implies $\mathbb{1}\left\{\tilde{S} > \hat{\tau}_t\right\} = 1$. Thus, we have

$$\nabla_{\hat{\tau}_t} l_{1-\alpha}(\hat{\tau}_t, \tilde{S}_t) = -\mathbb{1}\left\{\tilde{S} > \hat{\tau}_t\right\} + \alpha = -1 + \alpha$$

which follows that

$$\hat{\tau}_{T+1} = \hat{\tau}_T - \eta \cdot \nabla_{\hat{\tau}_t} l_{1-\alpha}(\hat{\tau}_t, \tilde{S}_t) = \hat{\tau}_T - \eta(-1 + \alpha) \in [-\alpha\eta, (1-\alpha)\eta] \subset [-\alpha\eta, 1 + (1-\alpha)\eta]$$

Combining three cases, we can conclude that

$$-\alpha\eta \le \hat{\tau}_t \le 1 + (1-\alpha)\eta$$

$\square$

**Proposition E.2** (Restatement of Proposition 3.1). *Consider online conformal prediction under uniform label noise with noise rate $\epsilon \in (0, 1)$. Given Assumptions 2.1 and 2.2, when updating the threshold according to Eq. (3), for any $\delta \in (0, 1)$ and $T \in \mathbb{N}^+$, the following bound holds with probability at least $1 - \delta$:*

$$\alpha - \frac{1}{T}\sum_{t=1}^{T}\mathbb{1}\{Y_t \notin \mathcal{C}_t(X_t)\} \le \frac{2 - \epsilon}{1 - \epsilon}\sqrt{2\log\left(\frac{4}{\delta}\right)} \cdot \frac{1}{\sqrt{T}} + \frac{1}{1-\epsilon}\left(\frac{1}{\eta} + 1 - \alpha\right) \cdot \frac{1}{T}$$

$$+ \frac{\epsilon}{1-\epsilon} \cdot \frac{1}{T}\sum_{t=1}^{T}\left((1-\alpha) - \frac{1}{K}\mathbb{E}\left[\mathcal{C}_t(X_t)\right]\right).$$

*Proof.* Consider the gradient of clean pinball loss:

$$\sum_{t=1}^{T}\nabla_{\tau_t} l_{1-\alpha}(\tau_t, S_t) = \sum_{t=1}^{T}\nabla_{\tau_t} l_{1-\alpha}(\tau_t, S_t) - \sum_{t=1}^{T}\mathbb{E}_S\left[\nabla_{\tau_t} l_{1-\alpha}(\tau_t, S_t)\right] + \sum_{t=1}^{T}\mathbb{E}_S\left[\nabla_{\tau_t} l_{1-\alpha}(\tau_t, S_t)\right]$$

$$\le \underbrace{\left|\sum_{t=1}^{T}\nabla_{\tau_t} l_{1-\alpha}(\tau_t, S_t) - \sum_{t=1}^{T}\mathbb{E}_S\left[\nabla_{\tau_t} l_{1-\alpha}(\tau_t, S_t)\right]\right|}_{(a)} + \underbrace{\sum_{t=1}^{T}\mathbb{E}_S\left[\nabla_{\tau_t} l_{1-\alpha}(\tau_t, S_t)\right]}_{(b)}$$

Part (a): Lemma F.3 gives us that

$$\mathbb{P}\left\{\left|\sum_{t=1}^{T}\mathbb{E}_{S_t}\left[\nabla_{\hat{\tau}_t} l_{1-\alpha}(\hat{\tau}_t, S_t)\right] - \sum_{t=1}^{T}\nabla_{\hat{\tau}_t} l_{1-\alpha}(\hat{\tau}_t, S_t)\right| \le \sqrt{2T\log\left(\frac{4}{\delta}\right)}\right\} \ge 1 - \frac{\delta}{2}.$$

Part (b): Recall that the expected gradient of the clean pinball loss satisfies

$$\mathbb{E}_S\left[\nabla_{\hat{\tau}_t} l_{1-\alpha}(\hat{\tau}_t, S_t)\right] = \mathbb{P}\{S_t \le \hat{\tau}_t\} - (1-\alpha)$$

$$\overset{(a)}{=} \left(\frac{1}{1-\epsilon}\mathbb{P}\left\{\tilde{S}_t \le \hat{\tau}_t\right\} - \frac{\epsilon}{K(1-\epsilon)}\sum_{y=1}^{K}\mathbb{P}\left\{S_{t,y} \le \hat{\tau}_t\right\}\right) - (1-\alpha)$$

$$= \frac{1}{1-\epsilon}\left(\mathbb{P}\left\{\tilde{S}_t \le \hat{\tau}_t\right\} - (1-\alpha)\right) - \frac{\epsilon}{1-\epsilon}\left(\frac{1}{K}\sum_{y=1}^{K}\mathbb{P}\left\{S_{t,y} \le \hat{\tau}_t\right\} - (1-\alpha)\right)$$

$$= \frac{1}{1-\epsilon}\mathbb{E}_{\tilde{S}_t}\left[\nabla_{\hat{\tau}_t} l_{1-\alpha}(\hat{\tau}_t, \tilde{S}_t)\right] - \frac{\epsilon}{1-\epsilon}\left(\frac{1}{K}\mathbb{E}\left[\mathcal{C}_t(X_t)\right] - (1-\alpha)\right)$$

where (a) comes from Lemma F.1. In addition, Lemma F.3 implies that

$$\mathbb{P}\left\{\left|\sum_{t=1}^{T}\mathbb{E}_{\tilde{S}_t}\left[\nabla_{\hat{\tau}_t}l_{1-\alpha}(\hat{\tau}_t,\tilde{S}_t)\right]-\nabla_{\hat{\tau}_t}l_{1-\alpha}(\hat{\tau}_t,\tilde{S}_t)\right|\leq\sqrt{2T\log\left(\frac{4}{\delta}\right)}\right\}\geq 1-\frac{\delta}{2}$$

Thus, we can derive an upper bound for (b) as follows

$$\sum_{t=1}^{T}\mathbb{E}_{S}\left[\nabla_{\hat{\tau}_t}l_{1-\alpha}(\hat{\tau}_t,S)\right]$$

$$=\frac{1}{1-\epsilon}\sum_{t=1}^{T}\mathbb{E}_{\tilde{S}}\left[\nabla_{\hat{\tau}_t}l_{1-\alpha}(\hat{\tau}_t,\tilde{S})\right]+\frac{\epsilon}{1-\epsilon}\sum_{t=1}^{T}\left((1-\alpha)-\frac{1}{K}\mathbb{E}\left[\mathcal{C}_t(X_t)\right]\right)$$

$$=\frac{1}{1-\epsilon}\left(\sum_{t=1}^{T}\mathbb{E}_{\tilde{S}}\left[\nabla_{\hat{\tau}_t}l_{1-\alpha}(\hat{\tau}_t,\tilde{S})\right]-\sum_{t=1}^{T}\nabla_{\hat{\tau}_t}l_{1-\alpha}(\hat{\tau}_t,\tilde{S})+\sum_{t=1}^{T}\nabla_{\hat{\tau}_t}l_{1-\alpha}(\hat{\tau}_t,\tilde{S})\right)+\frac{\epsilon}{1-\epsilon}\sum_{t=1}^{T}\left((1-\alpha)-\frac{1}{K}\mathbb{E}\left[\mathcal{C}_t(X_t)\right]\right)$$

$$\leq\frac{1}{1-\epsilon}\left|\sum_{t=1}^{T}\mathbb{E}_{\tilde{S}}\left[\nabla_{\hat{\tau}_t}l_{1-\alpha}(\hat{\tau}_t,\tilde{S})\right]-\sum_{t=1}^{T}\nabla_{\hat{\tau}_t}l_{1-\alpha}(\hat{\tau}_t,\tilde{S})\right|+\frac{1}{1-\epsilon}\left|\sum_{t=1}^{T}\nabla_{\hat{\tau}_t}l_{1-\alpha}(\hat{\tau}_t,\tilde{S})\right|$$

$$+\frac{\epsilon}{1-\epsilon}\sum_{t=1}^{T}\left((1-\alpha)-\frac{1}{K}\mathbb{E}\left[\mathcal{C}_t(X_t)\right]\right)$$

$$\leq\frac{1}{1-\epsilon}\sqrt{2T\log\left(\frac{4}{\delta}\right)}+\frac{1}{1-\epsilon}\underbrace{\left|\sum_{t=1}^{T}\nabla_{\hat{\tau}_t}l_{1-\alpha}(\hat{\tau}_t,\tilde{S})\right|}_{(c)}+\frac{\epsilon}{1-\epsilon}\sum_{t=1}^{T}\left((1-\alpha)-\frac{1}{K}\mathbb{E}\left[\mathcal{C}_t(X_t)\right]\right)$$

Then, we derive an upper bound for (c). The update rule (Eq. (3)) gives us that

$$\hat{\tau}_{t+1}=\hat{\tau}_t-\eta\nabla_{\hat{\tau}_t}l_{1-\alpha}(\hat{\tau}_t,\tilde{S}_t)\quad\Longrightarrow\quad\nabla_{\hat{\tau}_t}l_{1-\alpha}(\hat{\tau}_t,\tilde{S}_t)=\frac{1}{\eta}(\hat{\tau}_t-\hat{\tau}_{t+1}).$$

Accumulating from $t=1$ to $t=T$ and taking absolute value gives

$$\left|\sum_{t=1}^{T}\nabla_{\hat{\tau}_t}l_{1-\alpha}(\hat{\tau}_t,S_t)\right|=\left|\sum_{t=1}^{T}\frac{1}{\eta}(\hat{\tau}_t-\hat{\tau}_{t+1})\right|=\frac{1}{\eta}|\hat{\tau}_1-\hat{\tau}_{T+1}|\overset{(a)}{\leq}\frac{1}{\eta}\left(1+(1-\alpha)\eta\right)=\frac{1}{\eta}+1-\alpha$$

where (a) follows from the assumption that $\hat{\tau}_1\in[0,1]$ and Lemma E.1. Thus, we have

$$\sum_{t=1}^{T}\mathbb{E}_{S}\left[\nabla_{\hat{\tau}_t}l_{1-\alpha}(\hat{\tau}_t,S)\right]\leq\frac{1}{1-\epsilon}\sqrt{2T\log\left(\frac{4}{\delta}\right)}+\frac{1}{1-\epsilon}\left(\frac{1}{\eta}+1-\alpha\right)+\frac{\epsilon}{1-\epsilon}\sum_{t=1}^{T}\left((1-\alpha)-\frac{1}{K}\mathbb{E}\left[\mathcal{C}_t(X_t)\right]\right)$$

By taking union bound and combining two parts, we can obtain that

$$\sum_{t=1}^{T}\nabla_{\tau_t}l_{1-\alpha}\left(\tau_t,S_t\right)\leq\sqrt{2T\log\left(\frac{4}{\delta}\right)}+\frac{1}{1-\epsilon}\sqrt{2T\log\left(\frac{4}{\delta}\right)}+\frac{1}{1-\epsilon}\left(\frac{1}{\eta}+1-\alpha\right)$$

$$+\frac{\epsilon}{1-\epsilon}\sum_{t=1}^{T}\left(\frac{1}{K}\mathbb{E}\left[\mathcal{C}_t(X_t)\right]-(1-\alpha)\right)$$

$$=\frac{2-\epsilon}{1-\epsilon}\sqrt{2T\log\left(\frac{4}{\delta}\right)}+\frac{1}{1-\epsilon}\left(\frac{1}{\eta}+1-\alpha\right)+\frac{\epsilon}{1-\epsilon}\sum_{t=1}^{T}\left((1-\alpha)-\frac{1}{K}\mathbb{E}\left[\mathcal{C}_t(X_t)\right]\right)$$

holds with at least $1-\delta$ probability. Recall that

$$\sum_{t=1}^{T}\nabla_{\tau_t}l_{1-\alpha}\left(\tau_t,S_t\right)=\sum_{t=1}^{T}\left(\mathbb{1}\left\{S_t\leq\tau_t\right\}-(1-\alpha)\right)=\sum_{t=1}^{T}\left(-\mathbb{1}\left\{S_t\geq\tau_t\right\}+\alpha\right)=\sum_{t=1}^{T}\left(-\mathbb{1}\left\{Y_t\notin\mathcal{C}_t(X_t)\right\}+\alpha\right)$$

We can conclude that

$$\alpha - \frac{1}{T}\sum_{t=1}^{T}\mathbb{1}\left\{Y_t \notin \mathcal{C}_t(X_t)\right\} \le \frac{2-\epsilon}{1-\epsilon}\sqrt{2\log\left(\frac{4}{\delta}\right)} \cdot \frac{1}{\sqrt{T}} + \frac{1}{1-\epsilon}\left(\frac{1}{\eta}+1-\alpha\right)\cdot\frac{1}{T}$$

$$+ \frac{\epsilon}{1-\epsilon}\cdot\frac{1}{T}\sum_{t=1}^{T}\left((1-\alpha)-\frac{1}{K}\mathbb{E}\left[\mathcal{C}_t(X_t)\right]\right)$$

□

## E.2   Proof for Proposition 4.1

**Proposition E.3** (Restatement of Proposition 4.1). *The robust pinball loss defined in Eq.* (4) *satisfies the following two properties:*

$$(1)\mathbb{E}_S\left[l_{1-\alpha}(\tau, S)\right] = \mathbb{E}_{\tilde{S},S_y}\left[\tilde{l}_{1-\alpha}(\tau, \tilde{S}, \{S_y\}_{y=1}^{K})\right];$$

$$(2)\mathbb{E}_S\left[\nabla_\tau l_{1-\alpha}(\tau, S)\right] = \mathbb{E}_{\tilde{S},S_y}\left[\nabla_\tau\tilde{l}_{1-\alpha}(\tau, \tilde{S}, \{S_y\}_{y=1}^{K})\right].$$

*Proof.* **The property (1):** We begin by proving the first property. Taking expectation on pinball loss gives

$$\mathbb{E}_S\left[l_{1-\alpha}(\tau, S)\right] = \mathbb{E}_S\left[\alpha(\tau-S)\mathbb{1}\{\tau\ge S\} + (1-\alpha)(S-\tau)\mathbb{1}\{\tau\le S\}\right]$$

$$= \alpha\tau\mathbb{P}\{\tau\ge S\} - \alpha\int_0^\tau s\mathbb{P}\{S=s\}\,ds - (1-\alpha)\tau\mathbb{P}\{\tau<S\} + (1-\alpha)\int_\tau^1 s\mathbb{P}\{S=s\}\,ds$$

$$= \underbrace{\alpha\tau\left(\frac{1}{1-\epsilon}\mathbb{P}\left\{\tilde{S}\le\tau\right\} + \frac{\epsilon}{1-\epsilon}\mathbb{P}\{\bar{S}\le\tau\}\right)}_{(a)} - \underbrace{\alpha\int_0^\tau s\left(\frac{1}{1-\epsilon}\mathbb{P}\left\{\tilde{S}\le\tau\right\} + \frac{\epsilon}{1-\epsilon}\mathbb{P}\{\bar{S}\le\tau\}\right)ds}_{(b)} -$$

$$\underbrace{(1-\alpha)\tau\left(\frac{1}{1-\epsilon}\mathbb{P}\left\{\tilde{S}>\tau\right\} + \frac{\epsilon}{1-\epsilon}\mathbb{P}\{\bar{S}>\tau\}\right)}_{(c)} + \underbrace{(1-\alpha)\int_\tau^1 s\left(\frac{1}{1-\epsilon}\mathbb{P}\left\{\tilde{S}>\tau\right\} + \frac{\epsilon}{1-\epsilon}\mathbb{P}\{\bar{S}>\tau\}\right)}_{(d)}$$

Part (a):

$$\alpha\tau\left(\frac{1}{1-\epsilon}\mathbb{P}\left\{\tilde{S}\le\tau\right\} + \frac{\epsilon}{1-\epsilon}\mathbb{P}\{\bar{S}\le\tau\}\right) = \alpha\tau\left(\frac{1}{1-\epsilon}\mathbb{P}\left\{\tilde{S}\le\tau\right\} + \frac{\epsilon}{K(1-\epsilon)}\sum_{y=1}^{K}\mathbb{P}\{S_y\le\tau\}\right)$$

$$= \mathbb{E}_{\tilde{S},S_y}\left[\alpha\tau\left(\frac{1}{1-\epsilon}\mathbb{1}\left\{\tilde{S}\le\tau\right\} + \frac{\epsilon}{K(1-\epsilon)}\sum_{y=1}^{K}\mathbb{1}\left\{S_y\le\tau\right\}\right)\right]$$

Part (b):

$$\alpha\int_0^\tau s\left(\frac{1}{1-\epsilon}\mathbb{P}\left\{\tilde{S}\le\tau\right\} + \frac{\epsilon}{1-\epsilon}\mathbb{P}\{\bar{S}\le\tau\}\right)ds = \alpha\int_0^\tau s\left(\frac{1}{1-\epsilon}\mathbb{P}\left\{\tilde{S}\le\tau\right\} + \frac{\epsilon}{K(1-\epsilon)}\sum_{y=1}^{K}\mathbb{P}\{S_y\le\tau\}\right)ds$$

$$= \mathbb{E}_{\tilde{S},S_y}\left[\alpha\left(\frac{\tilde{S}}{1-\epsilon}\mathbb{1}\left\{\tilde{S}\le\tau\right\} + \frac{\epsilon S_y}{K(1-\epsilon)}\sum_{y=1}^{K}\right)\mathbb{1}\left\{S_y\le\tau\right\}\right]$$

Part (c):

$$(1-\alpha)\tau\left(\frac{1}{1-\epsilon}\mathbb{P}\left\{\tilde{S}>\tau\right\} + \frac{\epsilon}{1-\epsilon}\mathbb{P}\{\bar{S}>\tau\}\right) = (1-\alpha)\tau\left(\frac{1}{1-\epsilon}\mathbb{P}\left\{\tilde{S}>\tau\right\} + \frac{\epsilon}{K(1-\epsilon)}\sum_{y=1}^{K}\mathbb{P}\{S_y>\tau\}\right)$$

$$= \mathbb{E}_{\tilde{S},S_y}\left[(1-\alpha)\tau\left(\frac{1}{1-\epsilon}\mathbb{1}\left\{\tilde{S}>\tau\right\} + \frac{\epsilon}{K(1-\epsilon)}\sum_{y=1}^{K}\mathbb{1}\left\{S_y>\tau\right\}\right)\right]$$

Part (d):

$$(1 - \alpha) \int_\tau^1 s \left( \frac{1}{1 - \epsilon} \mathbb{P} \left\{ \tilde{S} > \tau \right\} + \frac{\epsilon}{1 - \epsilon} \mathbb{P} \left\{ \bar{S} > \tau \right\} \right) ds$$

$$= (1 - \alpha) \int_0^\tau s \left( \frac{1}{1 - \epsilon} \mathbb{P} \left\{ \tilde{S} > \tau \right\} + \frac{\epsilon}{K(1 - \epsilon)} \sum_{y=1}^K \mathbb{P} \left\{ S_y > \tau \right\} \right) ds$$

$$= \mathbb{E}_{\tilde{S}, S_y} \left[ (1 - \alpha) \left( \frac{\tilde{S}}{1 - \epsilon} \mathbb{1} \left\{ \tilde{S} > \tau \right\} + \frac{\epsilon S_y}{K(1 - \epsilon)} \sum_{y=1}^K \right) \mathbb{1} \left\{ S_y > \tau \right\} \right]$$

Combining (a), (b), (c), and (d), we can conclude that

$$\mathbb{E}_S \left[ l_{1-\alpha}(\tau, S) \right]$$

$$= \mathbb{E}_{\tilde{S}, S_y} \left[ \alpha \tau \left( \frac{1}{1 - \epsilon} \mathbb{1} \left\{ \tilde{S} \le \tau \right\} + \frac{\epsilon}{K(1 - \epsilon)} \sum_{y=1}^K \mathbb{1} \left\{ S_y \le \tau \right\} \right) - \alpha \left( \frac{\tilde{S}}{1 - \epsilon} \mathbb{1} \left\{ \tilde{S} \le \tau \right\} - \frac{\epsilon S_y}{K(1 - \epsilon)} \sum_{y=1}^K \right) \mathbb{1} \left\{ S_y \le \tau \right\} \right] -$$

$$\mathbb{E}_{\tilde{S}, S_y} \left[ (1 - \alpha) \tau \left( \frac{1}{1 - \epsilon} \mathbb{1} \left\{ \tilde{S} > \tau \right\} + \frac{\epsilon}{K(1 - \epsilon)} \sum_{y=1}^K \mathbb{1} \left\{ S_y > \tau \right\} \right) \right.$$

$$\left. + (1 - \alpha) \left( \frac{\tilde{S}}{1 - \epsilon} \mathbb{1} \left\{ \tilde{S} > \tau \right\} + \frac{\epsilon S_y}{K(1 - \epsilon)} \sum_{y=1}^K \mathbb{1} \left\{ S_y > \tau \right\} \right) \right]$$

$$= \mathbb{E} \left[ \frac{1}{1 - \epsilon} \left( \alpha(\tau - \tilde{S}) \mathbb{1} \left\{ \tilde{S} \le \tau \right\} + (1 - \alpha)(\tilde{S} - \tau) \mathbb{1} \left\{ \tilde{S} > \tau \right\} \right) \right] +$$

$$\mathbb{E} \left[ \frac{\epsilon}{K(1 - \epsilon)} \sum_{y=1}^K \left( \alpha(\tau - \tilde{S}) \mathbb{1} \left\{ S_y \le \tau \right\} + (1 - \alpha)(\tilde{S} - \tau) \mathbb{1} \left\{ S_y > \tau \right\} \right) \right]$$

$$= \mathbb{E} \left[ \frac{1}{1 - \epsilon} l_{1-\alpha}(\tau, \tilde{S}) + \frac{\epsilon}{K(1 - \epsilon)} \sum_{y=1}^K l_{1-\alpha}(\tau, S_y) \right]$$

$$= \mathbb{E}_{\tilde{S}, S_y} \left[ \tilde{l}_{1-\alpha}(\tau, \tilde{S}, \{S_y\}_{y=1}^K) \right]$$

**The property (2):** We proceed by proving the second property, which demonstrates that the expected gradient of the robust pinball loss (computed using noise and random scores) equals the expected gradient of the true pinball loss. Consider the gradient of robust pinball loss:

$$\mathbb{E}_{\tilde{S}, S_y} \left[ \nabla_\tau \tilde{l}_{1-\alpha}(\tau, \tilde{S}, \{S_y\}_{y=1}^K) \right]$$

$$= \mathbb{E}_{\tilde{S}, S_y} \left[ \frac{\alpha}{1 - \epsilon} \mathbb{1} \left\{ \tau \ge \tilde{S} \right\} - \sum_{y=1}^K \frac{\alpha \epsilon}{K(1 - \epsilon)} \mathbb{1} \left\{ \tau \ge S_y \right\} - \frac{1 - \alpha}{1 - \epsilon} \mathbb{1} \left\{ \tau < \tilde{S} \right\} + \sum_{y=1}^K \frac{(1 - \alpha)\epsilon}{K(1 - \epsilon)} \mathbb{1} \left\{ \tau < S_y \right\} \right]$$

$$= \frac{\alpha}{1 - \epsilon} \mathbb{P} \left\{ \tau \ge \tilde{S} \right\} - \sum_{y=1}^K \frac{\alpha \epsilon}{K(1 - \epsilon)} \mathbb{P} \left\{ \tau \ge S_y \right\} - \frac{1 - \alpha}{1 - \epsilon} \mathbb{P} \left\{ \tau < \tilde{S} \right\} + \sum_{y=1}^K \frac{(1 - \alpha)\epsilon}{K(1 - \epsilon)} \mathbb{P} \left\{ \tau < S_y \right\}$$

$$= \alpha \mathbb{P} \left\{ \tau \ge S \right\} - (1 - \alpha) \mathbb{P} \left\{ \tau < S \right\}$$

$$= \mathbb{E}_S \left[ \nabla_\tau l_{1-\alpha}(\tau, S) \right]$$

$\square$

### E.3  Proof for Proposition 4.2

**Lemma E.4.** *Consider online conformal prediction under uniform label noise with noise rate $\epsilon \in (0, 1)$. Given Assumptions 2.1 and 2.2, when updating the threshold according to Eq. (5), for any $\delta \in (0, 1)$ and $T \in \mathbb{N}^+$, we have*

$$-\alpha \eta - \frac{\epsilon \eta}{1 - \epsilon} \le \hat{\tau}_T \le 1 - \alpha \eta + \frac{\eta}{1 - \epsilon}. \tag{9}$$

*Proof.* We prove this by induction. First, we know $\hat{\tau}_1 \in [0,1]$ by assumption, which indicates that Eq. (9) is satisfied at $t = 1$. Then, we assume that Eq. (9) holds for $t = T$, and we will show that $\hat{\tau}_{T+1}$ lies in this range. Consider three cases:

**Case 1.** If $\hat{\tau}_T \in [0,1]$, we have

$$\hat{\tau}_{T+1} = \hat{\tau}_T - \eta \cdot \nabla_{\hat{\tau}_T}\tilde{l}_{1-\alpha}(\hat{\tau}_T, \tilde{S}_T, \{S_{t,y}\}_{y=1}^K) \overset{(a)}{\in} \left[-\alpha\eta - \frac{\epsilon\eta}{1-\epsilon}, 1 - \alpha\eta + \frac{\eta}{1-\epsilon}\right]$$

where (a) follows from Lemma F.2.

**Case 2.** Consider the case where $\hat{\tau}_T \in [1, 1 + \alpha\eta - \eta/(1-\epsilon)]$. The assumption that $\tilde{S}_T, S_{t,y} \in [0,1]$ implies $\mathbb{1}\left\{\tilde{S}_T \leq \hat{\tau}_T\right\} = \mathbb{1}\left\{S_{t,y} \leq \hat{\tau}_T\right\} = 1$. Thus, we have

$$\nabla_{\hat{\tau}_T}\tilde{l}_{1-\alpha}(\hat{\tau}_T, \tilde{S}_T, \{S_{t,y}\}_{y=1}^K) = \frac{\alpha}{1-\epsilon}\mathbb{1}\left\{\hat{\tau}_T \geq \tilde{S}_T\right\} - \sum_{y=1}^K \frac{\alpha\epsilon}{K(1-\epsilon)}\mathbb{1}\left\{\hat{\tau}_T \geq S_{t,y}\right\} - \frac{1-\alpha}{1-\epsilon}\mathbb{1}\left\{\hat{\tau}_T \geq \tilde{S}_T\right\}$$

$$+ \sum_{y=1}^K \frac{(1-\alpha)\epsilon}{K(1-\epsilon)}\mathbb{1}\left\{\hat{\tau}_T \geq S_{t,y}\right\}$$

$$= \frac{\alpha}{1-\epsilon} - \sum_{y=1}^K \frac{\alpha\epsilon}{K(1-\epsilon)} = \alpha,$$

which follows that

$$\hat{\tau}_{T+1} = \hat{\tau}_T - \eta \cdot \nabla_{\hat{\tau}_T}\tilde{l}_{1-\alpha}(\hat{\tau}_T, \tilde{S}_T, \{S_{t,y}\}_{y=1}^K) = \hat{\tau}_T - \eta\alpha \in \left[1 - \eta\alpha, 1 - \frac{\eta}{1-\epsilon}\right] \subset \left[-\alpha\eta - \frac{\epsilon\eta}{1-\epsilon}, 1 - \alpha\eta + \frac{\eta}{1-\epsilon}\right]$$

**Case 3.** Consider the case where $\hat{\tau}_T \in [-\alpha\eta - \epsilon\eta/(1-\epsilon), 0]$. The assumption that $\tilde{s}, s_y \in [0,1]$ implies $\mathbb{1}\{\tilde{s} \leq \hat{\tau}_T\} = \mathbb{1}\{s_y \leq \hat{\tau}_T\} = 0$. Thus, we have

$$\nabla_{\hat{\tau}_T}\tilde{l}_{1-\alpha}(\hat{\tau}_T, \tilde{S}_T, \{S_{t,y}\}_{y=1}^K) = \frac{\alpha}{1-\epsilon}\mathbb{1}\left\{\hat{\tau}_T \geq \tilde{S}_T\right\} - \sum_{y=1}^K \frac{\alpha\epsilon}{K(1-\epsilon)}\mathbb{1}\left\{\hat{\tau}_T \geq S_{t,y}\right\} - \frac{1-\alpha}{1-\epsilon}\mathbb{1}\left\{\hat{\tau}_T \geq \tilde{S}_T\right\}$$

$$+ \sum_{y=1}^K \frac{(1-\alpha)\epsilon}{K(1-\epsilon)}\mathbb{1}\left\{\hat{\tau}_T \geq S_{t,y}\right\}$$

$$= -\frac{1-\alpha}{1-\epsilon}\mathbb{1}\{\tau < \tilde{s}\} + \sum_{y=1}^K \frac{(1-\alpha)\epsilon}{K(1-\epsilon)} = \alpha - 1,$$

which follows that

$$\hat{\tau}_{T+1} = \hat{\tau}_T - \eta \cdot \nabla_{\hat{\tau}_T}\tilde{l}_{1-\alpha}(\hat{\tau}_T, \tilde{S}_T, \{S_{t,y}\}_{y=1}^K) = \hat{\tau}_T + \eta(1-\alpha) \in \left[-\alpha\eta - \frac{\epsilon\eta}{1-\epsilon} + \eta(1-\alpha), \eta(1-\alpha)\right]$$

$$\subset \left[-\alpha\eta - \frac{\epsilon\eta}{1-\epsilon}, 1 - \alpha\eta + \frac{\eta}{1-\epsilon}\right]$$

Combining three cases, we can conclude that

$$-\alpha\eta - \frac{\epsilon\eta}{1-\epsilon} \leq \hat{\tau}_t \leq 1 - \alpha\eta + \frac{\eta}{1-\epsilon}$$

$\square$

**Proposition E.5** (Restatement of Proposition 4.2). *Consider online conformal prediction under uniform label noise with noise rate $\epsilon \in (0,1)$. Given Assumptions 2.1 and 2.2, when updating the threshold according to Eq. (5), for any $\delta \in (0,1)$ and $T \in \mathbb{N}^+$, the following bound holds with probability at least $1 - \delta$:*

$$\text{ExErr(T)} \leq \sqrt{\frac{\log(2/\delta)}{1-\epsilon}} \cdot \frac{1}{\sqrt{T}} + \left(\frac{1}{\eta} - \alpha + \frac{\epsilon}{1-\epsilon}\right) \cdot \frac{1}{T}.$$

*Proof.* The update rule of the threshold $\hat{\tau}_t$ gives us that

$$\hat{\tau}_{t+1} = \hat{\tau}_t - \eta_t \cdot \nabla_{\hat{\tau}_t} \tilde{l}_{1-\alpha}(\hat{\tau}_t, \tilde{S}_t, \{S_{t,y}\}_{y=1}^K) \implies \nabla_{\hat{\tau}_t} \tilde{l}_{1-\alpha}(\hat{\tau}_t, \tilde{S}_t, \{S_{t,y}\}_{y=1}^K) = \frac{1}{\eta}(\hat{\tau}_t - \hat{\tau}_{t+1}).$$

Accumulating from $t = 1$ to $t = T$ and taking absolute value gives

$$\left| \sum_{t=1}^T \nabla_{\hat{\tau}_t} \tilde{l}_{1-\alpha}(\hat{\tau}_t, \tilde{S}_t, \{S_{t,y}\}_{y=1}^K) \right| = \left| \sum_{t=1}^T \frac{1}{\eta}(\hat{\tau}_t - \hat{\tau}_{t+1}) \right| = \frac{1}{\eta}|\hat{\tau}_1 - \hat{\tau}_{T+1}| \overset{(a)}{\leq} \frac{1}{\eta}\left(1 - \alpha\eta + \frac{\epsilon\eta}{1-\epsilon}\right) = \frac{1}{\eta} - \alpha + \frac{\epsilon}{1-\epsilon}$$

where (a) follows from the assumption that $\hat{\tau}_1 \in [0,1]$ and Lemma E.4. In addition, Lemma F.5 gives us that

$$\mathbb{P}\left\{ \left| \sum_{t=1}^T \mathbb{E}_{\tilde{S}_t, S_{t,y}} \left[ \nabla_{\hat{\tau}_t} \tilde{l}_{1-\alpha}(\hat{\tau}_t, \tilde{S}_t, \{S_{t,y}\}_{y=1}^K) \right] - \sum_{t=1}^T \nabla_{\hat{\tau}_t} \tilde{l}_{1-\alpha}(\hat{\tau}_t, \tilde{S}_t, \{S_{t,y}\}_{y=1}^K) \right| \leq \frac{\sqrt{2T\log(2/\delta)}}{1-\epsilon} \right\} \geq 1 - \delta$$

Thus, with at least $1 - \delta$ probability, we have

$$\left| \sum_{t=1}^T \mathbb{E}_{\tilde{S}_t, S_{t,y}} \left[ \nabla_{\hat{\tau}_t} \tilde{l}_{1-\alpha}(\hat{\tau}_t, \tilde{S}_t, \{S_{t,y}\}_{y=1}^K) \right] \right|$$

$$= \left| \sum_{t=1}^T \mathbb{E}_{\tilde{S}_t, S_{t,y}} \left[ \nabla_{\hat{\tau}_t} \tilde{l}_{1-\alpha}(\hat{\tau}_t, \tilde{S}_t, \{S_{t,y}\}_{y=1}^K) \right] - \sum_{t=1}^T \nabla_{\hat{\tau}_t} \tilde{l}_{1-\alpha}(\hat{\tau}_t, \tilde{S}_t, \{S_{t,y}\}_{y=1}^K) + \sum_{t=1}^T \nabla_{\hat{\tau}_t} \tilde{l}_{1-\alpha}(\hat{\tau}_t, \tilde{S}_t, \{S_{t,y}\}_{y=1}^K) \right|$$

$$\leq \left| \sum_{t=1}^T \mathbb{E}_{\tilde{S}_t, S_{t,y}} \left[ \nabla_{\hat{\tau}_t} \tilde{l}_{1-\alpha}(\hat{\tau}_t, \tilde{S}_t, \{S_{t,y}\}_{y=1}^K) \right] - \sum_{t=1}^T \nabla_{\hat{\tau}_t} \tilde{l}_{1-\alpha}(\hat{\tau}_t, \tilde{S}_t, \{S_{t,y}\}_{y=1}^K) \right| + \left| \sum_{t=1}^T \nabla_{\hat{\tau}_t} \tilde{l}_{1-\alpha}(\hat{\tau}_t, \tilde{S}_t, \{S_{t,y}\}_{y=1}^K) \right|$$

$$\leq \frac{\sqrt{2T\log(2/\delta)}}{1-\epsilon} + \frac{1}{\eta} + \alpha + \frac{\epsilon}{1-\epsilon}$$

Applying the second property in Proposition 4.1, we have

$$\sum_{t=1}^T \mathbb{E}_{\tilde{S}, S_y} \left[ \nabla_{\hat{\tau}_t} \tilde{l}_{1-\alpha}(\hat{\tau}_t, \tilde{S}, \{S_y\}_{y=1}^K) \right] = \sum_{t=1}^T \mathbb{E}_{S_t} \left[ \nabla_{\hat{\tau}_t} l_{1-\alpha}(\hat{\tau}_t, S_t) \right] = \sum_{t=1}^T \left[ \mathbb{P}\{S_t \leq \hat{\tau}_t\} - (1-\alpha) \right]$$

which implies that with at least $1 - \delta$ probability,

$$\left| \sum_{t=1}^T \left[ \mathbb{P}\{Y_t \notin \mathcal{C}_t(X_t)\} - \alpha \right] \right| = \left| \sum_{t=1}^T \left[ \mathbb{P}\{S_t \leq \hat{\tau}_t\} - (1-\alpha) \right] \right| \leq \frac{\sqrt{2T\log(2/\delta)}}{1-\epsilon} + \frac{1}{\eta} - \alpha + \frac{\epsilon}{1-\epsilon}$$

$$(10)$$

Therefore, we can conclude that

$$\text{ExErr(T)} = \left| \frac{1}{T} \sum_{t=1}^T \mathbb{P}\{Y_t \notin \mathcal{C}_t(X_t)\} - \alpha \right| \leq \sqrt{\frac{\log(2/\delta)}{1-\epsilon}} \cdot \frac{1}{\sqrt{T}} + \left( \frac{1}{\eta} - \alpha + \frac{\epsilon}{1-\epsilon} \right) \cdot \frac{1}{T}$$

holds with at least $1 - \delta$ probability. $\qquad\square$

## E.4  Proof for Proposition 4.3

**Proposition E.6** (Restatement of Proposition 4.3). *Consider online conformal prediction under uniform label noise with noise rate $\epsilon \in (0,1)$. Given Assumptions 2.1 and 2.2, when updating the threshold according to Eq. (5), for any $\delta \in (0,1)$ and $T \in \mathbb{N}^+$, the following bound holds with probability at least $1 - \delta$:*

$$\text{EmErr(T)} \leq \frac{2-\epsilon}{1-\epsilon}\sqrt{2\log\left(\frac{4}{\delta}\right)} \cdot \frac{1}{\sqrt{T}} + \left( \frac{1}{\eta} - \alpha + \frac{\epsilon}{1-\epsilon} \right) \cdot \frac{1}{T}.$$

*Proof.* Lemma F.3 gives us that

$$\mathbb{P}\left\{ \left| \sum_{t=1}^T \mathbb{E}_{S_t}\left[ \nabla_{\hat{\tau}_t} l_{1-\alpha}(\hat{\tau}_t, S_t) \right] - \sum_{t=1}^T \nabla_{\hat{\tau}_t} l_{1-\alpha}(\hat{\tau}_t, S_t) \right| \leq \sqrt{2T\log\left(\frac{4}{\delta}\right)} \right\} \geq 1 - \frac{\delta}{2}$$

Follows from Proposition 4.2 (see Eq. (10)), we know

$$\left|\sum_{t=1}^{T} \mathbb{E}_{S_t}\left[\nabla_{\hat{\tau}_t} l_{1-\alpha}(\hat{\tau}_t, S_t)\right]\right| = \left|\sum_{t=1}^{T} \mathbb{E}_{\tilde{S}, S_y}\left[\nabla_{\hat{\tau}_t} \tilde{l}_{1-\alpha}(\hat{\tau}_t, \tilde{S}, \{S_y\}_{y=1}^{K})\right]\right| \leq \frac{\sqrt{2T\log(4/\delta)}}{1-\epsilon} + \frac{1}{\eta} - \alpha + \frac{\epsilon}{1-\epsilon}$$

holds with at least $1 - \delta/2$ probability. In addition, recall that

$$\sum_{t=1}^{T} \nabla_{\hat{\tau}_t} l_{1-\alpha}(\hat{\tau}_t, S_t) = \sum_{t=1}^{T}\left[\mathbb{1}\{S_t \leq \hat{\tau}_t\} - (1-\alpha)\right]$$

Thus, by union bound, we can obtain that with at least probability $1 - \delta$,

$$\left|\sum_{t=1}^{T} \nabla_{\hat{\tau}_t} l_{1-\alpha}(\hat{\tau}_t, S_t)\right| = \left|\sum_{t=1}^{T} \nabla_{\hat{\tau}_t} l_{1-\alpha}(\hat{\tau}_t, S_t) - \sum_{t=1}^{T} \mathbb{E}_{S_t}\left[\nabla_{\hat{\tau}_t} l_{1-\alpha}(\hat{\tau}_t, S_t)\right] + \sum_{t=1}^{T} \mathbb{E}_{S_t}\left[\nabla_{\hat{\tau}_t} l_{1-\alpha}(\hat{\tau}_t, S_t)\right]\right|$$

$$\leq \left|\sum_{t=1}^{T} \nabla_{\hat{\tau}_t} l_{1-\alpha}(\hat{\tau}_t, S_t) - \sum_{t=1}^{T} \mathbb{E}_{S_t}\left[\nabla_{\hat{\tau}_t} l_{1-\alpha}(\hat{\tau}_t, S_t)\right]\right| + \left|\sum_{t=1}^{T} \mathbb{E}_{S_t}\left[\nabla_{\hat{\tau}_t} l_{1-\alpha}(\hat{\tau}_t, S_t)\right]\right|$$

$$\leq \sqrt{2T\log\left(\frac{4}{\delta}\right)} + \frac{\sqrt{2T\log(4/\delta)}}{1-\epsilon} + \frac{1}{\eta} - \alpha + \frac{\epsilon}{1-\epsilon}$$

$$= \frac{2-\epsilon}{1-\epsilon} \cdot \sqrt{2T\log\left(\frac{4}{\delta}\right)} + \frac{1}{\eta} - \alpha + \frac{\epsilon}{1-\epsilon}$$

Recall that

$$\left|\sum_{t=1}^{T}\left[\mathbb{1}\{Y_t \notin \mathcal{C}_t(X_t)\} - \alpha\right]\right| = \left|\sum_{t=1}^{T}\left[\mathbb{1}\{S_t \leq \hat{\tau}_t\} - (1-\alpha)\right]\right| = \left|\sum_{t=1}^{T} \nabla_{\hat{\tau}_t} l_{1-\alpha}(\hat{\tau}_t, S_t)\right|$$

Therefore, we can conclude that

$$\text{EmErr(T)} = \left|\frac{1}{T}\sum_{t=1}^{T} \mathbb{1}\{Y_t \notin \mathcal{C}_t(X_t)\} - \alpha\right| = \left|\frac{1}{T}\sum_{t=1}^{T} \nabla_{\hat{\tau}_t} l_{1-\alpha}(\hat{\tau}_t, S_t)\right|$$

$$\leq \frac{2-\epsilon}{1-\epsilon}\sqrt{2\log\left(\frac{4}{\delta}\right)} \cdot \frac{1}{\sqrt{T}} + \left(\frac{1}{\eta} - \alpha + \frac{\epsilon}{1-\epsilon}\right) \cdot \frac{1}{T}$$

holds with at least $1 - \delta$ probability. $\qquad\square$

## E.5 Proof for Proposition 4.5

**Proposition E.7** (Restatement of Proposition 4.5). *Consider online conformal prediction under uniform label noise with noise rate $\epsilon \in (0, 1)$. Given Assumptions 2.1 and 2.2, when updating the threshold according to Eq. (6), for any $\delta \in (0, 1)$ and $T \in \mathbb{N}^+$, the following bound holds with probability at least $1 - \delta$:*

$$\text{ExErr(T)} \leq \sqrt{\frac{\log(2/\delta)}{1-\epsilon}} \cdot \frac{1}{\sqrt{T}} + \left[\left(1 + \max_{1 \leq t \leq T-1} \eta_t \cdot \frac{1+\epsilon}{1-\epsilon}\right)\sum_{t=1}^{T} \left|\eta_t^{-1} - \eta_{t-1}^{-1}\right|\right] \cdot \frac{1}{T}.$$

*Proof.* Define $\eta_0^{-1} = 0$:

$$\left| \sum_{t=1}^{T} \nabla_{\hat{\tau}_t} \tilde{l}_{1-\alpha}(\hat{\tau}_t, \tilde{S}_t, \{S_{t,y}\}_{y=1}^{K}) \right| = \left| \sum_{t=1}^{T} \eta_t^{-1} \cdot \left( \eta_t \cdot \nabla_{\hat{\tau}_t} \tilde{l}_{1-\alpha}(\hat{\tau}_t, \tilde{S}_t, \{S_{t,y}\}_{y=1}^{K}) \right) \right|$$

$$= \left| \sum_{t=1}^{T} (\eta_t^{-1} - \eta_{t-1}^{-1}) \cdot \left( \sum_{s=t}^{T} \eta_s \cdot \nabla_{\hat{\tau}_s} \tilde{l}_{1-\alpha}(\hat{\tau}_s, \tilde{S}_s, \{S_{y,s}\}_{y=1}^{K}) \right) \right|$$

$$\overset{(a)}{=} \left| \sum_{t=1}^{T} (\eta_t^{-1} - \eta_{t-1}^{-1}) \cdot (\hat{\tau}_T - \hat{\tau}_t) \right|$$

$$\overset{(b)}{\leq} \sum_{t=1}^{T} \left| \eta_t^{-1} - \eta_{t-1}^{-1} \right| \cdot |\hat{\tau}_T - \hat{\tau}_t|$$

$$\overset{(c)}{\leq} \left( 1 + \max_{1 \leq t \leq T-1} \eta_t \cdot \frac{1+\epsilon}{1-\epsilon} \right) \sum_{t=1}^{T} \left| \eta_t^{-1} - \eta_{t-1}^{-1} \right|$$

where (a) comes from the update rule (Eq. (6)), (b) is due to triangle inequality, and (c) follows from Lemma **??**. In addition, Lemma F.5 gives us that

$$\mathbb{P} \left\{ \left| \sum_{t=1}^{T} \mathbb{E}_{\tilde{S}_t, S_{t,y}} \left[ \nabla_{\hat{\tau}_t} \tilde{l}_{1-\alpha}(\hat{\tau}_t, \tilde{S}_t, \{S_{t,y}\}_{y=1}^{K}) \right] - \sum_{t=1}^{T} \nabla_{\hat{\tau}_t} \tilde{l}_{1-\alpha}(\hat{\tau}_t, \tilde{S}_t, \{S_{t,y}\}_{y=1}^{K}) \right| \leq \frac{\sqrt{2T \log(2/\delta)}}{1-\epsilon} \right\} \geq 1 - \delta$$

Thus, with at least $1 - \delta$ probability, we have

$$\left| \sum_{t=1}^{T} \mathbb{E}_{\tilde{S}_t, S_{t,y}} \left[ \nabla_{\hat{\tau}_t} \tilde{l}_{1-\alpha}(\hat{\tau}_t, \tilde{S}_t, \{S_{t,y}\}_{y=1}^{K}) \right] \right|$$

$$= \left| \sum_{t=1}^{T} \mathbb{E}_{\tilde{S}_t, S_{t,y}} \left[ \nabla_{\hat{\tau}_t} \tilde{l}_{1-\alpha}(\hat{\tau}_t, \tilde{S}_t, \{S_{t,y}\}_{y=1}^{K}) \right] - \sum_{t=1}^{T} \nabla_{\hat{\tau}_t} \tilde{l}_{1-\alpha}(\hat{\tau}_t, \tilde{S}_t, \{S_{t,y}\}_{y=1}^{K}) + \sum_{t=1}^{T} \nabla_{\hat{\tau}_t} \tilde{l}_{1-\alpha}(\hat{\tau}_t, \tilde{S}_t, \{S_{t,y}\}_{y=1}^{K}) \right|$$

$$\leq \left| \sum_{t=1}^{T} \mathbb{E}_{\tilde{S}_t, S_{t,y}} \left[ \nabla_{\hat{\tau}_t} \tilde{l}_{1-\alpha}(\hat{\tau}_t, \tilde{S}_t, \{S_{t,y}\}_{y=1}^{K}) \right] - \sum_{t=1}^{T} \nabla_{\hat{\tau}_t} \tilde{l}_{1-\alpha}(\hat{\tau}_t, \tilde{S}_t, \{S_{t,y}\}_{y=1}^{K}) \right| + \left| \sum_{t=1}^{T} \nabla_{\hat{\tau}_t} \tilde{l}_{1-\alpha}(\hat{\tau}_t, \tilde{S}_t, \{S_{t,y}\}_{y=1}^{K}) \right|$$

$$\leq \frac{\sqrt{2T \log(2/\delta)}}{1-\epsilon} + \left( 1 + \max_{1 \leq t \leq T-1} \eta_t \cdot \frac{1+\epsilon}{1-\epsilon} \right) \sum_{t=1}^{T} \left| \eta_t^{-1} - \eta_{t-1}^{-1} \right|$$

Applying the second property in Proposition 4.1, we have

$$\sum_{t=1}^{T} \mathbb{E}_{\tilde{S}, S_y} \left[ \nabla_{\hat{\tau}_t} \tilde{l}_{1-\alpha}(\hat{\tau}_t, \tilde{S}, \{S_y\}_{y=1}^{K}) \right] = \sum_{t=1}^{T} \mathbb{E}_{S_t} \left[ \nabla_{\hat{\tau}_t} l_{1-\alpha}(\hat{\tau}_t, S_t) \right] = \sum_{t=1}^{T} \left[ \mathbb{P}\{S_t \leq \hat{\tau}_t\} - (1-\alpha) \right]$$

which implies that with at least $1 - \delta$ probability,

$$\left| \sum_{t=1}^{T} \left[ \mathbb{P}\{Y_t \notin C_t(X_t)\} - \alpha \right] \right| = \left| \sum_{t=1}^{T} \left[ \mathbb{P}\{S_t \leq \hat{\tau}_t\} - (1-\alpha) \right] \right|$$

$$\leq \frac{\sqrt{2T \log(2/\delta)}}{1-\epsilon} + \left( 1 + \max_{1 \leq t \leq T-1} \eta_t \cdot \frac{1+\epsilon}{1-\epsilon} \right) \sum_{t=1}^{T} \left| \eta_t^{-1} - \eta_{t-1}^{-1} \right| \tag{11}$$

Therefore, we can conclude that

$$\text{ExErr(T)} = \left| \frac{1}{T} \sum_{t=1}^{T} \mathbb{P}\{Y_t \notin C_t(X_t)\} - \alpha \right| \leq \sqrt{\frac{\log(2/\delta)}{1-\epsilon}} \cdot \frac{1}{\sqrt{T}} + \left[ \left( 1 + \max_{1 \leq t \leq T-1} \eta_t \cdot \frac{1+\epsilon}{1-\epsilon} \right) \sum_{t=1}^{T} \left| \eta_t^{-1} - \eta_{t-1}^{-1} \right| \right] \cdot \frac{1}{T}$$

holds with at least $1 - \delta$ probability. $\qquad\square$

### E.6 Proof for Proposition 4.6

**Proposition E.8** (Restatement of Proposition 4.6). *Consider online conformal prediction under uniform label noise with noise rate $\epsilon \in (0, 1)$. Given Assumptions 2.1 and 2.2, when updating the threshold according to Eq. (6), for any $\delta \in (0, 1)$ and $T \in \mathbb{N}^+$, the following bound holds with probability at least $1 - \delta$:*

$$\text{EmErr(T)} \leq \frac{2 - \epsilon}{1 - \epsilon} \sqrt{2 \log\left(\frac{4}{\delta}\right)} \cdot \frac{1}{\sqrt{T}} + \left(1 + \max_{1 \leq t \leq T-1} \eta_t \cdot \frac{1 + \epsilon}{1 - \epsilon}\right) \sum_{t=1}^{T} |\eta_t^{-1} - \eta_{t-1}^{-1}| \cdot \frac{1}{T}.$$

*Proof.* Lemma F.3 gives us that

$$\mathbb{P}\left\{\left|\sum_{t=1}^{T} \mathbb{E}_{S_t}\left[\nabla_{\hat{\tau}_t} l_{1-\alpha}(\hat{\tau}_t, S_t)\right] - \sum_{t=1}^{T} \nabla_{\hat{\tau}_t} l_{1-\alpha}(\hat{\tau}_t, S_t)\right| \leq \sqrt{2T \log\left(\frac{4}{\delta}\right)}\right\} \geq 1 - \frac{\delta}{2}$$

Follows from Proposition 4.5 (see Eq. (11)), we know

$$\left|\sum_{t=1}^{T} \mathbb{E}_{S_t}\left[\nabla_{\hat{\tau}_t} l_{1-\alpha}(\hat{\tau}_t, S_t)\right]\right| = \left|\sum_{t=1}^{T} \mathbb{E}_{\tilde{S}, S_y}\left[\nabla_{\hat{\tau}_t} \tilde{l}_{1-\alpha}(\hat{\tau}_t, \tilde{S}, \{S_y\}_{y=1}^{K})\right]\right|$$

$$\leq \frac{\sqrt{2T \log(2/\delta)}}{1 - \epsilon} + \left(1 + \max_{1 \leq t \leq T-1} \eta_t \cdot \frac{1 + \epsilon}{1 - \epsilon}\right) \sum_{t=1}^{T} |\eta_t^{-1} - \eta_{t-1}^{-1}|$$

holds with at least $1 - \delta/2$ probability. In addition, recall that

$$\sum_{t=1}^{T} \nabla_{\hat{\tau}_t} l_{1-\alpha}(\hat{\tau}_t, S_t) = \sum_{t=1}^{T} \left[\mathbb{1}\{S_t \leq \hat{\tau}_t\} - (1 - \alpha)\right]$$

Thus, by union bound, we can obtain that with at least probability $1 - \delta$,

$$\left|\sum_{t=1}^{T} \nabla_{\hat{\tau}_t} l_{1-\alpha}(\hat{\tau}_t, S_t)\right| = \left|\sum_{t=1}^{T} \nabla_{\hat{\tau}_t} l_{1-\alpha}(\hat{\tau}_t, S_t) - \sum_{t=1}^{T} \mathbb{E}_{S_t}\left[\nabla_{\hat{\tau}_t} l_{1-\alpha}(\hat{\tau}_t, S_t)\right] + \sum_{t=1}^{T} \mathbb{E}_{S_t}\left[\nabla_{\hat{\tau}_t} l_{1-\alpha}(\hat{\tau}_t, S_t)\right]\right|$$

$$\leq \left|\sum_{t=1}^{T} \nabla_{\hat{\tau}_t} l_{1-\alpha}(\hat{\tau}_t, S_t) - \sum_{t=1}^{T} \mathbb{E}_{S_t}\left[\nabla_{\hat{\tau}_t} l_{1-\alpha}(\hat{\tau}_t, S_t)\right]\right| + \left|\sum_{t=1}^{T} \mathbb{E}_{S_t}\left[\nabla_{\hat{\tau}_t} l_{1-\alpha}(\hat{\tau}_t, S_t)\right]\right|$$

$$\leq \sqrt{2T \log\left(\frac{4}{\delta}\right)} + \frac{\sqrt{2T \log(2/\delta)}}{1 - \epsilon} + \left(1 + \max_{1 \leq t \leq T-1} \eta_t \cdot \frac{1 + \epsilon}{1 - \epsilon}\right) \sum_{t=1}^{T} |\eta_t^{-1} - \eta_{t-1}^{-1}|$$

$$= \frac{2 - \epsilon}{1 - \epsilon} \cdot \sqrt{2T \log\left(\frac{4}{\delta}\right)} + \left(1 + \max_{1 \leq t \leq T-1} \eta_t \cdot \frac{1 + \epsilon}{1 - \epsilon}\right) \sum_{t=1}^{T} |\eta_t^{-1} - \eta_{t-1}^{-1}|$$

Recall that

$$\left|\sum_{t=1}^{T} \left[\mathbb{1}\{Y_t \notin C_t(X_t)\} - \alpha\right]\right| = \left|\sum_{t=1}^{T} \left[\mathbb{1}\{S_t \leq \hat{\tau}_t\} - (1 - \alpha)\right]\right| = \left|\sum_{t=1}^{T} \nabla_{\hat{\tau}_t} l_{1-\alpha}(\hat{\tau}_t, S_t)\right|$$

Therefore, we can conclude that

$$\text{EmErr(T)} = \left|\frac{1}{T} \sum_{t=1}^{T} \mathbb{1}\{Y_t \notin C_t(X_t)\} - \alpha\right| = \left|\frac{1}{T} \sum_{t=1}^{T} \nabla_{\hat{\tau}_t} l_{1-\alpha}(\hat{\tau}_t, S_t)\right|$$

$$\leq \frac{2 - \epsilon}{1 - \epsilon} \sqrt{2 \log\left(\frac{4}{\delta}\right)} \cdot \frac{1}{\sqrt{T}} + \left(1 + \max_{1 \leq t \leq T-1} \eta_t \cdot \frac{1 + \epsilon}{1 - \epsilon}\right) \sum_{t=1}^{T} |\eta_t^{-1} - \eta_{t-1}^{-1}| \cdot \frac{1}{T}$$

holds with at least $1 - \delta$ probability. □

### E.7 Proof for Proposition B.1

**Proposition E.9** (Restatement of Proposition B.1). *Consider online conformal prediction under uniform label noise with noise rate $\epsilon \in (0, 1)$. Given Assumptions 2.1 and 2.2, when updating the threshold according to Eq. (6), for any $T \in \mathbb{N}^+$, we have:*

$$\sum_{t=1}^{T}\left(\tilde{l}_{1-\alpha}(\hat{\tau}_t, \tilde{S}_t, \{S_{t,y}\}_{y=1}^{K}) - \tilde{l}_{1-\alpha}(\tau^*, \tilde{S}_t, \{S_{t,y}\}_{y=1}^{K})\right) \le \frac{1}{2\eta_T}\left(1 + \max_{1\le t\le T-1}\eta_t \cdot \frac{1+\epsilon}{1-\epsilon}\right)^2 + \left(\frac{1+\epsilon}{1-\epsilon}\right)^2 \cdot \sum_{t=1}^{T}\frac{\eta_t}{2}.$$

*Proof.* The update rule (Eq. (2)) gives us that

$$(\hat{\tau}_{t+1} - \tau^*)^2 = (\hat{\tau}_t - \tau^*)^2 + 2\eta_t(\tau^* - \hat{\tau}_t)\cdot\nabla_{\hat{\tau}_t}\tilde{l}_{1-\alpha}(\hat{\tau}_t, \tilde{S}_t, \{S_{t,y}\}_{y=1}^{K}) + \eta_t^2\cdot\left(\nabla_{\hat{\tau}_t}\tilde{l}_{1-\alpha}(\hat{\tau}_t, \tilde{S}_t, \{S_{t,y}\}_{y=1}^{K})\right)^2.$$

Recall that the robust pinball loss is defined as

$$\tilde{l}_{1-\alpha}(\tau, \tilde{S}, \{S_y\}_{y=1}^{K}) = \frac{1}{1-\epsilon}l_{1-\alpha}(\tau, \tilde{S}) + \frac{\epsilon}{K(1-\epsilon)}\sum_{y=1}^{K}l_{1-\alpha}(\tau, S_y).$$

Since pinball loss is convex, robust pinball loss inherits the convexity property. Thus, we have

$$(\tau^* - \hat{\tau}_t)\cdot\nabla_{\hat{\tau}_t}\tilde{l}_{1-\alpha}(\hat{\tau}_t, \tilde{S}_t, \{S_{t,y}\}_{y=1}^{K}) \le \tilde{l}_{1-\alpha}(\tau^*, \tilde{S}_t, \{S_{t,y}\}_{y=1}^{K}) - \tilde{l}_{1-\alpha}(\hat{\tau}_t, \tilde{S}_t, \{S_{t,y}\}_{y=1}^{K}).$$

It follows that

$$(\hat{\tau}_{t+1} - \tau^*)^2 \le (\hat{\tau}_t - \tau^*)^2 + 2\eta_t\cdot\left(\tilde{l}_{1-\alpha}(\tau^*, \tilde{S}_t, \{S_{t,y}\}_{y=1}^{K}) - \tilde{l}_{1-\alpha}(\hat{\tau}_t, \tilde{S}_t, \{S_{t,y}\}_{y=1}^{K})\right) +$$
$$\eta_t^2\cdot\left(\nabla_{\hat{\tau}_t}\tilde{l}_{1-\alpha}(\hat{\tau}_t, \tilde{S}_t, \{S_{t,y}\}_{y=1}^{K})\right)^2. \tag{12}$$

Following from Lemma F.2, we have

$$\left(\nabla_{\hat{\tau}}\tilde{l}_{1-\alpha}(\hat{\tau}_t, \tilde{S}_t, \{S_{t,y}\}_{y=1}^{K})\right)^2 \le \left(\frac{1+\epsilon}{1-\epsilon}\right)^2.$$

Dividing this inequality by $\eta_t$ and summing over $t = 1, 2, \cdots, T$ provides

$$\sum_{t=1}^{T}\left(\tilde{l}_{1-\alpha}(\hat{\tau}_t, \tilde{S}_t, \{S_{t,y}\}_{y=1}^{K}) - \tilde{l}_{1-\alpha}(\tau^*, \tilde{S}_t, \{S_{t,y}\}_{y=1}^{K})\right)$$
$$\le \sum_{t=1}^{T}\left(\frac{1}{2\eta_t}(\hat{\tau}_t - \tau^*)^2 - \frac{1}{2\eta_t}(\hat{\tau}_{t+1} - \tau^*)^2\right) + \left(\frac{1+\epsilon}{1-\epsilon}\right)^2\cdot\sum_{t=1}^{T}\frac{\eta_t}{2}$$
$$= \frac{1}{2\eta_1}(\hat{\tau}_1 - \tau^*)^2 - \frac{1}{2\eta_{T+1}}(\hat{\tau}_{T+1} - \tau^*)^2 + \sum_{t=1}^{T-1}\left(\frac{1}{2\eta_{t+1}} - \frac{1}{2\eta_t}\right)(\hat{\tau}_t - \tau^*)^2 + \left(\frac{1+\epsilon}{1-\epsilon}\right)^2\cdot\sum_{t=1}^{T}\frac{\eta_t}{2}.$$

Lemma F.4 gives us that

$$(\hat{\tau}_t - \tau^*)^2 \le \left(1 + \max_{1\le t\le T-1}\eta_t\cdot\frac{1+\epsilon}{1-\epsilon}\right)^2,$$

which follows that

$$\sum_{t=1}^{T}\left(\tilde{l}_{1-\alpha}(\hat{\tau}_t, \tilde{S}_t, \{S_{t,y}\}_{y=1}^{K}) - \tilde{l}_{1-\alpha}(\tau^*, \tilde{S}_t, \{S_{t,y}\}_{y=1}^{K})\right)$$
$$\le \frac{1}{2\eta_1}\left(1 + \max_{1\le t\le T-1}\eta_t\cdot\frac{1+\epsilon}{1-\epsilon}\right)^2 + \left(\frac{1}{2\eta_T} - \frac{1}{2\eta_1}\right)\cdot\left(1 + \max_{1\le t\le T-1}\eta_t\cdot\frac{1+\epsilon}{1-\epsilon}\right)^2 + \left(\frac{1+\epsilon}{1-\epsilon}\right)^2\cdot\sum_{t=1}^{T}\frac{\eta_t}{2}$$
$$= \frac{1}{2\eta_T}\left(1 + \max_{1\le t\le T-1}\eta_t\cdot\frac{1+\epsilon}{1-\epsilon}\right)^2 + \left(\frac{1+\epsilon}{1-\epsilon}\right)^2\cdot\sum_{t=1}^{T}\frac{\eta_t}{2}.$$

□

### E.8 Proof for Proposition B.3

**Lemma E.10.** *Under Assumption B.2, the pinball loss satisfies*

$$|\hat{\tau} - \tau|^q \leq \frac{q(1-q)}{b} \left(\frac{1}{2\varepsilon_0}\right)^q \cdot E\left[l_{1-\alpha}(\hat{\tau}, S) - l_{1-\alpha}(\tau, S)\right]$$

*Proof.* We employ the proof technique from Lemma C.4 in [11]. The Assumption B.2 gives us that

$$\mathbb{P}\{R = \hat{\gamma}\} \geq b\,|\hat{\gamma} - \gamma|^{q-2} \iff \mathbb{P}\{S = \hat{\tau}\} \geq 2b\,|2(\hat{\tau} - \tau)|^{q-2}, \quad \hat{\gamma} = 2\hat{\tau} - 1$$

By Theorem 2.7 of [66], we have

$$|\hat{\gamma} - \gamma| \leq 2^{1-1/q} q^{1/q} \gamma^{-1/q} \cdot (E\left[l_{1-\alpha}(\hat{\gamma}, R) - l_{1-\alpha}(\gamma, R)\right])^{1/q} = 2\left(\frac{q(1-q)}{b}\left(\frac{1}{2\varepsilon_0}\right)^q \cdot E\left[l_{1-\alpha}(\hat{\gamma}, R) - l_{1-\alpha}(\gamma, R)\right]\right)^{1/q}$$

Since $|\hat{\gamma} - \gamma| = 2\,|\hat{\tau} - \tau|$ and $l_{1-\alpha}(\hat{\gamma}, R) = l_{1-\alpha}(\hat{\tau}, S)$, we can obtain

$$|\hat{\tau} - \tau|^q \leq \frac{q(1-q)}{b}\left(\frac{1}{2\varepsilon_0}\right)^q \cdot E\left[l_{1-\alpha}(\hat{\tau}, S) - l_{1-\alpha}(\tau, S)\right]$$

$\square$

**Proposition E.11** (Restatement of Proposition B.3)**.** *Consider online conformal prediction under uniform label noise with noise rate $\epsilon \in (0,1)$. Given Assumptions 2.1, 2.2 and B.2, when updating the threshold according to Eq. (6), for any $T \in \mathbb{N}^+$, we have:*

$$|\bar{\tau} - \tau|^q \leq \frac{q(1-q)}{b}\left(\frac{1}{2\varepsilon_0}\right)^q \cdot \left[\frac{(\hat{\tau}_1 - \tau^*)^2}{2\sum_{t=1}^T \eta_t} + \frac{\sum_{t=1}^T \eta_t^2}{\sum_{t=1}^T \eta_t} \cdot \left(\frac{1+\epsilon}{1-\epsilon}\right)^2\right]$$

*where $\bar{\tau} = \sum_{t=1}^T \eta_t \hat{\tau}_t / \sum_{t=1}^T \eta_t$.*

*Proof.* We begin our proof from Eq. (12):

$$(\hat{\tau}_{t+1} - \tau)^2 \leq (\hat{\tau}_t - \tau)^2 + 2\eta_t \cdot \left(\tilde{l}_{1-\alpha}(\tau, \tilde{S}_t, \{S_{t,y}\}_{y=1}^K) - \tilde{l}_{1-\alpha}(\hat{\tau}_t, \tilde{S}_t, \{S_{t,y}\}_{y=1}^K)\right) + \eta_t^2 \cdot \left(\nabla_{\hat{\tau}_t} \tilde{l}_{1-\alpha}(\hat{\tau}_t, \tilde{S}_t, \{S_{t,y}\}_{y=1}^K)\right)^2.$$

By taking expectation condition on $(X_t, Y_t)$ (or equivalently on $\tilde{S}_t$ and $\{S_{t,y}\}_{y=1}^K$), and applying Lemma F.2 and Proposition 4.1, we have

$$\mathbb{E}\left[(\hat{\tau}_{t+1} - \tau^*)^2\right] \leq (\hat{\tau}_t - \tau^*)^2 + 2\eta_t \cdot (\mathbb{E}\left[l_{1-\alpha}(\tau, S)\right] - \mathbb{E}\left[l_{1-\alpha}(\hat{\tau}_t, S)\right]) + \eta_t^2 \cdot \left(\frac{1+\epsilon}{1-\epsilon}\right)^2$$

By rearranging and taking expectation, we have

$$2\eta_t \cdot (\mathbb{E}\left[l_{1-\alpha}(\hat{\tau}_t, S) - l_{1-\alpha}(\tau, S)\right]) \leq \mathbb{E}\left[(\hat{\tau}_t - \tau^*)^2\right] - \mathbb{E}\left[(\hat{\tau}_{t+1} - \tau^*)^2\right] + \eta_t^2 \cdot \left(\frac{1+\epsilon}{1-\epsilon}\right)^2$$

Summing over $t = 1, 2, \cdots, T$ provides

$$2\sum_{t=1}^T \eta_t \cdot (\mathbb{E}\left[l_{1-\alpha}(\hat{\tau}_t, S) - l_{1-\alpha}(\tau, S)\right]) \leq (\hat{\tau}_1 - \tau^*)^2 - \mathbb{E}\left[(\hat{\tau}_{t+1} - \tau^*)^2\right] + \left(\frac{1+\epsilon}{1-\epsilon}\right)^2 \cdot \sum_{t=1}^T \eta_t^2$$

$$\leq (\hat{\tau}_1 - \tau^*)^2 + \left(\frac{1+\epsilon}{1-\epsilon}\right)^2 \cdot \sum_{t=1}^T \eta_t^2$$

Let us denote $\bar{\tau} := \sum_{t=1}^T \eta_t \hat{\tau}_t / \sum_{t=1}^T \eta_t$. Applying Jensen's inequality and dividing both sides by $2\sum_{t=1}^T \eta_t$ gives

$$\mathbb{E}\left[l_{1-\alpha}(\bar{\tau}, S) - l_{1-\alpha}(\tau, S)\right] \leq \sum_{t=1}^T \frac{\eta_t}{\sum_{t=1}^T \eta_t} \cdot (\mathbb{E}\left[l_{1-\alpha}(\hat{\tau}_t, S) - l_{1-\alpha}(\tau, S)\right])$$

$$\leq \frac{(\hat{\tau}_1 - \tau^*)^2}{2\sum_{t=1}^T \eta_t} + \frac{\sum_{t=1}^T \eta_t^2}{\sum_{t=1}^T \eta_t} \cdot \left(\frac{1+\epsilon}{1-\epsilon}\right)^2$$

Continuing from Lemma E.10, we can conclude that

$$|\bar{\tau} - \tau|^q \le \frac{q(1-q)}{b} \left(\frac{1}{2\varepsilon_0}\right)^q \cdot \left[\frac{(\hat{\tau}_1 - \tau^*)^2}{2\sum_{t=1}^T \eta_t} + \frac{\sum_{t=1}^T \eta_t^2}{\sum_{t=1}^T \eta_t} \cdot \left(\frac{1+\epsilon}{1-\epsilon}\right)^2\right]$$

$\square$

## F   Helpful lemmas

**Lemma F.1.** *The distribution of the true non-conformity score, noise non-conformity score, and scores of all classes satisfy the following relationship:*

$$(1)\mathbb{P}\{S = s\} = \frac{1}{1-\epsilon}\mathbb{P}\left\{\tilde{S} = s\right\} - \frac{\epsilon}{K(1-\epsilon)}\sum_{y=1}^K \mathbb{P}\{S_y = s\};$$

$$(2)\mathbb{P}\{S \le s\} = \frac{1}{1-\epsilon}\mathbb{P}\left\{\tilde{S} \le s\right\} - \frac{\epsilon}{K(1-\epsilon)}\sum_{y=1}^K \mathbb{P}\{S_y \le s\};$$

$$(3)\mathbb{P}\{S > s\} = \frac{1}{1-\epsilon}\mathbb{P}\left\{\tilde{S} > s\right\} - \frac{\epsilon}{K(1-\epsilon)}\sum_{y=1}^K \mathbb{P}\{S_y > s\},$$

*where $S$, $\tilde{S}$, and $S_y$ denote the true score, noisy score, and score for class $y$ respectively.*

*Proof.* **(1):**

$$\mathbb{P}\left\{\tilde{S} = s\right\} = \mathbb{P}\left\{\tilde{S} = s|\tilde{S} = S\right\} \cdot \mathbb{P}\left\{\tilde{S} = S\right\} + \mathbb{P}\left\{\tilde{S} = s|\tilde{S} = \bar{S}\right\} \cdot \mathbb{P}\left\{\tilde{S} = \bar{S}\right\}$$

$$= \mathbb{P}\{S = s\} \cdot \mathbb{P}\left\{\tilde{Y} = Y\right\} + \mathbb{P}\{\bar{S} = s\} \cdot \mathbb{P}\left\{\tilde{Y} = \bar{Y}\right\}$$

$$= (1 - \epsilon)\mathbb{P}\{S = s\} + \epsilon\mathbb{P}\{\bar{S} = s\},$$

which follows that

$$\mathbb{P}\{S = s\} = \frac{1}{1-\epsilon}\mathbb{P}\left\{\tilde{S} = s\right\} - \frac{\epsilon}{1-\epsilon}\mathbb{P}\{\bar{S} = s\} = \frac{1}{1-\epsilon}\mathbb{P}\left\{\tilde{S} = s\right\} - \frac{\epsilon}{K(1-\epsilon)}\sum_{y=1}^K \mathbb{P}\{S_y = s\}.$$

**(2):**

$$\mathbb{P}\left\{\tilde{S} \le s\right\} = \mathbb{P}\left\{\tilde{S} \le s|\tilde{S} = S\right\} \cdot \mathbb{P}\left\{\tilde{S} = S\right\} + \mathbb{P}\left\{\tilde{S} \le s|\tilde{S} = \bar{S}\right\} \cdot \mathbb{P}\left\{\tilde{S} = \bar{S}\right\}$$

$$= \mathbb{P}\{S \le s\} \cdot \mathbb{P}\left\{\tilde{Y} = Y\right\} + \mathbb{P}\{\bar{S} \le s\} \cdot \mathbb{P}\left\{\tilde{Y} = \bar{Y}\right\}$$

$$= (1 - \epsilon)\mathbb{P}\{S \le s\} + \epsilon\mathbb{P}\{\bar{S} \le s\},$$

which follows that

$$\mathbb{P}\{S \le s\} = \frac{1}{1-\epsilon}\mathbb{P}\left\{\tilde{S} \le s\right\} - \frac{\epsilon}{1-\epsilon}\mathbb{P}\{\bar{S} \le s\} = \frac{1}{1-\epsilon}\mathbb{P}\left\{\tilde{S} \le s\right\} - \frac{\epsilon}{K(1-\epsilon)}\sum_{y=1}^K \mathbb{P}\{S_y \le s\}.$$

**(3):**

$$\mathbb{P}\left\{\tilde{S} > s\right\} = \mathbb{P}\left\{\tilde{S} > s|\tilde{S} = S\right\} \cdot \mathbb{P}\left\{\tilde{S} = S\right\} + \mathbb{P}\left\{\tilde{S} > s|\tilde{S} = \bar{S}\right\} \cdot \mathbb{P}\left\{\tilde{S} = \bar{S}\right\}$$

$$= \mathbb{P}\{S > s\} \cdot \mathbb{P}\left\{\tilde{Y} = Y\right\} + \mathbb{P}\{\bar{S} > s\} \cdot \mathbb{P}\left\{\tilde{Y} = \bar{Y}\right\}$$

$$= (1 - \epsilon)\mathbb{P}\{S > s\} + \epsilon\mathbb{P}\{\bar{S} > s\},$$

which follows that

$$\mathbb{P}\{S > s\} = \frac{1}{1-\epsilon}\mathbb{P}\left\{\tilde{S} > s\right\} - \frac{\epsilon}{1-\epsilon}\mathbb{P}\{\bar{S} > s\} = \frac{1}{1-\epsilon}\mathbb{P}\left\{\tilde{S} > s\right\} - \frac{\epsilon}{K(1-\epsilon)}\sum_{y=1}^K \mathbb{P}\{S_y > s\}.$$

$\square$

**Lemma F.2.** *The gradient of robust pinball loss can be bounded as follows*

$$\alpha - 1 - \frac{\epsilon}{1 - \epsilon} \leq \nabla_{\hat{\tau}} \tilde{l}_{1-\alpha}(\hat{\tau}, \tilde{S}, \{S_y\}_{y=1}^K) \leq \alpha + \frac{\epsilon}{1 - \epsilon}. \tag{13}$$

*Proof.* Consider the gradient of robust pinball loss:

$$\nabla_{\hat{\tau}} \tilde{l}_{1-\alpha}(\hat{\tau}, \tilde{S}, \{S_y\}_{y=1}^K) = \underbrace{\frac{\alpha}{1 - \epsilon} \mathbb{1}\left\{\hat{\tau} \geq \tilde{S}\right\}}_{(a)} - \underbrace{\sum_{y=1}^K \frac{\alpha \epsilon}{K(1 - \epsilon)} \mathbb{1}\left\{\hat{\tau} \geq S_y\right\}}_{(b)} - \underbrace{\frac{1 - \alpha}{1 - \epsilon} \mathbb{1}\left\{\hat{\tau} < \tilde{S}\right\}}_{(c)} + \underbrace{\sum_{y=1}^K \frac{(1 - \alpha)\epsilon}{K(1 - \epsilon)} \mathbb{1}\left\{\hat{\tau} < S_y\right\}}_{(d)}.$$

Due to fact that $\mathbb{1}\{\cdot\} \in [0, 1]$, we can bound each part as follows:

Part (a):

$$\frac{\alpha}{1 - \epsilon} \mathbb{1}\left\{\hat{\tau} \geq \tilde{S}\right\} \in \left[0, \frac{\alpha}{1 - \epsilon}\right].$$

Part (b):

$$\sum_{y=1}^K \frac{\alpha \epsilon}{K(1 - \epsilon)} \mathbb{1}\left\{\hat{\tau} \geq S_y\right\} \in \left[0, \frac{\alpha \epsilon}{1 - \epsilon}\right].$$

Part (c):

$$\frac{1 - \alpha}{1 - \epsilon} \mathbb{1}\left\{\hat{\tau} < \tilde{s}\right\} \in \left[0, \frac{1 - \alpha}{1 - \epsilon}\right].$$

Part (d):

$$\sum_{y=1}^K \frac{(1 - \alpha)\epsilon}{K(1 - \epsilon)} \mathbb{1}\left\{\hat{\tau} < S_y\right\} \in \left[0, \frac{(1 - \alpha)\epsilon}{1 - \epsilon}\right].$$

Combining four parts, we can conclude that

$$\nabla_{\hat{\tau}} \tilde{l}_{1-\alpha}(\hat{\tau}, \tilde{S}, \{S_y\}_{y=1}^K) \leq \frac{\alpha}{1 - \epsilon} - 0 - 0 + \frac{(1 - \alpha)\epsilon}{1 - \epsilon} = \alpha + \frac{\epsilon}{1 - \epsilon};$$

$$\nabla_{\hat{\tau}} \tilde{l}_{1-\alpha}(\hat{\tau}, \tilde{S}, \{S_y\}_{y=1}^K) \geq 0 - \frac{\alpha \epsilon}{1 - \epsilon} - \frac{1 - \alpha}{1 - \epsilon} + 0 = \alpha - 1 - \frac{\epsilon}{1 - \epsilon}.$$

$\square$

**Lemma F.3.** *With at least probability $1 - \delta$, we have*

$$\left| \sum_{t=1}^T \mathbb{E}_{S_t} \left[ \nabla_{\hat{\tau}_t} l_{1-\alpha}(\hat{\tau}_t, S_t) \right] - \sum_{t=1}^T \nabla_{\hat{\tau}_t} l_{1-\alpha}(\hat{\tau}_t, S_t) \right| \leq \sqrt{2T \log\left(\frac{4}{\delta}\right)}.$$

*Proof.* Define

$$Y_T = \sum_{t=1}^T \mathbb{E}_{S_t} \left[ \nabla_{\hat{\tau}_t} l_{1-\alpha}(\hat{\tau}_t, S_t) \right] - \sum_{t=1}^T \nabla_{\hat{\tau}_t} l_{1-\alpha}(\hat{\tau}_t, S_t);$$

$$D_T = Y_T - Y_{T-1} = \mathbb{E}_S \left[ \nabla_{\hat{\tau}_t} l_{1-\alpha}(\hat{\tau}_t, S_t) \right] - \nabla_{\hat{\tau}_t} l_{1-\alpha}(\hat{\tau}_t, S_t).$$

Now, we will verify that $\{Y_T\}$ is a martingale, and $\{D_T\}$ is a bounded martingale difference sequence. Due to the definition of $\{Y_T\}$, we have

$$\mathbb{E}_{S_t}[Y_T | Y_{T-1}, \cdots, Y_t] = \mathbb{E}_{S_t}[D_T + Y_{T-1} | Y_{T-1}, \cdots, Y_t] = \mathbb{E}_{S_t}[D_T | Y_{T-1}, \cdots, Y_t] + Y_{T-1} = Y_{T-1},$$

where the last equality follows from the definition of $\{D_T\}$. In addition, we have

$$\mathbb{E}_{S_t} \left[ \nabla_{\hat{\tau}_t} l_{1-\alpha}(\hat{\tau}_t, S_t) \right] = \mathbb{P}\left\{S_t \leq \hat{\tau}_t\right\} - (1 - \alpha) \in [\alpha - 1, \alpha],$$

and Eq. (13) gives us that

$$D_T = \mathbb{E}_{S_t}\left[\nabla_{\hat{\tau}_t} l_{1-\alpha}(\hat{\tau}_t, S_t)\right] - \nabla_{\hat{\tau}_t} l_{1-\alpha}(\hat{\tau}_t, S_t) \in [-1, 1].$$

Therefore, by applying Azuma–Hoeffding's inequality, we can have

$$\mathbb{P}\left\{\left|\sum_{t=1}^{T} D_t\right| \leq t\right\} = \mathbb{P}\left\{\left|\sum_{t=1}^{T} \mathbb{E}_{S_t}\left[\nabla_{\hat{\tau}_t} l_{1-\alpha}(\hat{\tau}_t, S_t)\right] - \sum_{t=1}^{T} \nabla_{\hat{\tau}_t} l_{1-\alpha}(\hat{\tau}_t, S_t)\right| \leq r\right\} \geq 1 - 2\exp\left\{-\frac{r^2}{2T}\right\}.$$

Using $r = \sqrt{2T \log(4/\delta)}$, we have

$$\mathbb{P}\left\{\left|\sum_{t=1}^{T} \mathbb{E}_{S_t}\left[\nabla_{\hat{\tau}_t} l_{1-\alpha}(\hat{\tau}_t, S_t)\right] - \sum_{t=1}^{T} \nabla_{\hat{\tau}_t} l_{1-\alpha}(\hat{\tau}_t, S_t)\right| \leq \sqrt{2T \log\left(\frac{4}{\delta}\right)}\right\} \geq 1 - \frac{\delta}{2}.$$

$\square$

**Lemma F.4.** *Consider online conformal prediction under uniform label noise with noise rate $\epsilon \in (0, 1)$. Given Assumptions 2.1 and 2.2, when updating the threshold according to Eq. (6), for any $T \in \mathbb{N}^+$, we have*

$$-\max_{1 \leq t \leq T-1} \eta_t \cdot \left(\alpha + \frac{\epsilon}{1-\epsilon}\right) \leq \hat{\tau}_T \leq 1 + \max_{1 \leq t \leq T-1} \eta_t \cdot \left(\frac{1}{1-\epsilon} - \alpha\right) \tag{14}$$

*for all $T \in \mathbb{N}^+$.*

*Proof.* We prove this by induction. First, we know $\hat{\tau}_1 \in [0, 1]$ by assumption, which indicates that Eq. (14) is satisfied at $t = 1$. Then, we assume that Eq. (14) holds for $t = T$, and we will show that $\hat{\tau}_{T+1}$ lies in this range. Consider three cases:

**Case 1.** If $\hat{\tau}_T \in [0, 1]$, we have

$$\hat{\tau}_{T+1} = \hat{\tau}_T - \eta_T \cdot \nabla_{\hat{\tau}_T} \tilde{l}_{1-\alpha}(\hat{\tau}_T, \tilde{S}_T, \{S_{t,y}\}_{y=1}^K) \stackrel{(a)}{\in} \left[0 - \eta_T \cdot \left(\alpha + \frac{\epsilon}{1-\epsilon}\right), 1 - \eta_T \cdot \left(\alpha - 1 - \frac{\epsilon}{1-\epsilon}\right)\right]$$

$$\subseteq \left[-\max_{1 \leq t \leq T-1} \eta_t \cdot \left(\alpha + \frac{\epsilon}{1-\epsilon}\right), 1 + \max_{1 \leq t \leq T-1} \eta_t \cdot \left(\frac{1}{1-\epsilon} - \alpha\right)\right]$$

where (a) follows from Eq. (13).

**Case 2.** Consider the case where $\hat{\tau}_T \in [1, 1 + \max_{1 \leq t \leq T-1} \eta_t \cdot (1/(1-\epsilon) - \alpha)]$. The assumption that $\tilde{S}_T, S_{t,y} \in [0, 1]$ implies $\mathbb{1}\left\{\tilde{S}_T \leq \hat{\tau}_T\right\} = \mathbb{1}\left\{S_{t,y} \leq \hat{\tau}_T\right\} = 1$. Thus, we have

$$\nabla_{\hat{\tau}_T} \tilde{l}_{1-\alpha}(\hat{\tau}_T, \tilde{S}_T, \{S_{t,y}\}_{y=1}^K) = \frac{\alpha}{1-\epsilon} \mathbb{1}\left\{\hat{\tau}_T \geq \tilde{S}_T\right\} - \sum_{y=1}^{K} \frac{\alpha\epsilon}{K(1-\epsilon)} \mathbb{1}\left\{\hat{\tau}_T \geq S_{t,y}\right\} - \frac{1-\alpha}{1-\epsilon} \mathbb{1}\left\{\hat{\tau}_T \geq \tilde{S}_T\right\}$$

$$+ \sum_{y=1}^{K} \frac{(1-\alpha)\epsilon}{K(1-\epsilon)} \mathbb{1}\left\{\hat{\tau}_T \geq S_{t,y}\right\}$$

$$= \frac{\alpha}{1-\epsilon} - \sum_{y=1}^{K} \frac{\alpha\epsilon}{K(1-\epsilon)} = \alpha,$$

which follows that

$$\hat{\tau}_{T+1} = \hat{\tau}_T - \eta_T \cdot \nabla_{\hat{\tau}_T} \tilde{l}_{1-\alpha}(\hat{\tau}_T, \tilde{S}_T, \{S_{t,y}\}_{y=1}^K) = \hat{\tau}_T - \eta_T \alpha$$

$$\in \left[1 - \eta_T \alpha, 1 + \max_{1 \leq t \leq T-1} \eta_t \cdot \left(\frac{1}{1-\epsilon} - \alpha\right) - \frac{\eta_T}{1-\epsilon}\right]$$

$$\subseteq \left[-\max_{1 \leq t \leq T-1} \eta_t \cdot \left(\alpha + \frac{\epsilon}{1-\epsilon}\right), 1 + \max_{1 \leq t \leq T-1} \eta_t \cdot \left(\frac{1}{1-\epsilon} - \alpha\right)\right]$$

**Case 3.** Consider the case where $\hat{\tau}_T \in [-\max_{1 \le t \le T-1} \eta \cdot (\alpha + \epsilon/(1-\epsilon)), 0]$. The assumption that $\tilde{S}, S_y \in [0,1]$ implies $\mathbb{1}\left\{\tilde{S} \le \hat{\tau}_T\right\} = \mathbb{1}\left\{S_y \le \hat{\tau}_T\right\} = 0$. Thus, we have

$$\nabla_{\hat{\tau}_T} \tilde{l}_{1-\alpha}(\hat{\tau}_T, \tilde{S}_T, \{S_{t,y}\}_{y=1}^K) = \frac{\alpha}{1-\epsilon}\mathbb{1}\left\{\hat{\tau}_T \ge \tilde{S}_T\right\} - \sum_{y=1}^K \frac{\alpha\epsilon}{K(1-\epsilon)}\mathbb{1}\left\{\hat{\tau}_T \ge S_{t,y}\right\} - \frac{1-\alpha}{1-\epsilon}\mathbb{1}\left\{\hat{\tau}_T \ge \tilde{S}_T\right\}$$

$$+ \sum_{y=1}^K \frac{(1-\alpha)\epsilon}{K(1-\epsilon)}\mathbb{1}\left\{\hat{\tau}_T \ge S_{t,y}\right\}$$

$$= -\frac{1-\alpha}{1-\epsilon}\mathbb{1}\left\{\tau < \tilde{s}\right\} + \sum_{y=1}^K \frac{(1-\alpha)\epsilon}{K(1-\epsilon)} = \alpha - 1,$$

which follows that

$$\hat{\tau}_{T+1} = \hat{\tau}_T - \eta_T \cdot \nabla_{\hat{\tau}_T} \tilde{l}_{1-\alpha}(\hat{\tau}_T, \tilde{S}_T, \{S_{t,y}\}_{y=1}^K) = \hat{\tau}_T + \eta_T \cdot (1-\alpha)$$

$$\in \left[-\max_{1 \le t \le T-1} \eta_t \cdot \left(\alpha + \frac{\epsilon}{1-\epsilon}\right) + \eta_T \cdot (1-\alpha), 0 + \eta_T \cdot (1-\alpha)\right]$$

$$\subseteq \left[-\max_{1 \le t \le T-1} \eta_t \cdot \left(\alpha + \frac{\epsilon}{1-\epsilon}\right), 1 + \max_{1 \le t \le T-1} \eta_t \cdot \left(\frac{1}{1-\epsilon} - \alpha\right)\right]$$

Combining three cases, we can conclude that

$$-\max_{1 \le t \le T-1} \eta_t \cdot \left(\alpha + \frac{\epsilon}{1-\epsilon}\right) \le \hat{\tau}_T \le 1 + \max_{1 \le t \le T-1} \eta_t \cdot \left(\frac{1}{1-\epsilon} - \alpha\right)$$

$\square$

**Lemma F.5.** *With at least probability $1 - \delta$, we have*

$$\left|\sum_{t=1}^T \mathbb{E}_{\tilde{S}_t, S_{t,y}}\left[\nabla_{\hat{\tau}_t}\tilde{l}_{1-\alpha}(\hat{\tau}_t, \tilde{S}_t, \{S_{t,y}\}_{y=1}^K)\right] - \sum_{t=1}^T \nabla_{\hat{\tau}_t}\tilde{l}_{1-\alpha}(\hat{\tau}_t, \tilde{S}_t, \{S_{t,y}\}_{y=1}^K)\right| \le \frac{\sqrt{2T\log(2/\delta)}}{1-\epsilon}.$$

*Proof.* Define

$$Y_T = \sum_{t=1}^T \mathbb{E}_{\tilde{S}_t, S_{t,y}}\left[\nabla_{\hat{\tau}_t}\tilde{l}_{1-\alpha}(\hat{\tau}_t, \tilde{S}_t, \{S_{t,y}\}_{y=1}^K)\right] - \sum_{t=1}^T \nabla_{\hat{\tau}_t}\tilde{l}_{1-\alpha}(\hat{\tau}_t, \tilde{S}_t, \{S_{t,y}\}_{y=1}^K);$$

$$D_T = Y_T - Y_{T-1} = \mathbb{E}_{\tilde{S}_t, S_{t,y}}\left[\nabla_{\hat{\tau}_t}\tilde{l}_{1-\alpha}(\hat{\tau}_t, \tilde{S}_t, \{S_{t,y}\}_{y=1}^K)\right] - \nabla_{\hat{\tau}_t}\tilde{l}_{1-\alpha}(\hat{\tau}_t, \tilde{S}_t, \{S_{t,y}\}_{y=1}^K).$$

Now, we will verify that $\{Y_T\}$ is a martingale, and $\{D_T\}$ is a bounded martingale difference sequence. Due to the definition of $\{Y_T\}$, we have

$$\mathbb{E}_{\tilde{S}_t, S_{t,y}}[Y_T|Y_{T-1}, \cdots, Y_t] = \mathbb{E}_{\tilde{S}_t, S_{t,y}}[D_T + Y_{T-1}|Y_{T-1}, \cdots, Y_t] = \mathbb{E}_{\tilde{S}_t, S_{t,y}}[D_T|Y_{T-1}, \cdots, Y_t] + Y_{T-1} = Y_{T-1},$$

where the last equality follows from the definition of $\{D_T\}$. In addition, we have

$$\mathbb{E}_{\tilde{S}_t, S_{t,y}}\left[\nabla_{\hat{\tau}_t}\tilde{l}_{1-\alpha}(\hat{\tau}_t, \tilde{S}_t, \{S_{t,y}\}_{y=1}^K)\right] \in [\alpha - 1, \alpha],$$

and Lemma F.2 gives us that

$$D_T = \mathbb{E}_{\tilde{S}_t, S_{t,y}}[\nabla_{\hat{\tau}_t}\tilde{l}_{1-\alpha}(\hat{\tau}_t, \tilde{S}_t, \{S_{t,y}\}_{y=1}^K)] - \nabla_{\hat{\tau}_t}\tilde{l}_{1-\alpha}(\hat{\tau}_t, \tilde{S}_t, \{S_{t,y}\}_{y=1}^K) \in \left[-\frac{1}{1-\epsilon}, \frac{1}{1-\epsilon}\right].$$

Therefore, by applying Azuma–Hoeffding's inequality, we can have

$$\mathbb{P}\left\{\left|\sum_{t=1}^T D_t\right| \le t\right\} = \mathbb{P}\left\{\left|\sum_{t=1}^T \mathbb{E}_{\tilde{S}_t, S_{t,y}}[\nabla_{\hat{\tau}_t}\tilde{l}_{1-\alpha}(\hat{\tau}_t, \tilde{S}_t, \{S_{t,y}\}_{y=1}^K)] - \sum_{t=1}^T \nabla_{\hat{\tau}_t}\tilde{l}_{1-\alpha}(\hat{\tau}_t, \tilde{S}_t, \{S_{t,y}\}_{y=1}^K)\right| \le r\right\}$$

$$\ge 1 - 2\exp\left\{-\frac{[r(1-\epsilon)]^2}{2T}\right\}.$$

Using $r = [2T \log(2/\delta)]^{-1/2} / (1 - \epsilon)$, we have

$$\mathbb{P} \left\{ \left| \sum_{t=1}^{T} \mathbb{E}_{\tilde{S}_t, S_{t,y}} [\nabla_{\hat{\tau}_t} \tilde{l}_{1-\alpha}(\hat{\tau}_t, \tilde{S}_t, \{S_{t,y}\}_{y=1}^{K})] - \sum_{t=1}^{T} \nabla_{\hat{\tau}_t} \tilde{l}_{1-\alpha}(\hat{\tau}_t, \tilde{S}_t, \{S_{t,y}\}_{y=1}^{K}) \right| \leq \frac{\sqrt{2T \log(2/\delta)}}{1 - \epsilon} \right\} \geq 1 - \delta.$$

$\square$

