# OpenReview forum: "Exploring the Noise Robustness of Online Conformal Prediction"
_NeurIPS.cc/2025/Conference — NeurIPS 2025 poster_

### Official Review · Reviewer_SDzf · 2025-06-15

**Clarity:** 3
**Significance:** 2
**Originality:** 3
**Rating:** 4
**Confidence:** 5

**Summary:**

This paper proposes a new online conformal prediction method designed to be robust against uniform label noise, based on minimizing the robust pinball loss. The authors provide convergence analysis under both constant and dynamic learning rates. Extensive experiments demonstrate the effectiveness of the proposed approach, particularly when the noise rate is known.

**Questions:**

Aside from the weaknees, I have two minor questions.

Q1: In Figure 2 (C), the method based on clean data is undercover, can the author explain this?

Q2: In Proposition 3.1., the result seems counterintuitive: if the prediction set $C_t(X_t)$ for each $t$ is infinite, then the left term should be $\alpha$ as the miscoverage is $0$, and the right term is $-\infty$ since the prediction size is infinity. Then it leads to $\alpha\leq-\infty$, which is strange.

**Ethical Concerns:**

["NO or VERY MINOR ethics concerns only"]

**Final Justification:**

I raise my score from 3 to 4 as the author provides the practical implementation of their approach by using an estimated noise rate.

**Limitations:**

yes

**Paper Formatting Concerns:**

No formatting issue.

**Quality:**

2

**Strengths And Weaknesses:**

Strength: The paper introduces a novel approach for online conformal prediction under label noise, with rigorous theoretical analysis supporting the proposed method.

Weakness 1: The primary limitation of the paper is its reliance on the assumption that the noise rate $\epsilon$ is known. In practice, this information is rarely available. The authors should at least propose a method for estimating the noise rate and provide empirical results using the estimated values. Without this, the practical applicability of the method is significantly limited.

Weakness 2: The main idea is highly specific to the noise mechanism to make it possible for the computation of the distribution of score function. This makes the proposed approach restricted to the uniform label noise setting, lacking generalization to other mechanism.

---

> ### Author Rebuttal · Authors · 2025-07-31
>
> We sincerely appreciate the reviewer’s constructive suggestions and address the raised issues below. (**W** for weakness, **Q** for question)
>
> > Assumption on a known noise rate [W1]
>
> Thank you for highlighting the concern. In the literature of label-noise learning, many methods [1,2] are proposed to leverage a small validation set of clean examples for estimating the noise ratio. Additionally, some approaches [3,4] estimate the noise rate without requiring clean labels during training. Such techniques make it reasonable to assume a known noise rate, which is widely recognized in the literature of label-noise learning [5-9].
>
> Moreover, we clarify that our method improves the performance even when the noise rate is not precisely estimated, as validated in Appendix C. To fully address this concern, we conduct an additional experiment where the noise rate is unknown and needs to be estimated. In particular, we apply an existing algorithm [3] to estimate the noise rate without access to clean data during the training of the pre-trained model. Then, we employ this estimated noise rate in our method for online conformal prediction. The experiments are conducted on CIFAR-10 dataset, using ResNet18 model with noise rate $\epsilon\in\{0.1,0.2,0.3\}$. We employ LAC score to generate prediction sets and use a constant learning rate $\eta=0.05$. The results are shown in the table below:
> | Noise rate | Estimated Noise rate | Method   | CovGap (%) | Size |
> |------------|----------------------|----------|------------|------|
> | $\epsilon=0.1$ | $\hat{\epsilon}=0.09$     | Baseline | 6.99       | 2.47 |
> |                |                     | Ours     | 0.82       | 0.94 |
> | $\epsilon=0.2$ |  $\hat{\epsilon}=0.23$  | Baseline | 8.72       | 5.41 |
> |                |                     | Ours     | 1.67       | 0.98 |
> | $\epsilon=0.3$ |  $\hat{\epsilon}=0.26$  | Baseline | 9.04       | 6.72 |
> |                |                     | Ours     | 2.32       | 1.15 |
>
> The results demonstrate that our method consistently achieves much lower coverage gap and smaller prediction sets compared to the baseline. This highlights the robustness and applicability of our method even when the noise rate is not precisely estimated.
>
> > Lack generalization to other noise mechanisms [W2]
>
> Thank you for highlighting this important point. We would like to emphasize that label noise is still a novel and underexplored challenge in conformal prediction, particularly in the online setting. As noted in Appendix A, the most relevant prior work [10] focuses solely on uniform noise in conformal prediction, only analyzing the overcoverage issue in online settings. In contrast, our work stands out as the first to specifically tackle label noise in online conformal prediction. While the current method is specific to uniform noise, it opens the door for extensions to more general noise mechanisms such as class-conditional label noise. Designing such extensions is a promising and important direction for future research. We hope this clarifies the significance of our work as a starting point in the field of noise-robust online conformal prediction.
>
> > SAOCP on clean data is undercover [Q1]
>
> Thank you for pointing this out. The under-coverage observed in Figure 2 $($C$)$ is a characteristic of the SAOCP method. A similar phenomenon is also noted in Tables 1 and 2 of the SAOCP manuscript [6], where the final coverage is around 0.88, which is slightly lower than the target coverage of 0.9.
>
> > The counterintuitive results in Proposition 3.1 [Q2]
>
> Thank you for the insightful question. In this work, we focus on online conformal prediction in multi-class classification (with K classes), where the maximum set size is the class number: $|\mathcal{C}_t(X_t)| \leq K$. In this online setting, the set size cannot always remain the maximum value $K$ for each time step $t$. This is because the threshold will be updated and decreased by online gradient descent at the next time step $t+1$, if the prediction set contains the true label at the time step $t$ (See Eq.(2)). Consequently, the prediction set size will be decreased until the true label is excluded. Therefore, the set size cannot always be infinite or $K$ for each t in online conformal prediction, so it would not lead to the counterintuitive results.
>
> ### References
> [1] Liu T, et al. Classification with noisy labels by importance reweighting. TPAMI 2015.
>
> [2] Yu X, et al. An efficient and provable approach for mixture proportion estimation using linear independence assumption. CVPR 2018.
>
> [3] Li X, et al. Provably end-to-end label-noise learning without anchor points. ICML 2021.
>
> [4] Xia X, et al. Are anchor points really indispensable in label-noise learning? NIPS 2019.
>
> [5] Han B, et al. Co-teaching: Robust training of deep neural networks with extremely noisy labels. NIPS 2018
>
> [6] Wei H, et al. Combating noisy labels by agreement: A joint training method with co-regularization. CVPR 2020.
>
> [7] Yu X, et al. How does disagreement help generalization against label corruption? ICML 2019.
>
> [8] Xia X, et al. Sample selection with uncertainty of losses for learning with noisy labels[J]. arXiv preprint 2021.
>
> [9] Xia X, et al. Combating noisy labels with sample selection by mining high-discrepancy examples. CVPR 2023.
>
> [10] Einbinder B S, et al. Label noise robustness of conformal prediction. JMLR 2024.
>
> [11] Bhatnagar A, et al. Improved online conformal prediction via strongly adaptive online learning. ICML 2023.

---

> > ### Comment · Reviewer_SDzf · 2025-08-02
> >
> > Thank you for the response, which addresses some of my concerns. I would be inclined to raise my score if the authors could include more thorough experiments using the estimated noise rate in a future revision. For example, the authors could treat the proposed approach with estimated noise as a new benchmark and report its results in all main figures and tables, as only estimated noise is available in real applications.

---

> ### Author Response · Authors · 2025-08-02
>
> Thank you for your prompt feedback and valuable suggestion.  In the final version, we will include the performance of the proposed method with estimated noise rates in all main figures and tables. We believe this would enhance the significance and contribution of this work by highlighting its practical applicability.

---

### Official Review · Reviewer_E9Fq · 2025-06-30

**Clarity:** 3
**Significance:** 3
**Originality:** 3
**Rating:** 5
**Confidence:** 4

**Summary:**

This paper studies online conformal prediction under uniform label noise. In online conformal prediction, given data sampled from a joint distribution that arrives in a sequential order, the goal is to compute a prediction set for each step such that the average coverage rate reaches the desired level. In the uniform label noise setting, with a fixed probability epsilon, the label of a data point is replaced by a label sampled uniformly at random from the label space.

In this paper, they first show that under uniform label noise, the vanilla adaptive conformal inference (ACI) method has a coverage gap, which is not necessarily diminishing. This shows that the ACI may have over-coverage or under-coverage depending on the expected size of the prediction sets.

They then propose a modified ACI with robust pinball loss. They show that this new conformal method achieves a diminishing expected coverage error and empirical coverage error with convergence rate O(T^{-1/2}) in both constant and dynamic learning rate settings.

Moreover, for dynamic learning rate, by choosing a decaying learning rate, their method can achieve an O(T^{1/2}) regret bound.
They evaluate their method on CIFAR-10 and ImageNet datasets and show that their method achieves the desired coverage with diminishing coverage gap while the baseline method has a coverage gap.

**Questions:**

1. A small question about assumptions 2.1 and 2.2: how do the results depend on these assumptions? Would a bounded score and threshold be enough?

2. Is there any connection between proposition 3.1 and the results for conservative coverage in previous work by Einbinder et al, 2024? For example, can we observe their results from the expected size of the prediction sets by adding the same distributional assumptions here?

3. Does Proposition 4.1 fail to hold under other label noise settings? If so, is this the primary limitation preventing the extension of the robust pinball loss to more general forms of label noise?

**Ethical Concerns:**

["NO or VERY MINOR ethics concerns only"]

**Final Justification:**

After reviewing the rebuttal, I would like to maintain my score and express my support for accepting this paper. As other reviewers have pointed out, the paper relies on the assumption that the noise rate is known. While I agree that this assumption is strong, I believe the paper provides valuable insights and may inspire future work that explores more general noise settings.

**Limitations:**

Yes, the authors mention that the current method is limited to uniform label noise with a known noise rate. They also mention the extension to other noise models with fewer assumptions as one future direction.

**Quality:**

3

**Strengths And Weaknesses:**

Strengths:

1.	This paper shows that the vanilla ACI method may provide conservative prediction sets with over-coverage under the uniform label noise. Their bound relates the coverage gap to the expected size of the prediction sets. Although previous work by Einbinder, Feldman, Bates, Angelopoulos, Gendler, and Romano (2024) observes that ACI obtains conservative prediction sets, they require distributional assumptions that the noisy score distribution stochastically dominates the clean score distribution. While this paper does not require distributional assumptions.

2.	They develop a new conformal method by using robust pinball loss and show that this method has a diminishing coverage gap with rate O(T^{-1/2}). They show that in expectation, the robust pinball loss under label noise is the same as the pinball loss for clean data and also has the same expected gradient as the pinball loss.

3.	Their experiments show that the proposed method achieves the desired coverage with diminishing coverage gap on real-world datasets, which validates their theoretical results.

Overall, I think this paper provides very interesting observations and new conformal methods for online conformal prediction under uniform label noise.

---

> ### Author Rebuttal · Authors · 2025-07-31
>
> Thank you for the insightful review. We provide point-by-point responses to these questions. (**W** for weakness, **Q** for question)
>
> > Assumptions in Section 2 [Q1]
>
> Thank you for the question. A bounded score and threshold are sufficient to achieve the desired coverage guarantees. The specific upper bound in these assumptions primarily impacts the constant in the coverage rate, but does not fundamentally alter the coverage guarantee of the method.
>
> > Connection between proposition 3.1 and the results by Einbinder et al, 2024 [Q2]
>
> Thank you for the insightful question. There is indeed a connection between Proposition 3.1 and the results for conservative coverage in [1]. In particular, Corollary 6 in [1] relies on the assumption that $\mathcal{C}_{noisy}$ contains the most likely (correct) labels. In this case, the model is effectively accurate, and the prediction sets remain small. Our Proposition 3.1 shows that when the prediction sets are small, label noise can result in over-coverage of prediction sets
> $$\frac{1}{T}\sum\_{t=1}^T1\\{Y_t\notin\mathcal{C}(X_t)\\}\leq\alpha$$
> which aligns with the conclusion from [1].
>
> > Limitation of Proposition 4.1 [Q3]
>
> Thank you for your thoughtful question. Proposition 4.1 is derived under the assumption of a uniform label noise, and it may not hold in more general label noise settings. This is indeed a primary limitation in extending the robust pinball loss to handle more complex label noise. We believe that extending our framework to handle more general label noise, such as noise transition matrix, is a promising avenue for future research.
>
> ### References
> [1] Einbinder B S, et al. Label noise robustness of conformal prediction. JMLR 2024.

---

> > ### Comment · Reviewer_E9Fq · 2025-08-07
> >
> > Thank you to the authors for their detailed response and to the other reviewers for their insightful comments. The authors have addressed all of my questions, and I would like to maintain my score.

---

> > > ### Author Response · Authors · 2025-08-08
> > >
> > > Great thanks for your recognition and support. We greatly appreciate your valuable feedback, which will significantly enhance the quality of this work.

---

### Official Review · Reviewer_ZVis · 2025-07-02

**Clarity:** 3
**Significance:** 1
**Originality:** 3
**Rating:** 4
**Confidence:** 4

**Summary:**

This paper looks at the online conformal prediction problem.  Arbitrary points $(X_t, Y_t) \in \mathcal{X} \times [K]$ arrive one by one, and the goal of the algorithm is, upon seeing $X_t$ to provide a confidence set $C_t \subseteq [K]$ for $Y_t$, such that the sequence of confidence sets achieves coverage $1 - \alpha$ on average over the whole sequence, i.e.
$$ \lim_{T \rightarrow \infty} \frac{1}{T}\sum_{i = 1}^T \mathbb{P} [ Y_t \in C_t ] \rightarrow 1- \alpha. $$
Once the algorithm outputs $C_t$, the true $Y_t$ is revealed to the algorithm.

They study this problem under the model of uniform label noise.  That is, instead of seeing $Y_t$, the algorithm is shown $\tilde{Y}_t$, where $\tilde{Y}_t$ is $Y_t$ with probability $1 - \varepsilon$ and $\tilde{Y} \sim \text{Unif}[K]$ with probability $\varepsilon$, for a known value of $\varepsilon$.  The goal is still to provide coverage with respect to the original $Y_t$, rather than the observed $\tilde{Y}_t$.

They show that existing online conformal prediction methods do not meet this guarantee. Existing online conformal methods work by estimating the top $1 - \alpha$ quantile of the score function, and outputting the interval from 0 to this quantile.  However, the quantiles of the scores computed from the observed $\tilde{Y}_t$ are not the same as the quantiles of the scores computed from the true $Y_t$, and therefore will achieve coverage over the $\tilde{Y}_t$ and not necessarily the $Y_t$s.
  To tackle this problem, they design a new loss function called the robust pinball loss, that allows them to estimate quantiles of the data robust to the known noise distribution.  This allows them to estimate the quantile of the scores if they had been computed according to the $Y_t$s.

**Questions:**

- Line 84: Are the datapoints $(X_t, Y_t)$ drawn i.i.d., or is the whole sequence drawn from a joint distribution?
- Line 108: Why is the correct measure to consider the absolute value of the expected deviation rather than the expectation of the absolute deviation?  That is, why is
$$\text{ExErr}(T) = \left| \frac{1}{T} \mathbb{P} [Y_t \notin C_t(X_t) ] - \alpha \right|$$
rather than
$$\text{ExErr}(T) = \frac{1}{T} \left|\mathbb{P} [Y_t \notin C_t(X_t) ] - \alpha \right|$$?

- Line 177: "This assumption is practical as the noise rate can be estimated from historical data": is this true when you are never given access to the $Y_t$?  Or does this assume that for the historical data you have access to the true $Y_t$?  If this can be estimated without access to the true $Y_t$, is there a way to do this on the fly on the prefix of the stream, and then use this value of $\varepsilon$ on the suffix of the stream?

**Ethical Concerns:**

["NO or VERY MINOR ethics concerns only"]

**Final Justification:**

I appreciate the authors thoughtful and thorough responses to my concerns.  I am satisfied with their justification of the known error rate, and the model where coverage is measure with respect to the unseen $Y$ values.  Thus I raise my score to 4: weak accept.

The main thing standing in the way of my raising the score beyond this is that I am still not convinced that these ideas can be extended beyond this very stylized error model.  For example, if the noise is not uniform, I believe a similar strategy would require a method to raise the threshold so much that the prediction sets would be very large and uninformative.  I see this as a fundamental drawback, as in practice, we don't expect noise to be drawn from a known distribution that is independent of the data.

**Limitations:**

Yes

**Quality:**

2

**Strengths And Weaknesses:**

**Strengths** \
It is nice that the formulation does not impose any structure on the score function.  This gives an elegant way to push forward the error from the label noise to the error in the score function.

**Weaknesses**
I think this model requires more justification.  Why is it a good objective to consider error of predicting the $Y_t$ rather than the $\tilde{Y}_t$?

I would interpret the online model as being one where the $\tilde{Y}_t$ are thought of as being both the training examples and the test examples.  In that interpretation, it makes sense the coverage of a predictor should also be measured against the $\tilde{Y}_t$ (test distribution) as well, which standard online conformal prediction would do out of the box.  Why does it make sense to instead optimize the coverage over the $Y_t$?  Can this coverage can even be estimated from samples? If the trained regression model is not very accurate, is it possible to know whether the uncertainty arises from the regression model being a bad fit, or whether it is due to noise in the labels? (See the example given below.)  Could you provide an example where it would be useful to have coverage against the unknown $Y_t$, which are never revealed to you?

The assumption that the noise distribution is known is very strong, since the motivation of conformal prediction (and especially online conformal prediction) to design distribution-free methods.  I appreciate that the authors acknowledge this in their conclusion, and hope that this result is a first step toward removing this strong assumption.  However, I am not convinced that this is actually possible.

The most general form of such a result might be to achieve this in the setting of distributionally robust optimization: where the noise $\tilde{Y}$ can be drawn from an arbitrary unknown distribution.   For this (strong) model of noise robustness, I believe that you can construct the following simple lower bound.  Suppose you are seeing $X_t = 0$ which are all the same, and $\tilde{Y}_t$ that are $\tilde{Y}_t = 1$ with probability $1 - \varepsilon$, and $\tilde{Y}_t \sim \text{Unif}[K]$ with probability $\varepsilon$.  Then from these samples, it is impossible to know whether it is the case that
- $Y_t = 1$ deterministically, and the label noise is uniform
- OR $Y_t = 1$ w.p. $1 - \frac{\varepsilon}{1 - \varepsilon}$, $Y_t \sim \text{Unif}[K]$ w.p. $\frac{\varepsilon}{1 - \varepsilon}$, and the noise is $\tilde{Y} = 1$ deterministically.
Suppose the noise level satisfies $\alpha/K < \varepsilon < \alpha$.  Then in the first case, there exists a a conformal set of size 1, containing just the label 1, that captures $Y_t = 1$ with probability $\ge 1 - \alpha$.  But in the second case, we require a conformal set of bigger size to capture $1 - \alpha$ of $Y_t$.  This suggests to me that it is not possible to design such a method for unknown noise distributions.
If this is true, then this means that such a model is restricted to providing results for settings where the noise distribution is known, which seems like a very strong assumption.  Is there a more benign class of noise distributions that you hope can be targeted?

Because of this modeling issue, I have given the submission a low score in significance.  Please let me know if there is some modeling consideration that I have not considered or misunderstood.

Please see "questions" for more detailed technical questions, mostly related to the modeling of this problem.

---

> ### Author Rebuttal · Authors · 2025-07-31
>
> We sincerely thank the reviewer for the constructive feedback. We reply to these concerns point by point. (**W** for weakness, **Q** for question)
>
> > Justification for optimizing the coverage over clean labels $Y_t$ [W1]
>
> Thank you for this insightful comment. The issue of label noise has been a common challenge in machine learning research, including conformal prediction [1–7] and online learning [8,9]. In conformal prediction, a line of work [1–7] aim to provide coverage guarantee with respect to clean labels: $\mathbb{P}\\{Y\in\mathcal{C}\_{noisy}(X)\\}\geq1-\alpha$, such as Corollary 6 in [2], Theorem 2 in [3], and Theorem 3.5 in [6]. Similarly, in online learning, prior work [8,9] investigate the setting where the learner observes noisy labels but still seeks to minimize the error with respect to the true labels, providing upper bounds on: $\mathbb{E}_{(X,Y)}[1\\{f(X)\neq Y\\}]$, such as Theorem 3 of [8] and Theorem 1 of [9]. In line with these efforts, our work is designed to provide coverage guarantees for the true labels, rather than the observed noisy labels.
>
> To demonstrate the practical necessity for this purpose, let us consider an online fraud detection system that classifies transactions as either fraudulent or legitimate. Due to human oversight or ambiguity in spending behavior, the training labels may occasionally be incorrect, leading to label noise, for instance, legitimate transactions mislabeled as fraudulent. If a conformal predictor guarantees coverage only against these noisy observed labels $\tilde{Y}_t$, then its prediction sets may fail to cover the true transaction status $Y_t$. This could harm the reliability of the system, resulting in unnecessary account freezes or missed fraud. We hope this clarifies the practical significance of providing guarantee for true labels.
>
> > Assumption of knowing the noise distribution [W2, Q3]
>
> Thank you for this thoughtful comment. In this work, we focus on the setting of uniform label noise as a first step towards noise robustness of online conformal prediction. Then, our assumption is that the noise rate can be estimated from the historical data in Section 2, which is widely recognized in the literature of label-noise learning [10-14]. In particular, some methods [15,16] can estimate the noise rate with a small subset of clean label data, while others [17,18] conduct this estimation directly during training without access to clean labels. These techniques complement our method, making it reasonable to assume a known noise rate.
>
> Moreover, we clarify that our method improves the performance even when the noise rate is not precisely estimated, as validated in Appendix C. To fully address this concern, we conduct an additional experiment where the noise rate is unknown and needs to be estimated. In particular, we apply an existing algorithm [17] to estimate the noise rate without access to clean data during the training of the pre-trained model. Then, we employ this estimated noise rate in our method for online conformal prediction. The experiments are conducted on CIFAR-10 dataset, using ResNet18 model with noise rate $\epsilon\in\{0.1,0.2,0.3\}$. We employ LAC score to generate prediction sets and use a constant learning rate $\eta=0.05$. The results are shown in the table below:
> | Noise rate | Estimated Noise rate | Method   | CovGap (%) | Size |
> |------------|----------------------|----------|------------|------|
> | $\epsilon=0.1$ | $\hat{\epsilon}=0.09$     | Baseline | 6.99       | 2.47 |
> |                |                     | Ours     | 0.82       | 0.94 |
> | $\epsilon=0.2$ |  $\hat{\epsilon}=0.23$  | Baseline | 8.72       | 5.41 |
> |                |                     | Ours     | 1.67       | 0.98 |
> | $\epsilon=0.3$ |  $\hat{\epsilon}=0.26$  | Baseline | 9.04       | 6.72 |
> |                |                     | Ours     | 2.32       | 1.15 |
>
> The results demonstrate that our method consistently achieves much lower coverage gap and smaller prediction sets compared to the baseline. This highlights the robustness and applicability of our method even when the noise rate is not precisely estimated.
>
> > The distributional assumption of data points [Q1]
>
> Following the framework of online conformal prediction [19-21], we do not assume that the data samples are i.i.d. sampled. Therefore, the data points can be drawn from an arbitrary distribution.
>
> > Measure of the expected deviation [Q2]
>
> Thank you for raising the question. This work aligns with prior research in online conformal prediction, which provides theoretical guarantees with respect to this metric—for example, in the discussion following Proposition 4.1 in [19], Theorem 6 in [20], and Theorem C.3 in [21].
>
> ### References
> [1] Einbinder B S, et al. Conformal prediction is robust to label noise. arXiv preprint 2022.
>
> [2] Einbinder B S, et al. Label noise robustness of conformal prediction. JMLR 2024.
>
> [3] Sesia M, et al. Adaptive conformal classification with noisy labels. JRSSB 2024.
>
> [4] Penso C, et al. Noise-robust conformal prediction for medical image classification. International Workshop on Machine Learning in Medical Imaging. 2024.
>
> [5] Feldman S, et al. Robust conformal prediction using privileged information. NIPS 2024.
>
> [6] Penso C, et al. Estimating the Conformal Prediction Threshold from Noisy Labels. arXiv preprint 2025.
>
> [7] Bortolotti T, et al. Noise-Adaptive Conformal Classification with Marginal Coverage. arXiv preprint 2025.
>
> [8] Nagarajan Nataraja, et al. Learning with noisy labels. NIPS 2013.
>
> [9] Changlong Wu, et al. Information-theoretic limits of online classification with noisy labels. NIPS 2024.
>
> [10] Han B, et al. Co-teaching: Robust training of deep neural networks with extremely noisy labels. NIPS 2018
>
> [11] Wei H, et al. Combating noisy labels by agreement: A joint training method with co-regularization. CVPR 2020.
>
> [12] Yu X, et al. How does disagreement help generalization against label corruption? ICML 2019.
>
> [13] Xia X, et al. Sample selection with uncertainty of losses for learning with noisy labels[J]. arXiv preprint 2021.
>
> [14] Xia X, et al. Combating noisy labels with sample selection by mining high-discrepancy examples. CVPR 2023.
>
> [15] Liu T, et al. Classification with noisy labels by importance reweighting. TPAMI 2015.
>
> [16] Yu X, et al. An efficient and provable approach for mixture proportion estimation using linear independence assumption. CVPR 2018.
>
> [17] Li X, et al. Provably end-to-end label-noise learning without anchor points. ICML 2021.
>
> [18] Xia X, et al. Are anchor points really indispensable in label-noise learning? NIPS 2019.
>
> [19] Gibbs I, et al. Adaptive conformal inference under distribution shift. NIPS 2021.
>
> [20] Gibbs I, et al. Conformal inference for online prediction with arbitrary distribution shifts. JMLR 2024.
>
> [21] Bhatnagar A, et al. Improved online conformal prediction via strongly adaptive online learning. ICML 2023.

---

> > ### Comment · Reviewer_ZVis · 2025-08-06
> >
> > I appreciate the authors thoughtful and thorough responses to my concerns. I am satisfied with their justification of the known error rate, and the model where coverage is measure with respect to the unseen
> >  values. Thus I raise my score to 4: weak accept.
> >
> > The main thing standing in the way of my raising the score beyond this is that I am still not convinced that these ideas can be extended beyond this very stylized error model. For example, if the noise is not uniform, I believe a similar strategy would require a method to raise the threshold so much that the prediction sets would be very large and uninformative. I see this as a fundamental drawback, as in practice, we don't expect noise to be drawn from a known distribution that is independent of the data.

---

> > > ### Author Response · Authors · 2025-08-08
> > >
> > > Thank you for raising the score. We are glad that our response has addressed some of your concerns about this work. We acknowledge the limitation that naively applying our method to more complex noise models may result in large and uninformative prediction sets. Despite this simplified setting, our work represents the first effort to tackle label noise in online conformal prediction. We view extending this approach to more complex settings as a promising future direction. We sincerely appreciate your time and effort in reviewing our work.

---

### Official Review · Reviewer_v3zM · 2025-07-02

**Clarity:** 2
**Significance:** 2
**Originality:** 2
**Rating:** 3
**Confidence:** 4

**Summary:**

This paper studies the robustness of online conformal prediction methods in the presence
of uniform label noise. The authors demonstrate that standard online conformal
algorithms, specifically ACI and its variants, can exhibit systematic coverage distortion
when labels are noisy, resulting from biased gradient updates due to corrupted supervision.
To address this, the authors propose a robust pinball loss, designed to be an
unbiased estimator of the true clean pinball loss. Theoretical results show that this loss
eliminates the bias and leads to convergence of the coverage error at rate $\mathcal{O}(T^{-1/2})$
The method is tested on CIFAR-100 and ImageNet under various noise level, with both
constant and dynamic learning rates, and across several conformity score functions.

**Questions:**

1. How much sample size is needed before the expectation-based correction becomes
accurate in practice?
2. How sensitive is performance to moderate misspecification in $\epsilon$? Could it be estimated
adaptively online?
3. Would the approach generalize to class-conditional or instance-dependent label
noise?

**Ethical Concerns:**

["NO or VERY MINOR ethics concerns only"]

**Limitations:**

See above weakenesses

**Quality:**

2

**Strengths And Weaknesses:**

Strengths:

1. The authors provide clear bounds showing how label noise affects coverage (Proposition 3.1), and they construct a principled loss correction (robust pinball loss) with matching expectations and gradients (Proposition 4.1).

2. The results span multiple datasets (CIFAR-100, ImageNet), multiple online conformal methods (ACI, SAOCP), and several conformity scores. The performance gains are consistent.

3. The robust pinball loss integrates cleanly with existing online conformal frameworks and doesn’t require major algorithmic changes.

Weaknesses:

1. The paper introduces the noise robustness issue briefly, without thoroughly justifying why this is a practical concern. Adding real-world scenarios where label noise impacts conformal prediction would help clarify the significance.

2. The method assumes that the noise rate $\epsilon$ is known or at least accurately estimated.
However, there is limited discussion of how sensitive the method is to errors in $\epsilon$
and how these errors affect the theoretical guarantees.

3. The robust methold maintains an $O\left(T^{-1 / 2}\right)$ convergence rate even when $\epsilon=0$, which is worse than the $O\left(T^{-1}\right)$ rate available to the standard online conformal method.

4. A major advantage of conformal methods is their finite-sample, distribution-free guarantee.
However, the robust pinball loss relies on matching expectations for both the
loss value and its gradient (Proposition 4.1). As a result, the theoretical guarantees
are asymptotic (e.g., convergence in expectation), and no finite-sample coverage
guarantee is preserved. This undermines one of the foundational motivations for
using the conformal prediction method.

---

> ### Author Rebuttal · Authors · 2025-07-31
>
> > The practical importance of noise robustness in conformal prediction [W1]
>
> Thank you for the suggestion. While most supervised learning tasks assume the provided labels reflect the ground truth, this assumption is often violated in the real world. For example, in medical imaging classification [4], label noise frequently emerges from subjective interpretations—for instance, when clinicians disagree on the malignancy of skin lesions. As introduced in the related work (Appendix A), the issue of label noise has been a common challenge in machine learning research, including conformal prediction [1-7]. In this work, we show that label noise can result in overly conservative prediction sets that include too many possible outcomes, thereby reducing their practical utility in decision-making (See Proposition 3.1 and Figure 1). This highlights the practical significance of addressing label noise in online conformal prediction. We will include the task importance in Section 2 of the final version.
>
> > The effect of errors in $\epsilon$ on the method [W2 & Q2]
>
> Thank you for raising the important point. In Appendix C, we evaluated the impact of misestimated noise rate on NR-OCP. Results in Figure 4 illustrate that overestimating the noise rate results in undercoverage, while underestimation yields conservative coverage with unnecessarily large prediction sets. Moreover, a small estimation error would reduce the coverage gap. Thus, it is recommended for practitioners to use a relatively low noise rate when the noise rate cannot be accurately estimated.
>
> In the literature of label-noise learning, many methods [8,9] leverage a small validation set of clean examples for estimating the noise rate. In real-world applications (including online settings), collecting a limited number of clean samples for noise rate estimation is often practicable. Additionally, some approaches [10,11] estimate the noise rate without requiring clean labels during training. Thus, such techniques make it reasonable to assume a known noise rate, which is widely recognized in the literature of label-noise learning [12-16].
>
> Moreover, we clarify that our method improves the performance even when the noise rate is not precisely estimated, as validated in Appendix C. To fully address this concern, we conduct an additional experiment where the noise rate is unknown and needs to be estimated. In particular, we apply an existing algorithm [8] to estimate the noise rate without access to clean data during the training of the pre-trained model. Then, we employ this estimated noise rate in our method for online conformal prediction. The experiments are conducted on CIFAR-10 dataset, using ResNet18 model with noise rate $\epsilon\in\{0.1,0.2,0.3\}$. We employ LAC score to generate prediction sets and use a constant learning rate $\eta=0.05$. The results are shown in the table below:
> | Noise rate | Estimated Noise rate | Method   | CovGap (%) | Size |
> |------------|----------------------|----------|------------|------|
> | $\epsilon=0.1$ | $\hat{\epsilon}=0.09$     | Baseline | 6.99       | 2.47 |
> |                |                     | Ours     | 0.82       | 0.94 |
> | $\epsilon=0.2$ |  $\hat{\epsilon}=0.23$  | Baseline | 8.72       | 5.41 |
> |                |                     | Ours     | 1.67       | 0.98 |
> | $\epsilon=0.3$ |  $\hat{\epsilon}=0.26$  | Baseline | 9.04       | 6.72 |
> |                |                     | Ours     | 2.32       | 1.15 |
>
> The results demonstrate that our method consistently achieves much lower coverage gaps and smaller prediction sets compared to the baseline. This highlights the robustness and applicability of our method even when the noise rate is not precisely estimated.
>
> > The limitation of the slow convergence rate [W3]
>
> We thank the reviewer for their insightful comment. As stated in Propositions 4.2 and 4.3, we prove that our robust method achieves convergence with a rate of  ($\mathcal{O}(T^{-1/2})$), which is slightly slower than standard online CP. However, we respectfully argue that the slightly slower convergence rate of our robust method is not a critical limitation in the context of online learning, and we provide the following clarifications to address this concern.
>
> First, our primary contribution is to eliminate the coverage gap under noisy conditions, where the standard online conformal method fails. The robust method is specifically designed to handle noisy settings, ensuring reliable convergence to the target coverage even in the presence of noise. As shown in Figure 2, our method converges rapidly to the target coverage in practice, highlighting the robust method’s advantages in noisy environments.
>
> Second, in online learning, the practical significance of a slightly slower convergence rate, such as $\mathcal{O}(T^{-1/2})$, is often outweighed by the need for robustness and adaptability. The robust method’s ability to maintain consistent performance across a range of noise levels offers a substantial advantage in terms of reliability and applicability. The trade-off in convergence speed is thus justified by the method’s ability to address challenging, non-idealized settings, which aligns with the primary motivation of our work.
>
> Lastly, we also emphasize that the theoretical results of convergence are presented to demonstrate the robustness and versatility of our approach, rather than to compete solely on convergence speed in idealized settings.
>
> > The method fails to provide the finite-sample coverage guarantee [W4]
>
> Thank you for raising this concern. We'd like to clarify that the framework of online conformal prediction (OCP) [17-20] aims to provide **asymptotic validity** in more general, potentially non-exchangeable, online settings, while split conformal prediction provides **finite-sample marginal coverage** under exchangeability assumptions. It's important to note that online conformal prediction makes no assumption on data distribution, which makes it impossible to achieve finite-sample coverage guarantee. Specifically, OCP aims to guarantee that the coverage error $|\frac{1}{T}\sum_{t=1}^T1\{Y_t\notin\mathcal{C}_t(X_t)\}-\alpha|$ converges to zero as $T\to\infty$, which is widely accepted in the online setting of conformal prediction. In this way, online conformal prediction can effectively handle dynamic environment where distribution shifts exist, while providing valid coverage guarantee [17].
>
> > The sample size for an accurate correction [Q1]
>
> Thank you for the question. In online learning, the sample size corresponds to the time step $T$, where a larger $T$ results in more accumulated samples. Theoretically, our method's convergence rate (see Propositions 4.2 and 4.3) requires $T \geq \frac{1}{(1-\epsilon)\beta^2}$ to achieve an error ExErr($T$)$\leq\beta$, where $\beta$ is an arbitrary small tolerance and $\epsilon$ is the noise rate. Experimentally, Figure 2 demonstrates that our method achieves a negligible coverage gap at approximately 2,000 time steps (or samples) with a constant learning rate and 1,000 time steps (or samples) with a dynamic learning rate.
>
> > Generalize to class-conditional or instance-dependent settings [Q3]
>
> We thank the reviewer for their insightful question. While our method does not directly extend to complex label noise settings, we emphasize that label noise remains a novel and underexplored challenge in conformal prediction, particularly in the online setting. As noted in Appendix A, the most relevant prior work [2] focuses solely on uniform noise in conformal prediction, only analyzing the overcoverage issue in online settings. In contrast, our work is the first to specifically tackle label noise in online conformal prediction, establishing a foundational step for future research in this area. We hope this highlights the significance of our contribution.
>
> ### References
> [1] Einbinder B S, et al. Conformal prediction is robust to label noise. arXiv preprint 2022.
>
> [2] Einbinder B S, et al. Label noise robustness of conformal prediction. JMLR 2024.
>
> [3] Sesia M, et al. Adaptive conformal classification with noisy labels. JRSSB 2024.
>
> [4] Penso C, et al. Noise-robust conformal prediction for medical image classification. International Workshop on Machine Learning in Medical Imaging. 2024.
>
> [5] Feldman S, et al. Robust conformal prediction using privileged information. NIPS 2024.
>
> [6] Penso C, et al. Estimating the Conformal Prediction Threshold from Noisy Labels. arXiv preprint 2025.
>
> [7] Bortolotti T, et al. Noise-Adaptive Conformal Classification with Marginal Coverage. arXiv preprint 2025.
>
> [8] Li X, et al. Provably end-to-end label-noise learning without anchor points. ICML 2021.
>
> [9] Xia X, et al. Are anchor points really indispensable in label-noise learning? NIPS 2019.
>
> [10] Yu X, et al. An efficient and provable approach for mixture proportion estimation using linear independence assumption. CVPR 2018.
>
> [11] Liu T, et al. Classification with noisy labels by importance reweighting. TPAMI 2015.
>
> [12] Han B, et al. Co-teaching: Robust training of deep neural networks with extremely noisy labels. NIPS 2018
>
> [13] Wei H, et al. Combating noisy labels by agreement: A joint training method with co-regularization. CVPR 2020.
>
> [14] Yu X, et al. How does disagreement help generalization against label corruption? ICML 2019.
>
> [15] Xia X, et al. Sample selection with uncertainty of losses for learning with noisy labels[J]. arXiv preprint 2021.
>
> [16] Xia X, et al. Combating noisy labels with sample selection by mining high-discrepancy examples. CVPR 2023.
>
> [17] Gibbs I, et al. Adaptive conformal inference under distribution shift. NIPS 2021.
>
> [18] Bhatnagar A, et al. Improved online conformal prediction via strongly adaptive online learning. ICML 2023.
>
> [19] Gibbs I, et al. Conformal inference for online prediction with arbitrary distribution shifts. JMLR 2024.
>
> [20] Angelopoulos A N, et al. Online conformal prediction with decaying step sizes. ICML 2024.

---

> > ### Comment · Reviewer_v3zM · 2025-08-06
> >
> > Thank you to the authors for addressing some of my earlier concerns. However, two major issues remain:
> >
> > 1. Known Noise Rate Assumption: The method relies on the assumption that the label noise rate is known, which is a strong and often unrealistic assumption in practice. In the measurement error literature, it is well established that when the error variance or rate must be estimated, the resulting uncertainty inflates the variance of downstream estimators and reduces statistical efficiency. Similarly, in conformal prediction, estimating the noise rate introduces additional variability that can lead to wider and less stable prediction sets. The paper does not account for how this estimation uncertainty propagates through the conformal prediction procedure or how it affects coverage, which limits the practical robustness of the proposed method.
> >
> > 2. Loss of Finite-Sample Guarantees: A key advantage of conformal prediction is its ability to provide finite-sample, distribution-free coverage guarantees. However, the theoretical foundation of this work relies on a robust pinball loss and expectations over both the loss and its gradient (as in Proposition 4.1), resulting in asymptotic guarantees rather than finite-sample ones. This undermines one of the core motivations for using conformal prediction in the first place.
> >
> > For these issues, I maintain my original score.

---

> ### Author Response · Authors · 2025-08-08
>
> Thank you for the detailed feedback. We are glad that some earlier concerns have been effectively addressed during our discussion. Below, we provide an in-depth response to the remaining points.
>
> > 1. How the estimation error affects the coverage.
>
> Thank you for highlighting the concern, also raised by Reviewers ZVis and SDzf. As noted in our response to W2&Q2, we can leverage established techniques from label-noise learning to estimate the noise rate effectively. In the rebuttal, we presented experiments with estimated noise rates demonstrating the practicality of this approach, which has addressed the concerns of Reviewers ZVis and SDzf. We agree that noise rate estimation may introduce uncertainty, potentially causing a coverage gap in conformal prediction. In Appendix C, we empirically evaluated the impact of misestimated noise rates on NR-OCP, showing that smaller estimation errors result in reduced coverage gaps.
>
> Here, we provide a theoretical analysis to explain this observation. Suppose the true noise rate is $\epsilon$, and the estimated noise rate is $\hat{\epsilon}$. For pinball loss $l_{1-\alpha}(\cdot,\cdot)$, we denote
> $$
> \tilde{l}\_{1-\alpha}(\tau,\tilde{S},\{S_y\}\_{y=1}^K)
> =\frac{1}{1-\epsilon}l\_{1-\alpha}(\tau,\tilde{S})
> -\frac{\epsilon}{K(1-\epsilon)}\sum_{y=1}^Kl\_{1-\alpha}(\tau,S_y).
> $$
> $$
> \tilde{l}^{'}\_{1-\alpha}(\tau,\tilde{S},\{S_y\}\_{y=1}^K)
> =\frac{1}{1-\hat{\epsilon}}l\_{1-\alpha}(\tau,\tilde{S})
> -\frac{\hat{\epsilon}}{K(1-\hat{\epsilon})}\sum\_{y=1}^Kl\_{1-\alpha}(\tau,S_y).
> $$
> We employ the following decomposition:
> $$
> EmErr(T)
> \leq\underbrace{\frac{1}{T}|\sum\_{t=1}^T\nabla_{\hat{\tau}\_t}l\_{1-\alpha}
> -\sum\_{t=1}^T\mathbb{E}[\nabla\_{\hat{\tau}\_t}l\_{1-\alpha}]|}\_{(a)}
> +\underbrace{\frac{1}{T}|\sum\_{t=1}^T\mathbb{E}[\nabla\_{\hat{\tau}\_t}\tilde{l}\_{1-\alpha}]
> -\sum\_{t=1}^T\nabla\_{\hat{\tau}\_t}\tilde{l}\_{1-\alpha}|}\_{(b)}
> +\underbrace{\frac{1}{T}|\sum\_{t=1}^T\nabla\_{\hat{\tau}\_t}\tilde{l}\_{1-\alpha}-\sum\_{t=1}^T\nabla\_{\hat{\tau}\_t}\tilde{l}^{'}\_{1-\alpha}|}\_{(c)}
> +\underbrace{\frac{1}{T}|\sum\_{t=1}^T\nabla\_{\hat{\tau}\_t}\tilde{l}^{'}\_{1-\alpha}|}\_{(d)}
> $$
>
> Azuma–Hoeffding inequality establishes that part (a) and (b) converge to zero with rate $\mathcal{O}(T^{-1/2})$, and part (d) achieves a convergence rate of $\mathcal{O}(T^{-1})$ following standard online conformal prediction theory. Then, we derive a bound for part (c):
> $$
> \frac{1}{T}|\sum\_{t=1}^T\nabla\_{\hat{\tau}\_t}\tilde{l}\_{1-\alpha}-\sum\_{t=1}^T\nabla\_{\hat{\tau}\_t}\tilde{l}^{'}\_{1-\alpha}|
> =\frac{1}{T}\sum\_{t=1}^T|\frac{1}{1-\epsilon}l\_{1-\alpha}(\hat{\tau}\_t,\tilde{S}\_t)
> -\frac{\epsilon}{K(1-\epsilon)}\sum\_{y=1}^Kl\_{1-\alpha}(\hat{\tau}\_t,S\_{t,y})
> -\frac{1}{1-\epsilon}l\_{1-\alpha}(\hat{\tau}\_t,\tilde{S}\_t)
> +\frac{\epsilon}{K(1-\epsilon)}\sum\_{y=1}^Kl\_{1-\alpha}(\hat{\tau}\_t,S\_{t,y})|
> \leq|\frac{1}{1-\epsilon}-\frac{1}{1-\hat{\epsilon}}|
> $$
> Now assume that $\hat{\epsilon}=\epsilon+\Delta$. Then, we have
> $$
> \frac{1}{T}|\sum\_{t=1}^T\nabla\_{\hat{\tau}\_t}\tilde{l}\_{1-\alpha}-\sum\_{t=1}^T\nabla\_{\hat{\tau}\_t}\tilde{l}^{'}\_{1-\alpha}|
> \leq |\frac{1}{1-\epsilon}-\frac{1}{1-\epsilon-\Delta}|.
> $$
> Therefore, we can conclude that
> $$
> \mathrm{EmErr(T)}\leq\mathcal{O}(T^{-1/2})+|\frac{1}{1-\epsilon}-\frac{1}{1-\epsilon-\Delta}|
> $$
> This result shows that a larger estimation error, $\Delta$, increases the coverage gap, consistent with our empirical observations. The theoretical analysis clearly illustrates how the estimation error impacts coverage, enhancing the understanding of our method in practical scenarios. Thank you again for the feedback, and we believe the additional analysis largely improves the quality and comprehensiveness of this work.
>
> > 2. Loss of finite-sample guarantees
>
> We thank the reviewer for noting this limitation. We clarify that **the asymptotic coverage guarantees are inherent to standard online conformal prediction (OCP)** and not introduced by our work.  Specifically, even with the clean pinball loss (assuming access to clean labels), standard online CP provides only asymptotic coverage in the online setting. This is a well-known property of OCP [1-4], and **our robust pinball loss does not alter this**. In particular, we can check the proof sketch of Prop. 4.2:
> * Part (a) quantifies the deviation between the gradients of the robust and clean pinball losses. It converges to zero at a rate of $\mathcal{O}(T^{-1/2})$, ensuring our method addresses the problem of label noise.
> * Part (b) leverages **standard results from online CP**, which yield **asymptotic coverage guarantees** with a convergence rate of $\mathcal{O}(T^{-1})$, as shown in prior work [1,2].
>
> In this work, we aim to address the label noise issue in OCP, which inherently provides only asymptotic guarantees. While exploring finite-sample guarantees for OCP is a promising future direction, this does not diminish the significance of our contributions in this work.

---

> > ### Author Response · Authors · 2025-08-08
> >
> > ### References
> > [1] Gibbs I, et al. Adaptive conformal inference under distribution shift. NIPS 2021.
> >
> > [2] Angelopoulos A N, et al. Online conformal prediction with decaying step sizes. ICML 2024.
> >
> > [3] Bhatnagar A, et al. Improved online conformal prediction via strongly adaptive online learning. ICML 2023.
> >
> > [4] Gibbs I, et al. Conformal inference for online prediction with arbitrary distribution shifts. JMLR 2024.

---

### Note · Authors · 2025-08-13

We express our sincere gratitude to the Area Chair (AC) and all reviewers for their time and constructive feedback, which have significantly enhanced the quality and clarity of our work. We are encouraged that most concerns have been well addressed, with multiple reviewers recognizing the **novelty**, **elegance**, **theoretical rigor**, and **consistent empirical performance**. Below, we summarize how we addressed the main concerns raised:
* Assumption of knowing the noise rate: This assumption is practical, as existing label-noise learning techniques enable reliable noise rate estimation. To address concerns, we conducted additional experiments using estimated noise rates, demonstrating that our method substantially reduces the coverage gap without requiring precise noise ratios. **These results have satisfied Reviewers ZVis and SDzf**, and we will incorporate them into all main figures and tables in the final manuscript as recommended. Furthermore, in response to Reviewer v3zM, we provided a comprehensive theoretical analysis to elucidate how estimation errors impact coverage, aligning with our empirical findings in Appendix C. This combined evidence underscores the robustness and reasonability of our approach.
* Lack of finite-sample coverage guarantee in online CP: We clarified that the asymptotic nature of coverage guarantees is inherent to standard online CP due to the unpredictable, non-exchangeable nature of future data in online settings. Our method does not introduce this limitation but instead maintains the same theoretical guarantees as standard online CP, while addressing the practical challenge of label noise through our robust pinball loss.

In summary, our approach represents **the first step to address label noise in online conformal prediction**. Our robust loss seamlessly integrates with existing online CP algorithms and non-conformity scores, highlighting its broad applicability. We believe this work lays a strong foundation for future methods designed to handle more general noise mechanisms, such as class-conditional noise. We once again thank the AC and reviewers for their insightful feedback, which has significantly strengthened this work.

---

### Decision · Program_Chairs · 2025-09-17

**Decision:**

Accept (poster)

**Comment:**

This paper investigates the robustness of online conformal prediction methods in the presence of uniform label noise. The authors show that standard online conformal algorithms, including ACI and its variants, can suffer from systematic coverage distortion when labels are noisy, due to biased gradient updates caused by corrupted supervision. To mitigate this issue, they propose a robust pinball loss that serves as an unbiased estimator of the true clean pinball loss.

I thank the authors for their thorough responses and additional experimental results. Based on the reviewers' discussions (including those with the authors) and my own evaluation, I believe the results of this paper are novel and of interest to the ICML community. While some reviewers expressed concerns that the work may be quite limited to this specific noise mechanism, I believe that some assumption about the noise is indeed necessary, and uniformity is a reasonable choice. However, in my opinion, the assumption that the noise power is known precisely might be too strong and somewhat limiting. Therefore, I would recommend that the authors explore whether knowing only an upper bound on the variance would suffice for their algorithms and results.